# Heme-deficient metabolism and impaired cellular differentiation as an evolutionary trade-off for human infectivity in *Trypanosoma brucei gambiense*

Resistance to African trypanosomes in humans relies in part on the high affinity targeting of a trypanosome lytic factor 1 (TLF1) to a trypanosome haptoglobin-hemoglobin receptor (HpHbR). While TLF1 avoidance by the inactivation of HpHbR contributes to *Trypanosoma brucei gambiense* human infectivity, the evolutionary trade-off of this adaptation is unknown, as the physiological function of the receptor remains to be elucidated. Here we show that uptake of hemoglobin *via* HpHbR constitutes the sole heme import pathway in the trypanosome bloodstream stage. *T. b. gambiense* strains carrying the inactivating mutation in HpHbR, as well as genetically engineered *T. b. brucei* HpHbR knock-out lines show only trace levels of intracellular heme and lack hemoprotein-based enzymatic activities, thereby providing an uncommon example of aerobic parasitic proliferation in the absence of heme. We further show that HpHbR facilitates the developmental progression from proliferating long slender forms to cell cycle-arrested stumpy forms in *T. b. brucei*. Accordingly, *T. b. gambiense* was found to be poorly competent for slender-to-stumpy differentiation unless a functional HpHbR receptor derived from *T. b. brucei* was genetically restored. Altogether, we identify heme-deficient metabolism and disrupted cellular differentiation as two distinct HpHbR-dependent evolutionary trade-offs for *T. b. gambiense* human infectivity.

Through the combination of human infections and livestock trypanosomiasis, the neglected tropical diseases caused by African trypanosomes belonging to *Trypanosoma brucei sensu lato* (*s.l.*) have a significant impact on sub-Saharan rural development. Two sub-species of *T. brucei* proliferate in humans: *T. brucei gambiense* is responsible for chronic and slowly progressing human trypanosomiasis, while *Trypanosoma b. rhodesiense* causes an acute form of the disease[1]. When injected by the blood feeding insect vector (tsetse fly; *Glossina* spp.) into the human tissue, animal-infecting *T. brucei brucei* is rapidly killed by a potent arm of the innate immune system, represented by trypanosome lytic factors (TLF) 1 and 2[2,3]. Both factors are high-density lipoprotein complexes containing haptoglobin-related protein[4] and apolipoprotein L-1[5]. In addition, TLF2 contains IgM molecules through which its uptake is mediated[6,7]. TLF1 targets the parasites efficiently by engaging the haptoglobin-hemoglobin receptor (HpHbR), one of the few cell surface receptors known to date in the kinetoplastid flagellates[8–10].

The principal function of HpHbR, which was named after its only ligand, is heme uptake[8]. The HpHb complexes are formed by binding of haptoglobin (Hp) and extracellular hemoglobin (Hb) resulting from intravascular hemolysis[11]. Indeed, same as other trypanosomes, *T. brucei s.l.* are heme auxotrophs[12], who in the mammalian-infective

✉ e-mail: horakova@paru.cas.cz; jula@paru.cas.cz

bloodstream stage (BS) acquire external heme via HpHbR[8]. The procyclic stage (PS) in the tsetse fly midgut contains more hemoproteins and obtains the ancient and omnipresent heme cofactor using another dedicated transporter, *Tb*Hrg[13,14]. The *Tb*Hrg transcript is developmentally regulated with the highest expression in the PS cells residing in the tsetse posterior midgut, gradually decreasing in the subsequent life cycle stages[13].

Nevertheless, HpHbR is not essential for the proliferation of BS since the monomorphic *T. b. brucei* knock-out for HpHbR shows only a mildly affected growth phenotype in vitro[8,15]. Moreover, despite being more sensitive to the host's oxidative stress, this cell line kills its rodent host before the first wave of immunoglobulin-based immunity develops[8]. Curiously, the capacity of *T. b. gambiense* to survive in human blood and cause infection is partially based on a point mutation in HpHbR that dramatically reduces its affinity for both TLF1 and HpHb[15–18]. Decreased uptake of TLF1 into *T. b. gambiense* is additionally modulated by lower HpHbR transcript levels than the corresponding receptors in *T. b. brucei* and *T. b. rhodesiense*[19].

HpHbR expression contributes to *T. b. brucei*'s fitness in its animal reservoir hosts, providing positive selection pressures for the conservation of this receptor[8,20]. Still, HpHbR acquired critical mutations that allowed *T. b. gambiense* to enter a new niche, the human host, despite the parallel attenuation in the primary animal hosts. The cost of this loss of HpHbR function was tolerable, although it potentially brought a reduced capacity for cyclical transmission by the tsetse fly when compared with the closely related *T. b. brucei* and *T. b. rhodesiense*[21].

Here we confirm that *T. b. brucei* HpHbR KO does not uptake heme and provide new evidence that this leads to the loss of ability to fuel hemoproteins with this cofactor. Moreover, the loss of this receptor is seemingly associated with a reduced ability to undergo differentiation in the mammalian host. In the absence of HpHbR, the fast dividing long slender forms of the bloodstream stage (BS-SL) do not transform into the quiescent, transmission-competent stumpy forms of the bloodstream stage (BS-ST). As a natural mutant for HpHbR, the human pathogenic *T. b. gambiense* is poorly capable of importing heme and generating BS-ST, while both of these key features are restored by heterologous expression of the *T. b. brucei*-derived HpHbR.

## Results

### Heme-free trypanosomes

*T. brucei s. l.* lost the capacity to synthesize heme and therefore has to acquire it from its hosts[12]. First, we confirmed previous fluorimetric measurements of cellular heme content[8] by a more sensitive method. Wild type (WT) *T. b. brucei* contains 100 pmol heme/$10^9$ cells, contrary to the *T. b. brucei* HpHbR KO cells, which exhibit only a trace amount of heme at the detection limit (Fig. 1a, b). For the first time, we established the amount of heme in WT *T. b. gambiense*, where a trace of heme was observed (6 pmol/$10^9$ cells), comparable to HpHbR KO cells (Fig. 1a, b).

Next, we studied how hemoproteins function under these conditions. Therefore, a V5-tagged human catalase (hCAT), a potent heme-dependent enzyme, is absent in the genome of *T. brucei s. l.*[22], was expressed in three cell lines, namely WT *T. b. brucei*, the derived HpHbR KO, and WT *T. b. gambiense* (Fig. 1c, d). The ectopically-expressed catalase, which showed a uniform distribution in the cytosol (Fig. 1e), is used as a sensor for cytosolic heme. Catalase activity was readily monitored via the production of molecular oxygen, forming visible bubbles in the cell suspension (Fig. 1f; inset) upon the addition of $H_2O_2$ as a substrate (Fig. 1c). This assay showed that catalase was active in WT *T. b. brucei* while inactive in both *T. b. brucei* HpHbR KO and *T. b. gambiense* (Fig. 1f). These results are best explained by the failure of the latter trypanosomes to import heme due to the absence of a functional HpHbR. In addition, we also demonstrated that when taken up by HpHbR, heme is delivered into the cytosol, where it is freely available for hemoproteins as a cofactor.

### Endogenous hemoprotein activity depends on HpHbR

Being involved in the sterol metabolism[23,24], the endogenous hemoprotein CYP51 is considered to be essential and, therefore, a promising drug target against different trypanosomatids[25–27]. Ketoconazole is one of the broadly used compounds, which binds to the active site of CYP51 and inhibits its activity[28]. The genome of *T. brucei s. l.* encodes a single copy of CYP51, which is highly transcribed in PS but is barely detectable in BS-SL and BS-ST (Fig. 2a). RNAi in PS led to an efficient reduction of *CYP51* mRNA (Supplementary Fig. 1A) followed by an almost complete absence of the corresponding protein 5 days post-induction (Supplementary Fig. 1B), as also illustrated by reduced α-CYP51 immunostaining in the RNAi-induced cells (Supplementary Fig. 1C). The gradual loss of CYP51 was associated with a significant growth reduction of the RNAi-interfered PS cells (Supplementary Fig. 1D). Of note, in PS we were unable to generate CYP51 KO cell lines, suggesting this protein is essential for this life cycle stage. In contrast to the insect stage parasites, we were able to generate CYP51 KO in the BS cells by homologous recombination of both alleles with hygromycin and phleomycin expression cassettes (Supplementary Fig. 1E). The growth phenotype of BS lacking CYP51 was only slightly affected, indicating that this protein is dispensable in this life stage (Fig. 2b).

We evaluated the sensitivity of WT and CYP51 KO cells to ketoconazole, a specific inhibitor of CYP51 (Fig. 2c). In agreement with the previously published $IC_{50}$ values[29], there was no statistically significant difference between the parental WT and CYP51 KO cells (Fig. 2c). Still, concentrations of ketoconazole ranging from 2 to 8 μM affect the WT and CYP51 KO cells differentially, since the former cells reduced their growth rate, and the latter remained unaffected (note the biphasic behavior of the dose-response curve) (Fig. 2c). Hence, CYP51 activity can be selectively inhibited with low doses of ketoconazole, discriminating between the cells with and without CYP51. In contrast, when higher doses of ketoconazole are applied, we do not see significant differences in the growth of the WT and CYP51 mutant cells, and the action of the drug should be assigned to the off-target effect.

### HpHb uptake deficiency confers insensitivity to CYP51 inhibition

The pharmacological conditions described above allow assessing the activity of the hemoprotein CYP51 in the BS cells under the conditions of defective HpHb uptake (Fig. 2d). First, we exposed the HpHbR KO cells to ketoconazole and showed they are insensitive to low doses of the drug (Fig. 2e), mimicking the phenotype observed for the CYP51 KO trypanosomes (Fig. 2c).

Next, the HpHbR pathway-dependent CYP51 activity was evaluated in the WT *T. b. rhodesiense* resistant to the lysis by the human serum independently of HpHb uptake[30]. Under different cultivation conditions, we were able to modulate their access to the receptor's cognate ligand. When grown in the Hp-containing human serum, in which the heterodimeric HpHb ligand is formed, *T. b. rhodesiense* is sensitive to micromolar concentrations of ketoconazole (Fig. 2f). In contrast, when grown in the anhaptoglobinemic serum[31,32], i.e., in the absence of Hp, cells exhibit reduced growth and become insensitive to the drug. Yet, complementation of the anhaptoglobinemic serum with purified human Hp reverted this phenotype (Fig. 2f), proving a positive correlation between the operational HpHb uptake and the CYP51 activity. Combined, we showed that CYP51 is a genuine hemoprotein and a downstream acceptor of the HpHb complex.

### Artificial expression of HpHbR in stumpy forms does not interfere with life cycle progression

Due to RNA polymerase I-mediated polycistronic transcription, in trypanosomes, most regulation occurs post-transcriptionally[33]. It was shown previously that the HpHbR expression is downregulated in the early phase of the BS-SL to BS-ST differentiation[34]. We noticed that *HpHbR* is located at the very end of the polycistronic transcription unit. Therefore, we modified pleomorphic WT *T. b. brucei* to a cell line,

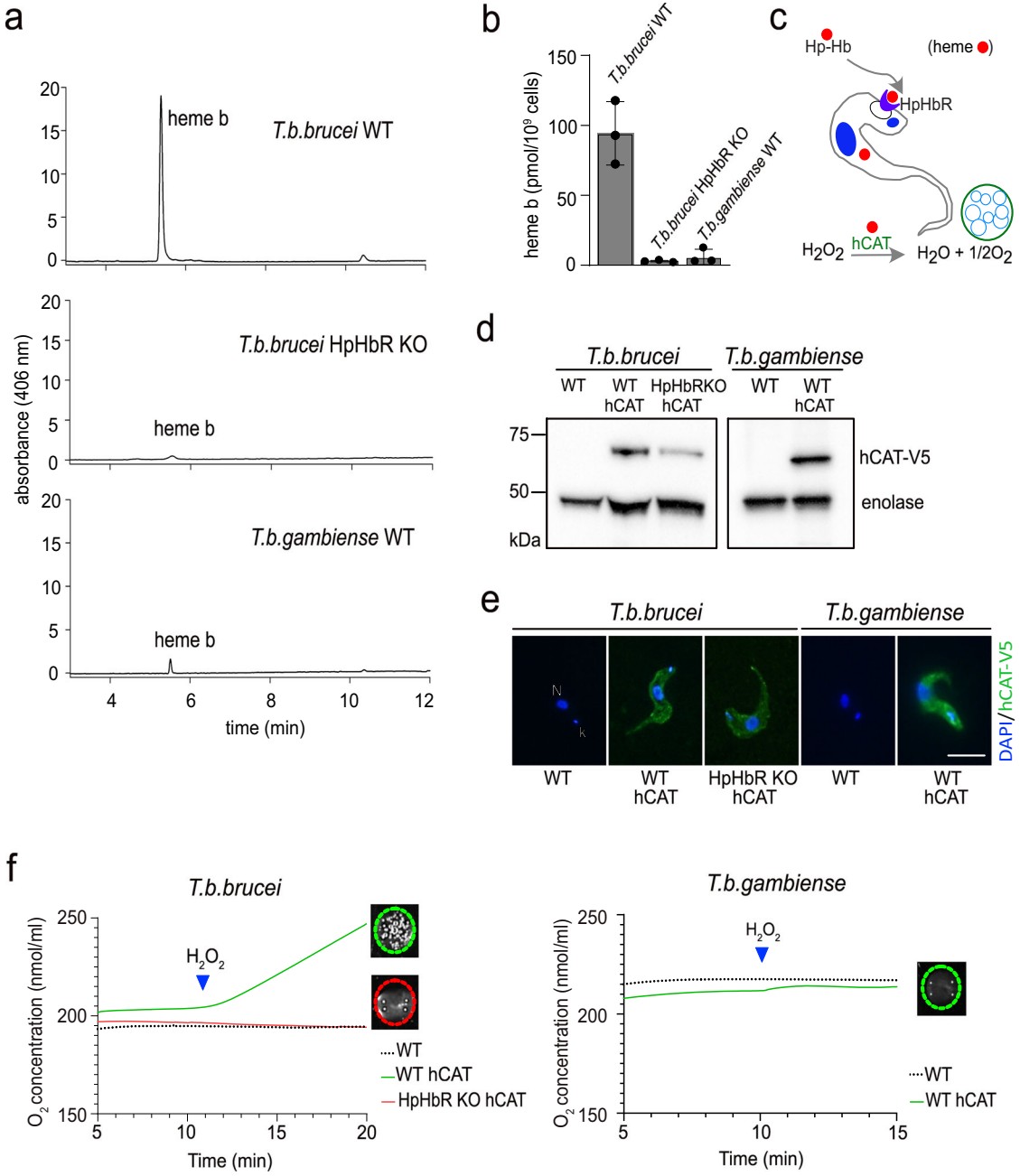

**Fig. 1 | Detection of heme and hemoproteins in bloodstream stages of *T. b. brucei* and *T. b. gambiense*. a** Heme *b* extracted from $1 \times 10^9$ cells was separated by HPLC and detected by a diode array detector. Representative chromatogram from wild type *T. b. brucei*, *T. b. brucei* knock-out for HpHbR and wild type *T. b. gambiense*. **b** Graph showing quantification of total heme *b* content in the same cell lines as in (**a**); ($n = 3$). **c** Schematic representation of the experimental design used to measure the activity of the N-terminally V5-tagged human catalase (hCAT) in the bloodstream *T. brucei*. **d** Western blot analysis with V5 antibody detecting human catalase (hCAT) in WT and HpHbR knock-out (KO) of *T. b. brucei* and WT *T. b.* *gambiense* overexpressing hCAT. Enolase was used as a loading control. **e** Immunofluorescence of hCAT detected with V5 antibody (green) in the same cell lines as in (**d**). DNA in the nucleus (N) and kinetoplast (k) was stained with DAPI (blue). Scale bar, 5 μm. **f** Measurement of human catalase activity using Oroboros oxygraph in *T. b. brucei* WT and HpHbR KO (left) and *T. b. gambiense* (right); ($n = 3$). Visual verification of measured activity via the $O_2$ production in the form of bubbles after the addition of 3% $H_2O_2$ is shown in the insets. Error bars indicate ±SD. Source data are provided as a Source Data file.

where the distance of *HpHbR* from the end of the polycistron was artificially increased by the insertion of a 10 kb-long *luciferase* gene in front of the *procyclin* gene (HpHbR-Luc) (Supplementary Fig. 2A). By following the expression of both *HpHbR* and *luciferase* genes during in vitro differentiation, we detected that the *luciferase* mRNA was continuously formed (Supplementary Fig. 2B), which was verified by luciferase activity measurements (Supplementary Fig. 2C). In contrast, the HpHbR transcript was downregulated in the HpHbR-Luc flagellates to the same extent as in the WT cells during the BS to PS differentiation

(Supplementary Fig. 2D). This data suggests that the distance from the end of the polycistronic unit is not a critical factor in orchestrating the transcription efficiency during life cycle progression.

It is well known that the 3′ untranslated regions (UTRs) have a critical role in the regulation of transcript stability in trypanosomes[33]. We forced the expression of *HpHbR* in BS-ST by fusing the *HpHbR* open reading frame to the 3′ UTR of *PAD1* gene (Supplementary Fig. 2E), the product of which is exclusively expressed in BS-ST[35]. We assumed that the artificial expression of

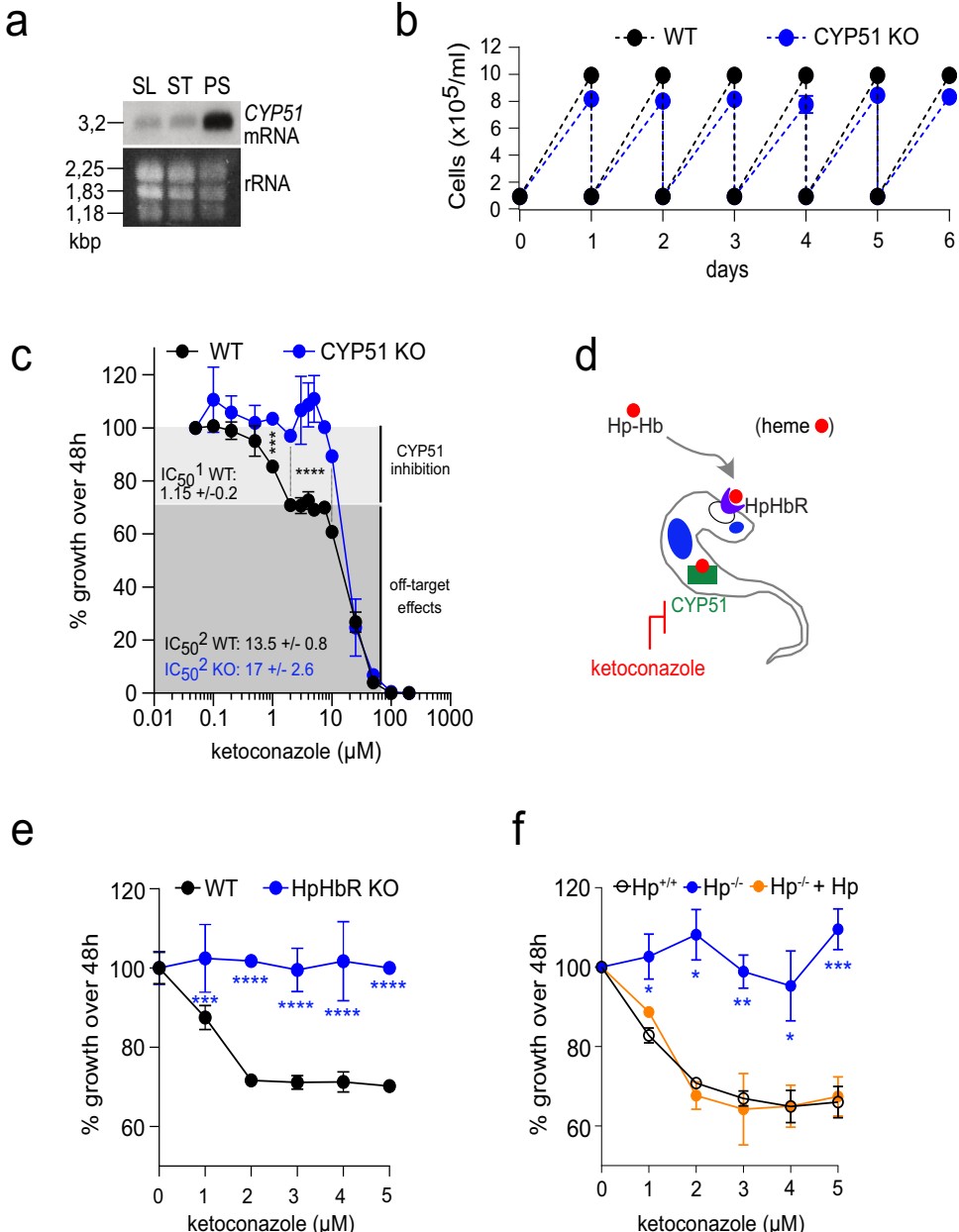

**Fig. 2 | Effect on growth and ketoconazole sensitivity after CYP51 invalidation in the bloodstream stage. a** Northern blot analysis showing stage-specific expression of *CYP51* mRNA. Total RNA from long slender (SL) and stumpy (ST) bloodstream stage, and procyclic stage (PS) was hybridized with the *Tb*CYP51 DNA probe. The staining of ribosomal (r)RNA was used as a loading control. **b** Growth curves of WT *T. b. brucei* (black dots and line) and CYP51 knock-out (CYP51 KO) (blue dots and line) for 6 days. **c** The depletion of CYP51 confers insensitivity to low doses of ketoconazole. WT (black dots and line) and CYP51 KO cells (blue dots and line) were grown under the same conditions. A light gray zone indicates concentrations of ketoconazole with an inhibitory effect on CYP51, and a dark gray zone indicates a concentration range with an off-target effect ($n = 3$). Sidak's multiple comparisons test *** $p < 0.001$; **** $p < 0.0001$. **d** Schematic representation

of the experimental design showing the inhibition of hemoprotein CYP51 activity by ketoconazole. **e** Deficiency in the haptoglobin-hemoglobin (HpHb) uptake results in mild growth phenotype and insensitivity to the CYP51 inhibitor. WT *T. b. brucei* and HpHbR knock-out (KO) were incubated for 48 h with 0 to 5 µM of ketoconazole ($n = 2$). Sidak's multiple comparisons test *** $p < 0.001$; **** $p < 0.0001$. **f** *T. b. rhodesiense* Etat 1.2R cells freshly isolated from mice and transferred to in vitro culture conditions with either normal human serum containing HpHb (Hp+/+), human serum lacking Hp (anhaptoglobinemic) (Hp−/−), or anhaptoglobinemic serum complemented with purified human Hp (Hp−/− + Hp). Their growth rate was determined after incubation for 48 h with 0 to 5 µM of ketoconazole ($n = 3$). Dunnet's multiple comparison test * $p < 0.1$; ** $p < 0.01$; *** $p < 0.001$. Error bars indicate ±SD. Source data are provided as a Source Data file.

HpHbR and subsequent continuous import of heme into cell cycle-arrested BS-ST could be detrimental.

In this HpHbR-3′PAD1 cell line, the *HpHbR* expression is maintained until the in vitro differentiation into the PS flagellates, which express 5x more *HpHbR* mRNA when compared to the WT PS (Fig. 3a). The capacity of the HpHbR-3′PAD1 cell line to differentiate in vivo from BS-SL to BS-ST was evaluated in mice, where

the typical parasitic wave characteristic for the WT trypanosomes was produced (Fig. 3b). Moreover, ex vivo cells collected on day 4 were examined morphologically, and the functionality of HpHbR was verified by the uptake of the fluorescently-labeled HpHb complexes (Fig. 3c, d). As expected, 75% of the WT cells transformed into the BS-ST trypanosomes, which were exclusively Hp-free. The HpHbR-3′PAD1 cells also retained the ability to form BS-

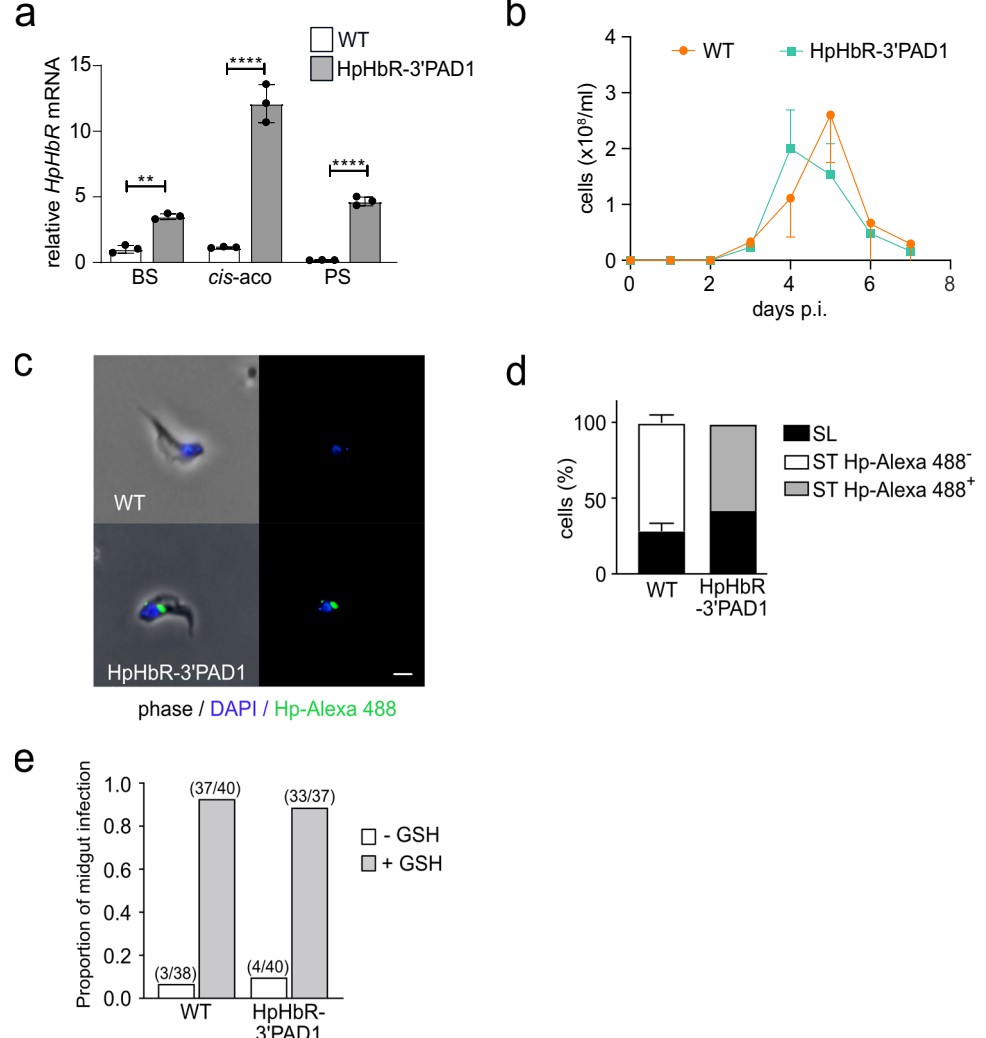

**Fig. 3 | Artificial expression of HpHbR in stumpy form does not interfere with life cycle progression. a** The expression of HpHbR was assessed as in (**b**) in the WT *T. b. brucei* and HpHbR-3'PAD1 trypanosomes. RNA was extracted from the BS, 2 h after differentiation was triggered by *cis*-aconitate (*cis*-aco) and the procyclic stage (PS). Tukey's multiple comparison test, n = 3. Error bars indicate ±SD. **b** In vivo infectivity of WT *T. b. brucei* and HpHbR-3'PAD1 cells was evaluated by infecting mice with 1 × 10⁴ cells. The parasitemia was counted daily till day 7. **c** A representative WT *T. b. brucei* and HpHbR-3'PAD1 cells were isolated from the blood of mice; the latter ones internalized the fluorescently-labeled HpHb complex (green). Scale bar, 3 μm. **d** Based on morphology and the HpHb complex uptake, the same cells as in (**e**) were categorized as slender (SL) or stumpy (ST), the latter with the HpHb complex internalized (Hp-Alexa 488+) or not (Hp-Alexa 488−). **e** The capacity to establish midgut infection in tsetse flies was determined for WT *T. b. brucei* and HpHbR-3'PAD1 cells isolated from the blood of mice. Bloodstream stage parasites were administrated in the fly's first bloodmeal without (-GSH) or with 10 mM glutathione (+GSH) supplementation. Source data are provided as a Source Data file.

ST (~60% by day 4), all of which were Hp-positive due to the artificial expression of the receptor (Fig. 3c, d).

Since the ability of the HpHbR-3'PAD1 cell line to differentiate has been tested only in the mammalian host, we decided to also evaluate its capacity for transmission via tsetse flies. The midguts of flies fed with blood containing either the WT or HpHbR-3'PAD1 BS parasites were dissected 10 days post-feeding, showing no significant difference in the infection rates (Fig. 3e). Therefore, we added reduced L-glutathione to the blood meal, enhancing the establishment of trypanosomes in the tsetse midgut[36]. Indeed, the infection rate reached up to 90%, but we did not observe a significant difference between the ability of WT and HpHbR-3'PAD1 cells to transform into PS and establish the tsetse midgut infection (Fig. 3e). Altogether, these results show that trypanosomes with the prolonged expression of HpHbR still differentiate into BS-ST and subsequent PS and retain the ability to infect tsetse flies. It also suggests that reducing heme influx into the cell (due to a missing HpHbR) does not represent a signal for differentiation into BS-ST.

## *T. b. brucei* HpHbR KO does not produce the stumpy form

The BS trypanosomes undergo extensive cellular differentiation in preparation for an abrupt transmission from the mammalian blood into tsetse fly[37]. As the intensity of infection increases through the rapid proliferation of BS-SL, a parasite-derived stumpy induction factor accumulates, promoting morphological transformation into BS-ST[38].

In order to test whether HpHbR plays any role in this critical phase of the life cycle, the HpHbR KO cells have also been generated in the pleomorphic 90-13 *T. b. brucei* by replacing both alleles with the puromycin or phleomycin cassette (Supplementary Fig. 3A, B). Moreover, we have created addback cell lines with the *T. b. brucei* HpHbR (ABb¹⁸ˢ; Supplementary Fig. 3A, C) and *T. b. gambiense* HpHbR (ABg¹⁸ˢ; Supplementary Fig. 3A, C) restored from the 18S rRNA locus. In another cell line, a single HpHbR allele was restored in situ (ABbⁱⁿ ˢⁱᵗᵘ) with more physiological levels of *T. b. brucei* HpHbR (Supplementary Fig. 3A, C).

First, the functionality of HpHbR in WT and the genetically engineered *T. b. brucei* cell lines was examined by fluorescence microscopy

of Hp labeled with a green fluorochrome Hp-Alexa 488. As expected, the uptake of the HpHb complex was disrupted entirely in the HpHbR KO cell line, while 80% of the WT cells were Hp-positive (Fig. 4a, b). Both above-described addback cell lines showed the re-establishment of the Hp uptake, although to a different extent. Overexpression of HpHbR from the 18S rRNA locus almost reached the WT values (70%), whereas only 20% of ABb$^{in situ}$ cells were labeled (Fig. 4a, b).

Next, the capacity to undergo differentiation in vitro was analyzed by exposing the individual cell lines to *cis*-aconitate and a temperature decrease to 27 °C, which is known to trigger the BS to PS transformation[39–41]. We followed their ability to proliferate and express the procyclin coat as a hallmark of PS (Fig. 4c, d). Under these conditions, the majority of WT cells (70%) became procyclin-positive by day 2 and reached high densities (mean value $6 \times 10^6$ cells/ml) by day 3 (Fig. 4c, d). In contrast, the HpHbR KO cells differentiated into only a few PS cells that did not divide and eventually died (Fig. 4c, d). Both AB cell lines showed a pretty decent capacity to differentiate in vitro, with 70% of cells covered by procyclin coat by day 2 (Fig. 4d). Although the cell numbers were not reaching the WT values by day 3, the cell line with higher expression of HpHbR (ABb$^{18S}$) did not statistically differ from WT (Fig. 4c).

Individual cell lines were analyzed for their capacity to differentiate in vivo. We followed their growth in the mouse model, where the HpHbR KO parasites initially proliferated somewhat slower as compared to WT (Fig. 4e). However, on day 6 the infection rapidly accelerated, achieving a high parasitemia of ~$3 \times 10^8$ cells/ml, leading to the termination of the experiment on the following day (Fig. 4e). In contrast, the WT (90–13) parasitemia declined on day 6, whereas the ABb$^{in situ}$ cells showed intermediate phenotypes, reaching a plateau on day 7 (Fig. 4e). Notably, the addbacks driven from the 18S locus gave different parasitemic profiles. When compared with each other, ABb$^{18S}$ caused a relatively low and delayed parasitic wave, while the infection with ABg$^{18S}$ resulted in a fast progression of the infection resembling the HpHbR KO phenotype (Supplementary Fig. 4A).

The blood-harvested parasites were further examined for PAD1 expression using immunofluorescence microscopy and western blot analysis (Fig. 4f–i). Since the PAD1 protein is specifically expressed on the surface of BS-ST and is prominently absent from BS-SL, it is used as a molecular marker for the former morphotype[42]. At the same time, we followed characteristic morphological features, such as the cell volume and the distance between the kinetoplast and the nucleus (Fig. 4i). We detected only a negligible number of PAD1 positives (1%) among the HpHbR KO trypanosomes, which is in line with their exclusive BS-SL morphology (Fig. 4f, g). On the contrary, 97% of the WT cells were PAD1-positive (Fig. 4g, h) and had the characteristic BS-ST morphology, associated with larger cell volume and shorter distance between the kinetoplast and the nucleus (Fig. 4i). Additionally, ABb$^{in situ}$ also expressed the PAD1 protein, although to a lesser extent (30%) (Fig. 4g, h), which was accompanied by the intermediate and BS-ST phenotype (Fig. 4f). On the contrary, the *T. b. gambiense* HpHbR addback (ABg$^{18S}$) failed to rescue HpHbR KO (Fig. 4g, h; Supplementary Fig. 4A), further supporting the importance of a fully functional receptor for the stumpy differentiation.

### Overexpression of *T. b. brucei* HpHbR in *T. b. gambiense* increases the stumpy formation

We studied the consequences of a restored HpHbR expression in *T. b. gambiense* for its life cycle progression in the mammalian host. At first, we used the laboratory strain of *T. b. gambiense* (LiTat1.3 strain) and *T. b. gambiense* expressing *T. b. brucei* HpHbR from the 18S rRNA locus (*T. b. gambiense* + b$^{18S}$), as described previously[18]. Both cell lines were injected into BALB/c mice and the resulting parasitemia was followed on a daily basis. The *T. b. gambiense* parasites emerged in the bloodstream between days 2 and 3 and sustained a rather mild infection (maximum of $3 \times 10^7$ cells/ml was reached on day 4), and on day 5

no trypanosomes were observed in the blood smears (Fig. 5a). *T. b. gambiense* expressing HpHbR of *T. b. brucei* triggered a yet significantly weaker infection, with cells detectable only until day 4, when they peaked at $5 \times 10^6$ cells/ml (Fig. 5a).

In *T. b. gambiense*, we detected only a negligible amount of the PAD1-expressing cells, while the picture was strikingly different for *T. b. gambiense* + b$^{18S}$ trypanosomes. At the peak of the infection on day 4, about 25% of cells were PAD1-positive (Fig. 5b, c), and this was associated with a significant repositioning of the nucleus towards the kinetoplast (Fig. 5d). In contrast, there was no pronounced increase in the cell volume (Fig. 5d), a feature typical for the PAD1-expressing BS-ST *T. b. brucei*.

Recent observations showed that stress from drug treatment and/ or gene expression changes might trigger pathways mimicking the canonical differentiation to BS-ST[43]. The overexpression of *T. b. brucei* HpHbR in *T. b. gambiense* (*T. b. gambiense* + b$^{18S}$; Supplementary Fig. 3D) produces a rather artificial amount of the protein (50× higher than the WT level)[18]. To rule out the possibility that the increased expression of PAD1 is caused by stress, we generated a cell line in *T. b. gambiense*, where its genuine HpHbR is overexpressed (*T. b. gambiense* + g$^{18S}$). This cell line overexpresses *T. b. gambiense* HpHbR approximately 80x more than the parental strain (Supplementary Fig. 3d), hence with values similar to the *T. b. gambiense* + b$^{18S}$ parasites. When we compared both overexpressing cell lines in vivo, they showed the same profile with a low parasitemic wave peaking on day 4 (Fig. 5a). Importantly, a morphological examination on day 4 revealed only negligible presence of PAD1-positive *T. b. gambiense* + g$^{18S}$ (Fig. 5c), indicating that only *T. b. brucei* HpHbR enhances the stumpy formation in *T. b. gambiense*.

### Variable expression of stumpy marker genes in *T. b. gambiense* group

Finally, we wondered whether the *T. b. gambiense* field isolate Bosendja sustains the ability to produce waves of parasitemia. Due to its limited viability in the axenic culture, Bosendja cells were injected directly from the blood stabilates into the BALB/c mice. The parasites proliferated rapidly in the bloodstream, reaching over $3 \times 10^8$ cells/ml on day 4, when the experiment was terminated (Fig. 5e). Careful morphological inspection categorized most trypanosomes as BS-SL, with a pronounced undulating membrane (Fig. 5f). Although in a very few cells (0.1%), the expression of PAD1 was detected on day 4 by immunofluorescence (Fig. 5f, g), the low amount of this marker protein remained undetectable by western blot analysis, where the *T. b. brucei* WT and the ABb$^{in situ}$ cells served as positive controls (Fig. 5h).

To further investigate whether the reduced stumpy formation is a general feature of *the T. b. gambiense* group, a laboratory strain Litat1.3 and two field isolates Bosendja and PA were compared to *T. b. brucei* BS-LS and BS-ST controls. We followed by qPCR the expression of several genes known to be up-or downregulated[44] in BS-ST (Supplementary Fig. 5). The analysis revealed that the picture is rather complex since Bosendja strain matched more BS-SL profile, unlike LiTat 1.3 and PA strains, which resembled BS-ST. Therefore, we conclude that *T. b. gambiense* produces bloodstream cells that do not fully correspond to either of these developmental stages, with the appearance of the stumpy-like cells being rather strain-dependent.

## Discussion

The heme group is needed to sustain the life of almost all eukaryotic organisms analyzed so far[45,46]. Nevertheless, it has remained unclear whether trypanosomes still require heme while residing in the mammalian host. Indeed, many hemoproteins are missing or are downregulated in the BS *T. b. brucei*. To provide heme for its hemoproteins, such as the subunits of respiratory complexes and enzymes involved in the sterol synthesis[24], the BS and PS *T. b. brucei* scavenge heme from their hosts via the HpHbR and *Tb*Hrg receptors, respectively[8,13].

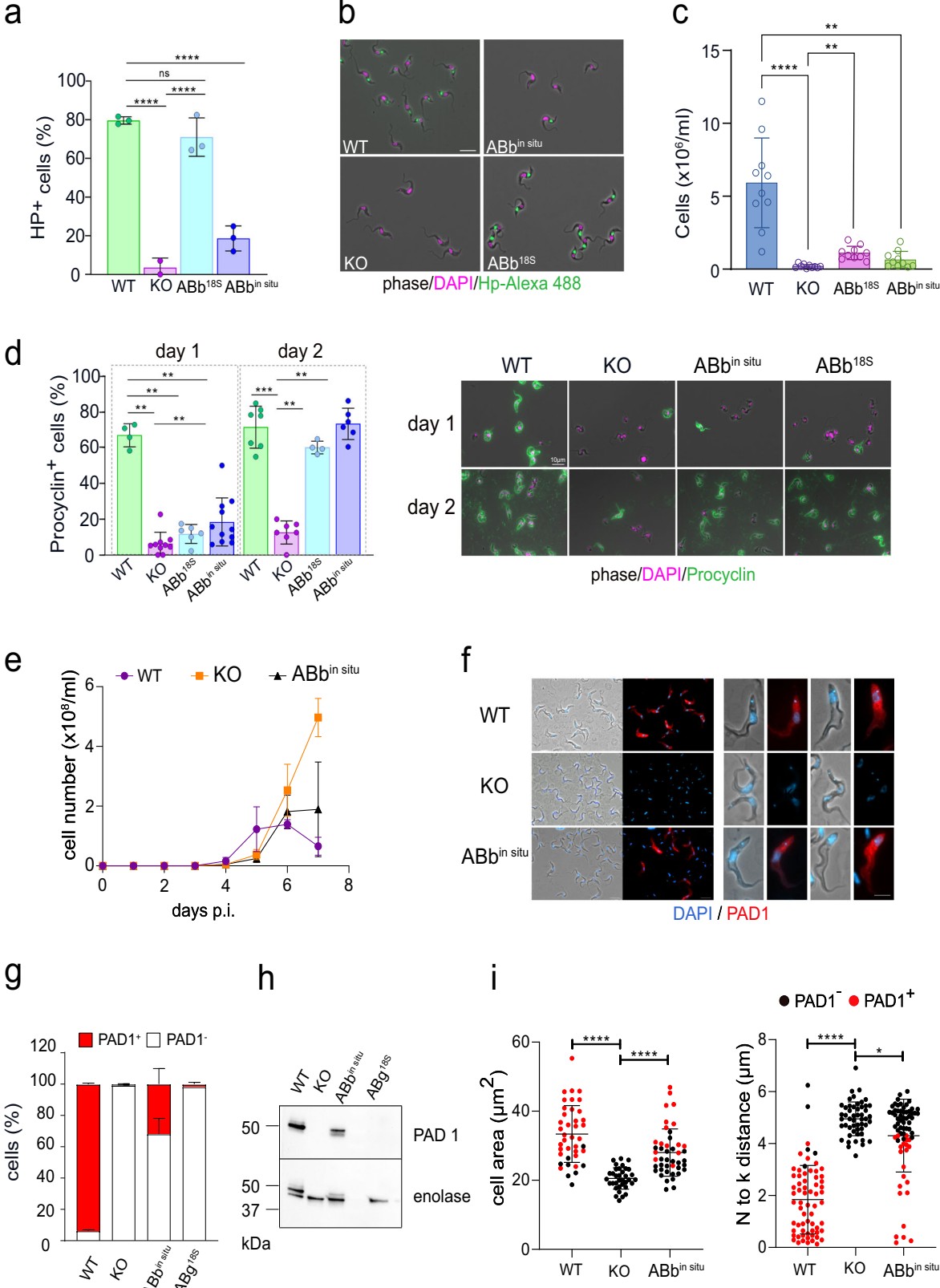

In *T. brucei s. l.* HpHbR has undergone a remarkable set of adaptations, likely triggered by coevolution with their distinct hosts. HpHbR of *T. b. gambiense* acquired specific mutations that decreased the affinity towards its ligand[15,16,18]. Its lower affinity for TLF compared to HpHb led to the proposal that the cells maintain the HpHb uptake while resisting lysis[17]. The closely related *Trypanosoma congolense* and *T. vivax* use HpHbR to obtain Hb rather than HpHb from the digested blood[47]. Moreover, while in *T. brucei s. l.* HpHbR is confined to the flagellar pocket, where it was shown to be downregulated during the stumpy formation[34], it is present on the cell surface of the *T. congolense* epimastigotes[48] at a density ~1000-fold higher than in the BS *T. b. brucei*[47].

**Fig. 4 | Slender-to-stumpy form differentiation is disrupted in HpHbR knock-out cells. a** The uptake of fluorescently-labeled haptoglobin (Hp) was monitored in WT *T. b. brucei* (*n* = 3), HpHbR knock-out (HpHbR KO; *n* = 2), as well as in add-back cells, in which HpHbR was expressed either from the 18S rRNA locus (ABb[18S]; *n* = 3), or the in situ locus (ABb[in situ]; *n* = 3). All cells were harvested 4 days post-infection and mixed and incubated for 2 h with Hp-A488. Dunnet's multiple comparison test, ****p < 0.0001. **b** Fluorescence microscopy of cell lines as in (**a**), with labeled Hp (green) and DNA visualized by DAPI (magenta). Note the absence of Hp uptake in HpHbR KO (scale bar, 10 µM). **c** In vitro differentiation of cell lines as in (**a**). Cells were placed in a DTM medium with 3 mM sodium citrate/*cis*-aconitate at 27 °C and procyclic cells were counted on day 3; (*n* = 10). Dunn's multiple comparison test **p < 0.01; ****p < 0.0001. **d** Examination for procyclin expression on day1 and day 2 Left panel: Percentage of procyclin positive cells (procyclin+) established by fluorescence microscopy; (*n* = 3). Mann–Whitney test **p < 0.01; ***p < 0.001. Right panel: Representative pictures for each cell line stained with procyclin antibody (green) and DAPI (magenta). **e** In vivo infectivity of WT *T. b. brucei*, HpHbR KO, and the ABb[in situ] was evaluated by infecting mice with 1 × 10⁴ cells. The parasitemia was counted daily till day 7 when the experiment was terminated. Representative growth curves (*n* = 3; animals per group) out of three biological repeats are shown. Parasites from infections were harvested on day 7 post-infection and separated from blood using the DEAE column. **f** Indirect immunofluorescence with PAD1 antibody (red), Left panel: Low magnification of the purified cells (scale bar, 10 µM); Right panel: Individual cells at high magnification (scale bar, 5 µM), with discernible DAPI-stained nucleus (N) and kinetoplast (k). **g** Quantification of PAD1-positive and PAD1-negative cells (ex vivo) in WT *T. b. brucei*, HpHbR KO, the ABb[in situ,] and ABg[18S]. In the columns, the stumpy form is shown in red. *n* = 3 biologically independent repeats (at least 200 cells counted per group). **h** Western blot analysis with PAD1 antibody of cell lines described in (**g**). Enolase antibody was used as a loading control. **i** Morphological characterization of cell lines, containing the PAD1-positive ST forms (red dots) along with the PAD1-negative (SL) cells (black dots). Left panel: cell area WT *T. b. brucei* (*n* = 40 cells), HpHbR KO (*n* = 35 cells), and the ABb[in situ] (*n* = 44 cells); Right panel: distance between the nucleus (N) and the kinetoplast (k) WT *T. b. brucei* (*n* = 66 cells), HpHbR KO (*n* = 51 cells), and the ABb[in situ] (*n* = 68 cells). Results were analyzed using the Mann–Whitney test (*p < 0.05; ***p < 0.001, ****p < 0.0001). Error bars indicate ±SD. Source data are provided as a Source Data file.

Following confirmation of the lack of heme *b* detected in the BS *T. b. brucei* devoid of HpHbR[8], here we have shown that BS *T. b. gambiense* also contains minimal if any heme. We imply that the BS trypanosomes can tolerate a very limited amount of heme, suggesting a lack of essentiality of this ancient cofactor in this life cycle stage. As we reported earlier, heme is also dispensable for the plant trypanosomatid parasite *Phytomonas serpens*, which when grown without heme incorporates lanosterol into its membranes instead of ergosterol, overcoming the need for hemoprotein CYP51[49]. In this study, we focused on the hemoprotein CYP51 in *T. b. brucei*, where we have shown that CYP51 is not essential for the BS parasites and represents an acceptor of the HpHbR-imported heme. We have further evaluated the HpHbR-dependent CYP51 inhibition in *T. b. rhodesiense* by modulating its access to the receptor's cognate ligand and proved a positive correlation between an operational HpHb uptake and the cytokinetic CYP51 activity. Moreover, to have a readout system for potent heme-dependent activity, we have overexpressed human catalase[50] in the WT and HpHbR KO *T. b. brucei*, as well as in the WT *T. b. gambiense*. The data conclusively showed that in the BS *T. b. brucei* HpHbR internalizes heme, which is subsequently incorporated into the cytosolic hemoprotein.

Under physiological conditions, human blood is low on free Hb. Lysed erythrocytes release an excess of Hb that binds to Hp with a picomolar affinity and forms a complex internalized by macrophages[51]. Trypanosomes imitate this process via their HpHbR, which competes for the ligand[52]. Impaired erythrocytes and their clearance eventually cause anemia, representing the primary pathological hallmark of animal trypanosomiasis[53]. Intriguingly, reports of severe infection-associated anemia are missing for human *T. b. gambiense* infections, which can be even asymptomatic[54,55]. The reasons behind the clearance of *T. b. gambiense* from the bloodstream are not entirely clear, but the parasites were reported to invade other tissues even more efficiently than *T. b. brucei*[55,56].

The erythrocytes from mice infected with trypanosomes exhibit an enhanced osmotic fragility and altered fatty acid membrane composition[57] caused by host immune response and parasite-derived factors[58]. It was also proposed that during the acute phase of mouse infection, *T. b. brucei* releases extracellular vesicles that fuse with erythrocytes and consequently increase their fragility and clearance[59]. Thus, we propose that the excess of free hemoglobin released by trypanosome infection possibly modulates heme uptake and, subsequently, parasitic waves.

One of the main processes associated with parasitemia control in the mammalian host is the transition from the dividing BS-SL to the quiescent BS-ST[60]. Several proteins and effector molecules involved in this complex process have been described[38,61,62]. In the absence of HpHbR, the key BS-SL to BS-ST transition is disrupted, suggesting that heme uptake may be an additional player in this life cycle progression. Our observation further backs this assumption since *T. b. gambiense* has a poor capability to generate typical BS-ST. Importantly, a mere replacement of the *T. b. gambiense* HpHbR by its functional *T. b. brucei* variant restores the capacity to progress into this life cycle stage. The accelerated pathogenicity for mice of *T. b. brucei* devoid of HpHbR is likely caused by the fast division of BS-SL and the failure to develop into the non-dividing BS-ST, which would otherwise lead to protracted progress of the infection.

HpHbR is known to be a gateway for the internalization of TLF, which results in the lysis of *T. b. brucei* by human serum[52]. Here we show that in *T. b. gambiense*, this receptor also effectively prevents the import of heme with consequences for the heme-requiring cellular processes. While this would be lethal for a typical aerobic eukaryote that depends on external heme, trypanosomatids such as the abovementioned *Phytomonas* have a unique capacity to tolerate its complete deficiency[46].

Changes in *T. b. gambiense* HpHbR seem to have far-reaching consequences. We suggest that heme dispensability may be associated with lower pathogenicity and possibly result in the chronic form of the disease. However, unless experimentally tested, this will remain speculation backed by the lower pathogenicity of the HpHbR KO *T. b. brucei* for both the Hp-carrying and Hp-lacking mice[8]. Significantly, trypanosomes with defective HpHbR struggle to progress into BS-ST, which was considered the only stage of *T. brucei s. l.* capable of transmission to tsetse flies[62]. However, this postulation has been challenged recently, showing that BS-SL and BS-ST *T. b. brucei* are equally competent in infecting tsetse flies, at least via artificial membrane feeding[63]. While BS-SL is likely capable of a short-cut transformation into PS, BS-ST seems to be better pre-adapted for survival in the hostile environment of the tsetse fly midgut. It is worth noting that the addition of glutathione and other antioxidants to the infective bloodmeal may mitigate the effect of oxidative bursts and result in an increased establishment of PS in the insect midgut[36]. Notably, the closely related zoonotic *T. congolense* infects tsetse flies without producing BS-ST[64], although these parasites generate cell-cycle arrested forms akin to stumpy forms of *T. brucei*, albeit without significant morphological transformation[65].

The importance or even mere existence of BS-ST for the transmissibility of *T. b. gambiense* remains controversial. Few studies proposed that BS-SL can differentiate directly or via the stumpy form into PS, directly in the tsetse midgut[66,67]. Moreover, high variability in the proportion of BS-SL to BS-ST was described in different field strains of *T. b. gambiense*[68,69], which supports our observation of variable stumpy marker expressions among *T. b. gambiense* isolates. Additionally, the proportion of BS-ST in *T. b. gambiense* did not correlate with the

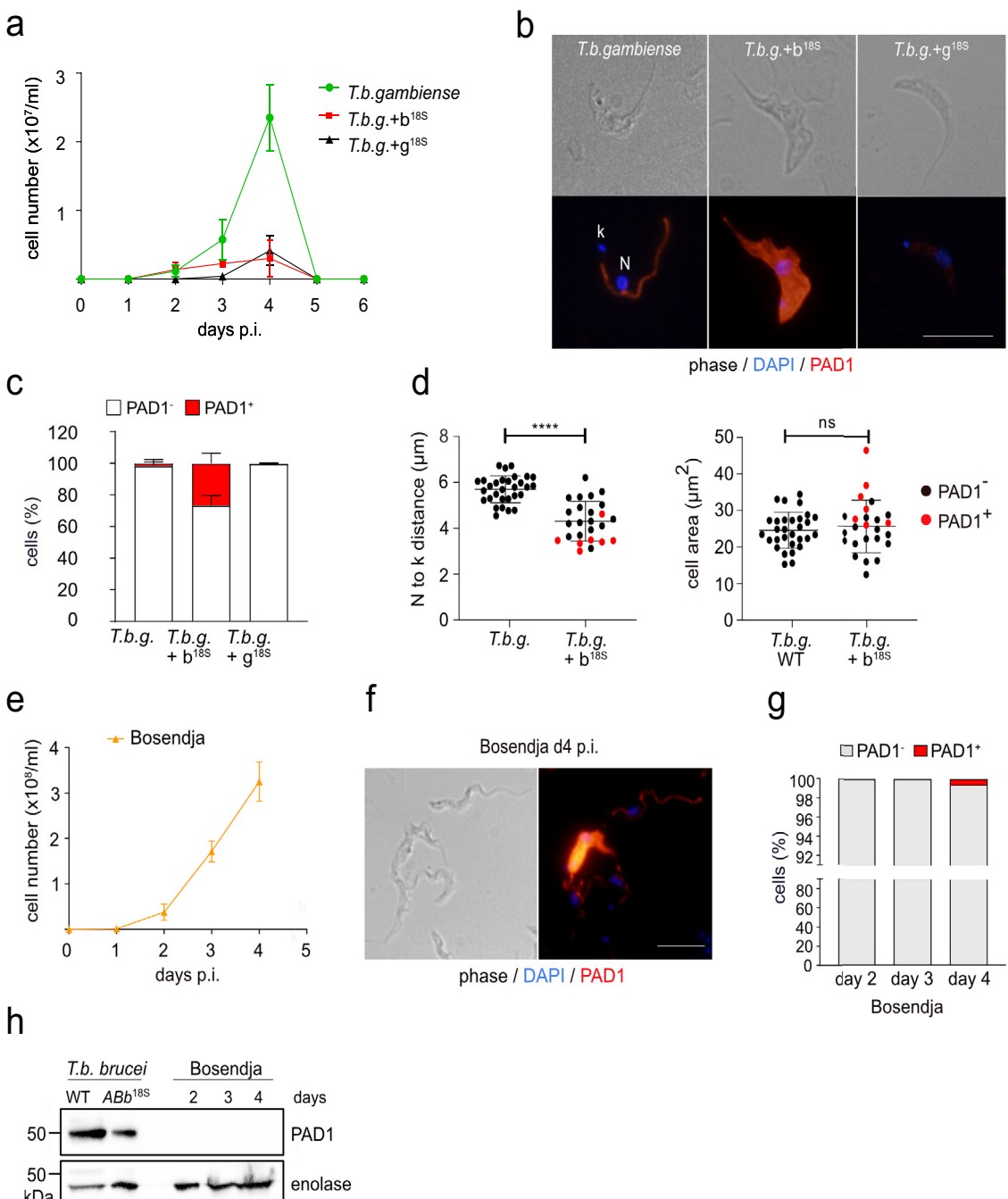

**Fig. 5 | Restoration of stumpy formation in *T. b. gambiense*. a** In vivo infectivity of *T. b. gambiense* LiTat 1.3 (*T. b. gambiense*; green line) and *T. b. gambiense* expressing *T. b. b.* HpHbR (*T. b. g.* + b[18S]; red line) and *T.b. g.* HpHbR (*T. b. g.* + g[18S]; black line) from the 18S rRNA locus was evaluated by infecting mice with $3 \times 10^6$ cells ($n = 3$). The parasitemia was counted daily till day 6 when the experiment was terminated. **b** Indirect immunofluorescence with PAD1 antibody (red), specific for BS-ST in cell lines described in (**a**). DNA in the nucleus (N) and kinetoplast (k) was stained with DAPI (blue). Scale bar, 5 μm. **c** Quantification of PAD1-positive and PAD1-negative cells (ex vivo, day 4 p.i.) in cell lines described in (**a**) ($n = 3$). In the columns, the stumpy form is shown in red. **d** Morphological characterization of *T. b. gambiense* and *T. b. g.* + b[18S] cell lines containing the PAD1-negative cells (black dots) or PAD1-positive cells (red dots). Left panel: the distance between the nucleus (N) and the kinetoplast (k). *T. b. gambiense* ($n = 30$ cells), *T. b. g.* + b[18S] ($n = 25$ cells); Right panel: cell area. *T. b. gambiense* ($n = 30$ cells), *T. b. g.* + b[18S] ($n = 25$ cells). Results were analyzed using the Mann–Whitney test (ns, $p < 0.05$; ****, $p < 0.0001$). **e** In vivo infectivity of *T. b. gambiense* Bosendja strain. The parasitemia was counted daily till day 4, when the experiment was terminated ($n = 3$). **f** Immunofluorescence using PAD1 antibody, revealing a fraction of the PAD1-positive cells. DNA in the nucleus (N) and kinetoplast (k) was stained with DAPI (blue). Scale bar, 5 μm. **g** Quantification of PAD1-positive and PAD1-negative cells in the *T. b. gambiense* Bosendja strain ex vivo days 2-4 p.i. **h** Western blot analysis of WT *T. b. brucei* and ABb[in situ], as well as *T. b. gambiense* Bosendja strain with PAD1 antibody. Enolase antibody was used as a loading control. Error bars indicate ±SD. Source data are provided as a Source Data file.

transmissibility to tsetse flies[69]. The effectivity of transmission is low even under controlled experimental conditions and makes the topic even more elusive[68,70].

Even though *T. b. gambiense* seems to be less prone than *T. b. brucei* to produce stumpy forms in the bloodstream, this difference

may not occur in other tissues. There is convincing evidence that the skin-dwelling stumpy forms are transmissible[55]. However, the data is available primarily for *T. b. brucei*, known to have a relatively high proportion of stumpy forms in the skin. Interestingly, *T. b. gambiense* showed even higher skin parasitemia than *T. b. brucei*, with cells

morphologically resembling BS-LS[56]. Indeed, *T. b. gambiense* was documented in human skin, yet the identity of the stages was not examined[71]. In this context, we point out that the *T. b. gambiense* stumpy forms we analyzed differ from the prototypical stumpy forms of *T. b. brucei* by having an unaltered cell volume, potentially making them more elusive during the examination of infected hosts.

In summary, we provided the first direct evidence for heme-based metabolism in bloodstream trypanosomes and their independence on this cofactor. We newly documented the unique depletion of heme in *T. b. gambiense* cells due to the attenuated haptoglobin-hemoglobin receptor. The loss of functional heme receptor in some trypanosomes results in the reduced occurrence of transmission-competent BS-ST cells from their life cycle and further affects the resulting parasitemia in the vertebrate host.

## Methods

### Cell growth and differentiation
Bloodstream *T. b. brucei* 90-13 (328-114 single marker), *T. b. rhodesiense* EtTat1.2R and *T. b. gambiense* LiTat 1.3 were routinely cultivated at 37 °C in HMI-9 medium (Thermo Fisher), supplemented with 20% heat-inactivated fetal bovine serum. Bloodstream *T. b. brucei* 427 was grown in the same medium with 10% fetal bovine serum. Field isolates of *T. b. gambiense* (Bosendja; AnTAR 6; MHOM/ZR/KIN001 and PA; AnTAR 6; MHOM/ZR/KIN001, both isolated from men in 1972 in the Equateur Province, Democratic Republic of Congo) were kept by passaging in mice. Cell densities were measured using the Z2 Coulter counter (Beckman Coulter) or by hemocytometer.

### Transgenic cell lines
Procyclic *T. b. brucei* 29-13 CYP51 RNAi and HpHbR KO (328-114) cell lines were generated by us previously[8,72]. *T. b. brucei* CYP51-KO (328-114) and 90-13 HpHbR-KO cell lines were generated by successive deletion of alleles with pPuro-KO or pHygro-KO, and pPhleo-KO constructs[18]. The pTSArib *Tb*HpHbR and *Tg*HpHbR constructs used to complement the 90-13 HpHbR-KO cells were obtained by subcloning *T. b. brucei* HpHbR and *T. b. gambiense* HpHbR open reading frame into pTSARib Ble (blasticidin resistance) plasmid[30], resulting in the addback cell lines ABb[18S] and ABg[18S], respectively. The same constructs were used to generate *T. b. gambiense* (LiTat 1.3 strain) overexpressing *Tb*HpHbR (18; *T. b. gambiense* + b[18S]) or *Tg*HpHbR (*T. b. gambiense* + g[18S]) from the 18S rRNA locus. The construct designed to endogenously modulate the level of HpHbR was generated from the pET-in situ construct[73] after replacing hygromycin with phleomycin (Ble) resistance. PCR-amplified DNA fragments were assembled and cloned into pET-in situ Ble plasmid by recombination (InFusion, Clontech) and the resulting cell line was named ABb[in situ]. The constructs are depicted in Supplementary Figs. 2A, E, 3A. Trypanosomes were transfected with linearized plasmid DNA (10 μg) or with gel-purified PCR products (5–10 μg) using a similar protocol published elsewhere[74]. Briefly, a total of $1 \times 10^7$ BS or $1 \times 10^8$ PS cells was harvested by centrifugation at $1000 \times g$ at 4 °C for 10 min and washed 1 time with PBS. Cells were resuspended in 100 μl of AMAXA Human T-cell solution and electroporated with AMAXA Nucleofactor apparatus. Predefined program X-001 and X-014 was used for BS and PS cells, respectively. Selection markers were applied 6 h post-transfection, and clones were generated by limited dilution.

### Real-time PCR, western blot analysis, and luciferase activity measurement
Total RNA was extracted using Trizol reagent, and the cDNA was synthesized using the PrimeScript™ RT Reagent Kit with gDNA Eraser (Takara) as recommended by the manufacturer with an oligo dT used instead of random primers. The qRT-PCR was performed with the cDNA using the SYBR-green stain (Takara). Primers used for qPCR are listed in the Supplementary Table 1 in the Supplementary information file.

The C1 primers were used as endogenous control[75].

In order to detect protein expression in BS, lysates from $5 \times 10^6$ cells were separated on a 12% SDS-polyacrylamide gel, transferred to a PVDF membrane and probed with the monoclonal anti-V5 and anti-α tubulin antibodies (Sigma) at 1:2000 and 1:5000 dilutions, respectively. Alternatively, the samples were prepared the same way (except the lysates were not boiled but heated to 37 °C) and probed with anti-PAD1 and anti-enolase antibodies at 1:1000 and 1: 10,000 dilutions, respectively. The images were detected with Biorad Image Lab software.

For full uncropped gels and blots see the Source Data file and Supplementary Information. $2 \times 10^6$ cells were centrifuged, lysed, and labeled by the Luciferase Assay kit as recommended by the manufacturer (Promega). The read-out was performed for 10 s with the Luminoskan TL Plus instrument (Labsystems).

### Overexpression and activity assay for catalase
The construct for expression of human catalase (hCAT, accession number: NP_001743.1; https://www.ncbi.nlm.nih.gov/protein/NP_001743.1) was modified from the construct generated by us previously[50] to be constitutively expressed. The obtained constructs were linearized by *Not*I, electroporated into BS *T. b. brucei* and *T. b. gambiense* and selected using 1 μg/ml hygromycin (Thermo Fisher).

The activity of catalase was monitored by respirometry using Oxygraph-2K (Oroboros) as a tool to measure the $O_2$ concentration produced after the addition of $H_2O_2$ and analyzed using the Oroboros DatLab Software. *T. b. brucei* 90–13 *and T. b. gambiense* LiTat 1.3 were used to establish the background of the respirometry experiment. Briefly, $1 \times 10^6$ of BS cells resuspended in 2 ml of HMI-9 medium were treated with 20 μl of 882 mM (3%) $H_2O_2$. Alternatively, $5 \times 10^6$ parasites were resuspended in 10 μl phosphate buffered saline (PBS) and placed on a microscopic slide. The same volume of 3% $H_2O_2$ was added to the cells, mixed, and the formation of $O_2$ visible as macroscopic bubbles were monitored as a read-out for the catalase activity. All measurements and statistics were calculated from three independent biological replicates.

### Indirect immunofluorescence assay
For immunofluorescence analysis, $1 \times 10^6$ to $1 \times 10^7$ cells were fixed with 4% paraformaldehyde and settled on microscopic slides. After 10 min incubation at room temperature, they were washed with PBS and permeabilized with 100% ice-cold methanol for 20 min. Cells were incubated with 5% fat-free milk in PBS-Tween (0.05%) for 1 h, followed by incubation with primary anti-V5, anti-PAD1 or anti-CYP51 at 1:1000 dilution and secondary Alexa Fluor-488 anti-mouse and Alexa Fluor-488 or Alexa Fluor-555 anti-rabbit IgG antibodies (Thermo Fisher) at 1:1000 dilution for 1 h at room temperature. After the last washing step, cells were stained with DAPI, mounted with an anti-fade reagent (Thermo Fisher) and visualized using a fluorescent microscope Zeiss Axioplan 2 (Carl Zeiss) and documented with Olympus cellSens Standard Imaging Software.

### High-performance liquid chromatography
A total of $5 \times 10^8$ BS cells was harvested by centrifugation at $1000 \times g$ at 4 °C for 10 min and washed 3 times with PBS on ice. Cells were resuspended in 60 μl $H_2O$, extracted with 400 μl acetone/0.2% HCl, and the supernatant was collected after centrifugation at $1000 \times g$ at 4 °C for 5 min. The pellet was resuspended in 200 μl acetone/0.2% HCl and centrifuged as described above. Both supernatants were combined, and 150 μl of each sample was immediately injected into a high-performance liquid chromatography system (Infinity 1200, Agilent Technologies) and separated using a reverse-phase column (4 μm particle size,

3.9 × 75 mm) (Waters) with 0.1% trifluoroacetic acid and acetonitrile/0.1% trifluoroacetic acid as solvents A and B, respectively. Heme *b* was eluted with a linear gradient of solvent B (30–100% in 12 min) followed by 100% of B at a flow rate of 0.8 ml/min at 40 °C. Heme was detected by diode array detector (Infinity 1200, Agilent Technologies) and identified by retention time and absorbance spectra according to commercially available standard (Sigma-Aldrich).

## Fly and mouse infections

Teneral tsetse flies (*Glossina morsitans morsitans*) were fed 24–48 h after emergence with *T. b. brucei* Antat 1.1 WT and HpHbR-3'PAD1 strains-infected blood meal, either or not supplemented with 10 mM reduced L-glutathione to increase the establishment of infection. Parasites were harvested from the blood of cyclophosphamide-immunosuppressed mice (Endoxan) at 6–7 days post-infection and mixed with defibrinated horse blood (E&O Laboratories). Flies were further fed every 2–3 days on uninfected defibrinated horse blood. Then, flies were dissected on day 10 after the first blood meal to assess the presence of parasites in the midgut (i.e., establishment of a PS midgut infection).

For each biological repeat, five female BALB/c mice(four to six-week-old purchased from AnLab, Czech Republic or Janvier Labs, Belgium) were intraperitoneally injected with $1 \times 10^4$ BS cells (*T. b. brucei* strains) or $3 \times 10^6$ (*T. b. gambiense* strains). The infection was followed daily by diluting tail snip blood in TrypFix buffer (3.7% formaldehyde, 1×SSC buffer) and manual counting of trypanosomes in a Neubauer hemocytometer. Mice were euthanized for the collection of parasites, which were separated from the erythrocytes on a diethylaminoethyl (DEAE) cellulose column using a standard protocol. Purified trypanosomes were washed once with PBS and subsequently used for downstream experiments.

The cell numbers in the study were documented using Excel and GraphPad Prism 8 Software.

## Hp-488 preparation and labeling

Hp was conjugated with Alexa 488 using the Dylight amine-reactive kit (Thermo Fischer) as recommended by the manufacturer. The blood from mice (5 female BALB/c mice, 4–6 weeks old) was collected at different times of infection from a tail puncture with a capillary containing heparin. Following their purification, trypanosomes were subsequently incubated at 37 °C for 15 min in HMI-9 medium containing the lysosomal protease inhibitor FMK-024. Next, they were incubated with 20 µg/ml (f. c.) Alexa 488-labeled Hp for 2 h, subsequently fixed with 4% paraformaldehyde for 10 min, stained with DAPI and analyzed with a Zeiss Axioplan 2 epifluorescence microscope equipped with a Zeiss AxioCam HRm digital camera (Carl Zeiss). The resulting images were analyzed using Adobe Photoshop software and Fiji.

## Statistical analysis

Statistical analyses were performed using GraphPad Prism 8.0 (GraphPad Software Inc.). Data are presented as mean± S.D.

## Reporting summary

Further information on research design is available in the Nature Portfolio Reporting Summary linked to this article.

## Data availability

All data generated in this study are provided in the Supplementary Information and Source Data files.

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

## Acknowledgements
We thank Eva Kriegová and Zuzana Vavrušková (Institute of Parasitology) for their help with animal experiments. We thank Nick Van Reet (Institute of Tropical Medicine, Antwerp) for providing the field strains of *T. b. gambiense*. This work was supported by the ERC CZ project LL1601 to J.L., ERD Funds project OPVVV 0000759 to J.L., and Czech Grant Agency projects 20-07186S and 21-09283S to J.L.

## Author contributions
E.H., B.V., and J.L. designed the study; E.H., L.L., R.S., and B.V. developed the methodology E.H., L.L., P. Cu., R.S., P. Ch., C.J.M.L., J.V.D.A., and B.V. performed the experiments and data analysis; E.H., L.L., P. Cu., J.V.D.A., B.V., and J.L. wrote the manuscript.

## Ethics statement
In the Czech Republic, the research was approved by the Central Commission for Animal Welfare, Biology Center (protocol No. 28/2016). All experimental procedures complied with the Czech law (Act No. 246/1992). In Belgium, the research was approved by the animal ethics committee of the Institute for Molecular Biology and Medicine and the Institute of Tropical Medicine (tsetse fly infection experiment). All mice were housed in a pathogen-free facility and the experiments were performed in compliance with the relevant laws and institutional guidelines (license LA1500474).

## Competing interests
The authors declare no competing interests.

## Additional information

Eva Horáková ®[1,2,10] ✉, Laurence Lecordier[3,10], Paula Cunha[3,7,10], Roman Sobotka ®[2,4], Piya Changmai[1,8], Catharina J. M. Langedijk[1,9], Jan Van Den Abbeele ®[5], Benoit Vanhollebeke ®[3,6] & Julius Lukeš ®[1,3] ✉

[1]Institute of Parasitology, Biology Centre, Czech Academy of Sciences, České Budějovice, Czech Republic. [2]Faculty of Science, University of South Bohemia, České Budějovice, Czech Republic. [3]Laboratory of Neurovascular Signaling, Department of Molecular Biology, Université libre de Bruxelles (ULB), Gosselies, Belgium. [4]Institute of Microbiology, Czech Academy of Sciences, Třeboň, Czech Republic. [5]Trypanosoma Unit, Institute of Tropical Medicine,

Antwerp, Belgium. ⁶Walloon Excellence in Life Sciences and Biotechnology, Antwerp, Belgium. ⁷Present address: Escola Paulista de Medicina, Universidade Federal de São Paulo, São Paulo, Brazil. ⁸Present address: Faculty of Science, University of Ostrava, Ostrava, Czech Republic. ⁹Present address: Cancer Center Amsterdam, Amsterdam UMC, VU, University Medical Center, Amsterdam, The Netherlands. ¹⁰These authors contributed equally: Eva Horáková, Laurence Lecordier, Paula Cunha. ✉e-mail: horakova@paru.cas.cz; jula@paru.cas.cz

