## [Peer review file · Nature Communications]

REVIEWER COMMENTS

Reviewer #1 (Remarks to the Author):

Horakova et al present a comprehensive study demonstrating the haemoglobin uptake in African trypanosomes is mediated almost exclusively by HpHbR. Moreover, the authors also showed that this process is required for a successful developmental progression into stumpy forms, indicating that resistance to human serum is involved in a trade-off whereby HpHbR evolved to have pleiotropic effects in the acquisition of heme proteins and aerobic metabolism, as well as differentiation. The paper is written very nicely. It was a pleasure to read. The data was presented in a logical manner. Below are some comments for consideration:

general comments:

One of the central points of this paper is that *T. b. gambiense* have reduced ability to turn stumpy. The authors should include references demonstrating this reduced capacity for cyclical transmission in comparison to other trypanosome species to back up these claims. *T.b. gambiense* strains vary in terms of their ability to differentiate. I suggest the authors include a paragraph on this and refer to Janelle et al (2009).

The ELIANE strain also used in this paper has been passaged for many years and has essentially become monomorphic, in a similar fashion to 427 for *T. brucei*. So I was pleased to see that a second strain was used to support their findings. However, I would like to know how many passages the Bosendja field strain had undergone? How close to the field is it? Both *T.b. gambiense* strains appear to cause very high parasitaemias and the experiments had to be terminated early. However, most *T. b. gambiense* strains cause extremely low parasitaemias in the mouse model and in humans. I would urge the authors to be cautious about extrapolating their finding to the whole sub-species.

I would also like to raise the point that the authors are only considering *T.b. gambiense* in an animal model (i.e. in the absence of human serum). All their experiments seem valid if *T.b. gambiense* only takes up Hb via the HpHbR, which tryps must do in animal serum where Hp-Hb is only source. But in human serum you have the TLF1 and TLF2 complexes. While *T.b. gambiense* can't take up TLF-1 or Hp-Hb via the HpHbR, that still leaves uptake of TLF2 via VSG-IgM. Could the authors speculate on heme uptake via TLF2? Would this provide enough heme to allow hemoprotein activity, stumpification etc in humans?

If this is true it would be interesting to speculate that *T.b. gambiense* can infect animals but that this is an infection deadend/ not a reservoir as the lack of heme prevents them being effective in making stumps to infect tsetse.

I do not necessarily agree with the conclusion that Heme uptake could have such key role in controlling chronicity in *T.b. gambiense* HAT. The finding reported here are certainly encouraging and must be

considered when understanding dynamics of infection, but I believe more work is require clarifying whether *T.b. gambiense* does not develop into stumpy forms in vivo. A more thorough measurement of other stumpy markers must be included, either in the form of qRT-PCR, IFAs or more unbiased approaches such as transcriptomics to determine if this is the case.

I would also like the authors to comment and reference recent studies suggesting long slenders are able to infect tsetse flies and so the differentiation to stumpies is not essential for transmission.

Specific comments

The introduction is written nicely, but I feel the outcomes can be explained in more details. In particular, in line 48, the authors could mention explicitly which these two processes are (aerobic metabolism and differentiation?).

Line 61, the authors could include a description of TLF-2.

Line 66, the authors could include a reference describing the intravascular haemolysis of haemoglobin.

Line 69, is it known whether TbHRG is developmentally regulated? If so, perhaps this can be mentioned in the text to stress that in the mammalian host, the only possible/known receptor involved in Heme uptake is HpHbR.

Line 212, perhaps it would be good to cite a review here to work describing this.

Line 231, please add references describing this phenomenon.

Results:

Figure 1

- Can you stress that this was an in vitro assay using cultured cells?
- Are the levels of hCAT comparable between cell lines? Based on D and E, it seems like hCAT is more abundant in *T. b. gambiense*.

Figure 2

- Although the results CYP51 KO cell line is very clear, I think the information regarding the lack of a complete mRNA knockdown using RNAi should be better documented as it might help with the narrative as well as in future studies attempting this approach.

- In line 156, the authors could include a reference to the figure they refer to when mentioning the CYP51 RNA and KO cells to ensure readability.

- I think figures 2f and 2g could benefit from including graphs similar to those in figure 2d, either as part of the main figure or as a supplementary file. Also, I think that, in addition to the main, the IC50 values reported in figure 2d should also include standard deviations and number of replicates, either in the figure itself or in the figure caption.

Figure 4

- Figure 4b lacks phase images in the left panel

- Perhaps the authors could elaborate on the differences observed between the in situ AB vs the 18S rRNA AB?

- Are the differences observed in figure 4a and c significant? If so, statistical information should be added to figure and/or figure legend.

Figure 5

- Can the authors elaborate on the seemingly contradictory results observed in figure 5a vs. figure 4f? There is clear nuclear repositioning, but the cells are not changing in volume, indicative of long slender forms.

- Can the authors elaborate on the differences in parasitaemia observed between *T. b. gambiense* LiTat1.3 vs. *T. b. gambiense* Bosendja? One could argue that WT Tbg LiTat does indeed undergo stumpy formation, as determined by a reduction in the levels of parasitaemia >4 dpi. Moreover, in both cases, the levels of PAD1+ cells seem consistent, begin the question of whether PAD1 is the best descriptor of stumpy formation in *T. b. gambiense*?

- Given the impact of such observation, I would recommend that the authors expand on the measurement of further canonical stumpy markers, either via qRT-PCR or IFA.

Conclusions:

- More information is required on lack of stumpy formation in *T. b. gambiense* to sustain the claims of the paper, so I would accept this for publication if more information is included (e.g. qPCRs on candidate stumpy markers).

- How do the authors reconcile their findings with previous reports showing stumpy forms in the skin? Similarly, how do the authors reconcile their findings with the existence of endemic foci where transmission still occur?

Line 103, 449, Were the heme assays performed on cultured *T.b. gambiense* in the absence of human serum?

Reviewer #2 (Remarks to the Author):

This manuscript analyzes the capacity for heme uptake in *Trypanosoma brucei* and human infective *Trypanosoma gambiense* and links expression of the HpHbR to this. The authors further demonstrate that the lack of a heme-dependent enzyme renders parasites less sensitive to an inhibitor of that enzyme, which can be used as an assay for heme uptake and functional delivery. Finally, the authors observe that there are changes in the capacity for the development of parasites to the stumpy stage (BS-ST) when HpHbR is deleted, and in *T. b. gambiense* (which has a naturally mutant HpHbR), and suggest the developmental capacity is restored, although incompletely, with reintroduction of a *T. brucei* HpHbR.

The heme uptake assays are clear (although this part is already known I think) and the assays of Cyp51 are consistent with the overall thesis of the study- i.e. that HpHbR allows heme uptake and supply to enzymes such as CYP51. However, for this reviewer there are quite a few experiments required to substantiate the link between heme uptake and the developmental capacity of *T. b. gambiense* and/or heme uptake capacity. The interpretation of the current experiments also needs clarification.

1. The experiments presented in Figure 1 appear clear, although I think the main conclusion is already known.
2. On Figure 2 the effects on cell growth caused by ketoconazole would be better normalized for each cell line to their own growth in the absence of drug (rather than to the wild type cells in the absence of drug).
3. Figure 3a measures luciferase activity, which many not reflect mRNA (or even protein) levels. The result seems OK, but levels of Luc determined by qRT-PCR or western blot might be valuable.
4. Figure 4e shows large variation based on the error bars. With mouse infections this is normal, but it is valuable to show the profile of the infection in each mouse to assess if the higher or lower average parasitaemia is a consequence of only one or two infections being atypical.
5. In the work presented in figure 4 I consider it important to add back the *T. b. gambiense* HpHbR to the *T. brucei* KO cells. This should not restore developmental capacity, unlike TbHpHbR. Otherwise there is the risk that the developmental phenotype caused by TbHpHbR is simply a consequence of heterologous receptor/protein expression unrelated to receptor function. This is relevant because cells experiencing unphysiological conditions can show some progression to BS-ST characteristics. It would also be interesting to see the phenotypes generated by either TbHpHbR or TgHpHbR expressed from the 18s rRNA locus, since the expression from the in situ locus is weak compared to wild type. If it is as strong as in wild type cells, then much more obvious developmental phenotypes should be seen if TbHpHbR

expression is limiting (and little would be seen with Tg HpHbR expression, again confirming the importance of receptor function).

6. Conversely, in Figure 5 I consider it would be important to 'add back' T. b. gambiense HpHbR expression in WT T. b. gambiense from the 18S rRNA locus to match the expression of the T. brucei add-back in that cell line. If the story is correct, the TgHpHbR should not drive any developmental response/PAD1 expression where the TbHpHbR does. This would confirm the growth effect and possible development seen with the expression of the T. brucei protein is due to its receptor function under conditions of equivalent protein expression.

7. Following on from point 6, what is the level of the expression of the add back T. brucei HpHbR in the T. b. gambiense cells? The restoration of developmental competence is quite subtle (notwithstanding the growth effect in vivo- the growth in vitro is not presented but would be of interest) : morphologically BS-ST cells are not clearly generated (by cell volume) and PAD1 expression is very limited. If expression of the T brucei protein were efficient I would have expected a better level of development if this receptor is an important contributor to the developmental response. The cell cycle status of cells expressing the TbHpHbR should be presented in addition to their growth also- since development to BS-ST should show an increase in cell cycle arrested forms (1K1N).

8. If heme levels are important in the process of development, the capacity of T brucei and T gambiense parasites to express PAD1 and develop to BS-ST forms could be explored in vitro using heme deficient or haptoglobin depleted medium, or in media with excess heme. These assays would allow quantitative and titratable effects to be observed to link uptake efficacy and development and are tractable for monitoring BS-SL and BS-ST expression in vitro.

Overall, I am convinced by the data on heme uptake and the importance of the HpHbR in this process- this is a valuable finding (although some of it is known). I remain to be convinced that there is a consistent and causal relationship between HpHbR and developmental capacity in T. b. gambiense. As the authors state (line 369-370), different isolates of T. b gambiense show different levels of slender and stumpy formation in different field isolates, although presumably all have HpHbR mutations to permit human infectivity. Hence, evidence that there is a robust relationship between these two observations (HpHbR function and developmental capacity) is not yet established – the T. b. gambiense lines used in their study are of uncertain passage history in mice; other field isolates may behave differently.

I also found the interpretation in the discussion to be quite muddy. On one hand, evidence is cited that T b. gambiense is transmitted by flies poorly, which would be consistent with a proposed link between HpHbR efficacy and development. On the other hand, an unpublished BioRxv report is cited of equivalent fly infectivity for slender and stumpy forms (albeit using unusually permissive flies) and used to suggest that the developmental capacity is not evolutionary constraint for T b gambiense. These arguments seem contradictory.

Finally, I assume that the heme dependent response is secondary to density dependent stumpy formation by quorum sensing (the dominant mechanism driving transmissibility)? Since heme availability and uptake would likely not be density dependent (nor limiting in vitro), it seems the developmental effects (if validated) would be a consequence of a heme-dependent processes in the BS-

ST pathway downstream of the initial stimulation. This can make sense and would not be surprising given the mitochondrial involvement and metabolic adaptations inherent in BS-ST development. However, without some further experimentation and validation I left uncertain of the importance or mechanism of the developmental aspect of the work, despite interest in the observations reported.

Minor issues

1. Figure 1d would benefit from molecular weight size markers (the relevant bands are labelled but the size would be good to add).

2. Line 125, referring to figure 2A, says that CYP51 is differentially transcribed in procyclic, slender and stumpy forms. Since most regulation is posttranscriptional, it is more likely to have differential mRNA abundance but not be differentially transcribed. This should be amended.

3. On Figure 2f and g the Y axis is broken to stretch out the region between 50-100% and to compress 0-40%. I don't think this is necessary and is potentially misrepresentative of the scale of effects seen.

4. Line 181 refers to the PS to BS differentiation, but the differentiation is actually BS to PS. Please correct.

5. Line 240 'bulky HpHbR expression'. I don't think bulky is quite the right word. There are several other places where the phraseology or English usage would benefit from attention.

Reviewer #3 (Remarks to the Author):

The human infectivity of Tbg is, unlike Tbr, multifactorial. Decreased functionality and decreased expression of the HpHbR is known to be one of the evolutionary modifications that have allowed Tbg to infect humans, through reduced uptake of TLF1. Here, the authors main claims are: (1) That both Tbb HpHbR KO and Tbg cells lack hemoprotein activity and therefore proliferate in the absence of heme. (2) That HpHbR KO prevents differentiation of the parasites into stumpy forms. (3) That Tbg is poorly competent to differentiate into stumpy forms, due to reduced functionality of HpHbR

This is an interesting and timely study that would certainly contribute to the field and raise questions for further study. There are some revisions that are required in order to support the claims presented and the text needs to be clarified to state only what is experimentally shown.

Main claim 1: That both Tbb HpHbR KO and Tbg cells lack hemoprotein activity and therefore proliferate in the absence of heme.

The Tbb HpHbR KO cell line has previously been generated and characterised for its inability to internalise HpHb or free heme, despite little growth effect in vitro (Vanhollebeke et al., 2008). It was also previously stated in the same work, that: “The steady-state heme content of WT cells isolated from mice was 2.3 ng/mg of protein, whereas heme was undetectable in KO cells” and that “heme appeared to be mostly incorporated into hemoproteins” which would together then strongly support that the Tbb HpHbR KO cell line should indeed have an absence of hemoprotein activity and proliferate in the absence of heme. Here, this previous data is confirmed with further experimental support. The measurement of the intracellular heme content of Tbg is novel and of significant interest, given the previously observed mutations and changes in gene expression of the TbgHpHbR. The authors also assess, for the first time, hemoprotein activity between these cell lines via introduction of a human hemoprotein in Tbb WT, Tbb HpHbR KO and Tbg cells and via endogenous CY51 hemoprotein activity in Tbb WT, Tbb HpHbR KO and Tbr cells.

Heme measurements are conducted and quantified, with measurements for TbbHpHbR KO and Tbg close to the limits of detection and with no significant difference detected between them. That being said, the chromatogram from Fig 1a and quantification from Fig 1b, does imply that Tbg has some, albeit very little, internal heme.

The addition of human catalase into Tbb, TbbHpHbR KO and Tbg cells was a good experiment as this allows the cell lines to be compared directly for hemoprotein activity without consideration for other mutations or changes in gene expression that may be distinct between the cell lines. This data demonstrates that the acquisition of heme in Tbg does not support any significant activity of human catalase, although again, the data does imply a very mild activity as the shape of the measurement curve (Fig 1f) does have a small jump at the time of H₂O₂ addition, unlike the TbbHpHbR KO cells.

Given the high level of intracellular heme in the Tbb WT cells and the trace amount in Tbg cells, it could be argued that the amount of heme present in Tbg is enough for some activity of endogenous hemoproteins but is not in excess and therefore unable to fuel the human catalase. The authors themselves describe this as ‘potent heme-dependent activity [through overexpression]’ (line 319). For this reason, it would be particularly valuable for the authors to assess the endogenous hemoproteins of Tbg, as they have done for Tbb WT and TbbHpHbR KO cells in Figure 2. In fact, it is very unclear as to

why the authors have included a ketoconazole sensitivity experiment for Tbr (Figure 2G), when they have not included this for Tbg. This should be added and would considerably strengthen the manuscript.

Also, with Tbr not particularly the focus of this work, it is not clear why the experiment in Fig 2G was conducted on Tbr at all, rather than the Tbb WT and Tbb HpHbR KO cell line.

Throughout, the manuscript text could be clarified to ensure that the authors are clear on the conclusion that they are presenting regarding the TbgHpHbR expression / function based on their data and previously published work. Absence is not the same as a little and poorly capable is not incapable. The data in this manuscript and in published literature fully supports a decreased functionality (i.e. reduced binding capacity for HpHb) and decreased expression of the TbgHpHbR (mRNA and protein). It does not support a complete loss of the TbgHpHbR. Perhaps the authors would like to comment or speculate on the observation that there has not been a complete loss either via expression or a true loss-of-function mutation. Would this imply that there is indeed some residual use of the receptor and/or benefit to maintain it?

Main claim 2: That HpHbR KO prevents differentiation of the parasites into stumpy forms.

HpHbR is downregulated upon transition of BS-SL cells to BS-ST cells. The authors investigate if this due to position on the polycistronic unit, which it is not. This section seems to be addressing a rather different hypothesis to the focus of the rest of the manuscript (mechanisms of gene expression during differentiation rather than evolutionary trade-offs of reduction of heme uptake) and could be moved to supplementary.

The authors also show that the presence of HpHbR in BS-ST forms does not in any way hinder the transition to BS-ST or PS forms. The authors could more clearly indicate what the hypothesis for this experiment was; was it that there may be a selective downregulation of the HpHbR at this stage as opposed to being subject to the general translational repression observed in BS-ST cells (Brecht and Parsons 1998)? Upon expression of the HpHbR in BS-ST forms, the authors state that the cells "...still differentiate into the BS-ST and subsequent PS and retain the ability to infect tsetse flies" (line 208). This language implies that they anticipated a hinderance.

To the contrary, given that they later state that presence of heme is required for differentiation of the parasites into stumpy, did they consider if the extended expression of HpHbR may actually benefit this transition? As shown in Figure 3d, the HpHbR-3'PAD1 cell line, as an average, peaks a full day earlier than the WT and at a moderately lower density which would be consistent with an earlier transition to BS-STs. Although I can see that the error bars on parasite load would likely overlap, the trend on rising and falling parasitaemia seems clear. Is this a reproducible difference between individual mice and/or different experiments? Was this ever studied with 12 hour time points between day 3-6? If the BS-ST

forms or PC forms have a greater reliance on hemoproteins, then perhaps the authors might consider if the extended expression of the HpHbR in this case increases heme availability/hemoprotein activity during this transition.

A pleomorphic HpHbR KO and two add-back cell lines (18S and in situ) are produced; no validation of the add-back cell lines is included and the authors should therefore add this to Supp Fig 2 (genetic validation or assessment of expression levels). These cell lines are assessed functionally through the number of Hp+ cells after exposure to labelled Hp. The WT cells have a value of 80% with the add-backs providing 70 or 20%. Could a low number of positive cells in the in situ add-back also represent a non-clonal culture where not all cells are expressing the gene?

At this stage, the authors have sought to “...test whether heme and/or HpHbR play any role in [BS-ST formation]” (line 217). The data collectively presented in Figure 4 convincingly demonstrates that the TbHpHbR KO cell line does not form BS-STs and that this is due to the absence of the HpHbR. But this does not differentiate between heme and/or HpHbR and it is not clear from the authors discussion what their conclusion for the molecular basis for this is. They state “The fact that in the absence of HpHbR, the key BS-SL to BS-ST transition is disrupted, prompts us to suggest that heme uptake may be an additional player in this life cycle progression” (line 340), and that “...HpHbR facilitates the developmental progression by inducing PAD-1 expression” (line 44). Both of these statements are implicating the HpHbR itself in developmental regulation and/or signalling. However this is distinct from a more likely hypothesis that BS-ST forms simply have at least one required hemoprotein and therefore unlike BS-SL forms, BS-ST forms do require heme. It would be beneficial to this work if the authors did experimentally address this distinction, by expressing the TbHrg in the TbHpHbR KO or assessing hemoprotein activity in BS-ST, for example. This also has implications for claim 3.

Main claim 3: That Tbg is poorly competent to differentiate into stumpy forms, due to reduced functionality of HpHbR.

The finding that Tbg produces little/no BS-ST would be particularly novel and of interest to the field. However, while the authors have clearly uncovered some interesting biology here, the data presented does not satisfactorily support the general claim that Tbg is poorly competent to produce BS-ST. It is very possible that, in evolving the adjust to the modifications of the reduced functionality of the TbgHpHbR, that Tbg BS-ST are not identical to Tbb BS-ST (perhaps reducing reliance on hemoproteins). This would be very interesting and authors have clearly considered this possibility as they state “WT Tbg has a poor capability to generate typical BS-ST” (line 343). The data presented should be published and is of interest, but I do not think it can be summarised as demonstrating that “Tbg is poorly competent to differentiate into stumpy forms”.

Figure 5a-d clearly demonstrates that very few PAD1+ cells/ morphologically BS-ST cells are identified in the Tbg WT and this is moderately recovered with add-back of the TbHpHbR. However, the growth curves from the mouse experiment indicates that the differences in dynamic here may be more complex. The authors have shown that an absence of HpHbR for Tbb results in an outgrowth of the parasites which would ultimately overwhelm the host (Figure 4e). This uncontrolled growth is typical of a 'monomorphic' cell line which does not produce BS-ST. This is not what is observed with the Tbg WT, where the parasite numbers do fall on day 5. Indeed, it is known than Tbg causes a chronic infection in humans and can result in asymptomatic infections, whereas an absence of BS-ST forms in Tbb (as per Fig 4e) is shown to cause uncontrolled growth. This again supports a hypothesis that there is a distinct situation in Tbg, but not simply through a lack of BS-ST forms.

The addition of the TbHpHbR to Tbg very much suppressed parasitaemia which, based on cell number alone, might indicate a dramatic shift towards differentiation to BS-ST, but this is not reflected in the PAD1+ assay, where only around 30% of the cells are positive. A straightforward additional experiment of interest here would be to assess the parasites on day 4 for their cell cycle status to determine if they are or are not undergoing a cell cycle arrest. Further, a consequence of loss of BS-ST formation for the TbbHpHbR KO was an additional reduced capacity to generate PS, which was not assessed for Tbg. Ideally it should be, since the authors do discuss a reduced capacity to infect flies in their discussion (Line 364) and it would support their hypothesis if they saw increased fly infectivity upon TbHpHbR overexpression in Tbg.

The data presented in Figure 5e-h regarding the Tbg Bosendja Field strain was generated to address if this strain "... sustains the ability to produce waves of parasitemia". This is not addressed as the experiment is terminated on day 4. It would have hugely benefited from being continued to day 5. The parasitaemia in the mice could have dropped as for the Tbg WT (Fig 5a) or continued to rise as for the Tbb HpHbR KO (Fig 4e) and so no conclusions can be drawn either way.

My final concern with the statement that Tbg is poorly competent to generate BS-ST is that "... publications describe a high variability in the proportion of the BS-SL to BS-ST cells in different field strains of Tbg" (line 369) and we know that different field isolates have different levels of HpHbR expression as well (Kieft et al., 2010). The data presented should be considered more broadly within the context of the variability of Tbg isolates, which may actually support the authors hypothesis, if the isolates with higher levels of reported BS-ST forms had correlatively higher levels of HpHbR expression.

Text

There are a few statements that are either incorrect or written in such a way as the scientific interpretation is incorrect or ambiguous. These should be modified to ensure accurate reflection of the literature or point being made.

Line 36: "... the physiological function of the receptor remains to be elucidated". Do the authors mean the function is yet to be elucidated in Tbg specifically? Because the function of the HpHbR as a receptor for HpHb in Tb is already known (Vanhollebeke et al., 2008; Stødtkilde et al., 2014).

Line 44: "... HpHbR facilitates the developmental progression by inducing PAD-1 expression..." The authors do not show that HpHbR induces PAD1 expression.

Line 63: "...HpHbR, the only invariant cell surface receptor known to date in kinetoplastid parasites". T. brucei Factor H receptor. (Macleod et al., 2020).

Lines 79-82: "Retaining the HpHbR expression contribute to trypanosomes fitness in their animal reservoir hosts, providing positive selection pressures for the conservation of this receptor". Are the authors referring to Tbg or non-Tbg Tb species? It is not clear if the point being made is that Tbg have retained the HpHbR and this contributes to fitness in animals or that there is advantage to non-gambienese Tb, which Tbg no longer have.

Line 82-85: Reference(s) required.

Line 174-176: Reference or names of 'other genes' and/or data required. By 'involved' do the authors mean differentially regulated (of which there are very many genes) or do they mean playing an active role in the transition?

Line 238-240 and lines 262-264: Can the authors clarify the justification here. What do they mean by bulky? Expression levels have not been shown in this manuscript. Further, the data in Figure 4a indicates that the 18S add back is more similar to the WT and the authors proceed to use the 18S locus for the Tbg add back.

Line 260: The title of this section and Figure 5 are "Restoration of stumpy formation in Tbg". This is not ideal, as it indicates that the authors have already shown Tbg do not form BS-ST. "Overexpression of TbbHpHbR in Tbg facilitates increased BS-ST formation" or similar, would be more suitable.

Line 267: "... on day 6..." should be on day 5.

Line 271: "... did not detect any PAD1-expressing cells". Data in 5c does have some indication of a very small number of PAD1+ cells.

Lines 323-329: The authors discuss the anaemia of animal trypanosomiasis and then comment of the lack of such observations in human Tbg studies. Are the authors implying a cause-and-effect relationship and in which case, which direction? Do Tbg have less use for a HpHbR because there is less free HpHb, which is not necessarily logical because reduced concentration of ligand in blood could select for greater binding capacity in order to compete with host macrophages.

Lines 330-336: Please clarify this hypothesis, particularly line 334-336.

Line 357-359: "... the lower pathogenicity of the HpHbR KO T. b. brucei for both the Hp-carrying and Hp-lacking mice (Vanhollebeke et al., 2008)". This is not what the data in this manuscript supports, which demonstrates increased pathogenicity after day 6 (Fig4e).

Lines 367-369: "To the best of our knowledge, the reporting of putative BS-ST cells in T. b. gambiense was based solely on morphological criteria and may have resulted in a misassignment to a different T.

brucei sub-species that is more prone to BS-ST transition” Are the authors stating they believe that publications 50, 51, 52 and 53 have all incorrectly utilised Tbb instead of Tbg in their studies?

Figures

Figure 2b: mRNA levels of CYP51 knock down should be shown in supplementary data to allow the reader to assess extent of knock-down.

Figure 2d: I question the normalisation of the CYP51 RNAi +dox to the growth of the CYP51 RNAi –dox and the CYP51 KO to the WT in this experiment because these cell lines already have a difference in growth rate and so at 0uM ketoconazole the growth over 48 hours would be reduced. This makes interpretation of the data somewhat problematic. The authors could replot the data with each cell normalised to its own growth rate to determine if this improves clarity. Particularly, it should then be clearer to observe the presence vs absence of the biphasic curve. This is simply a suggestion to try and improve visualisation of this data. Without being able to see what that would look like, I do appreciate that upon doing that it may not improve the figure. In either case, the area of the figure that is of most interest is the range of 1-10uM (given this is the range then used in Figure 2f and 2g, as well as the area required on the WT curves to observe the biphasic nature of the curve). This is not particularly clear in the figure and this should be modified where possible to improve this (data point size reduced, possibly open boxes for the CYP51 KD / KO).

Figure 3d: A supplementary file showing individual mouse infection profiles would be beneficial here.

Figure 3e and 4d: Scale bars not defined.

Figure 4c: The Y-axis is labelled procyclics. Is this accurate in that only ‘procyclic’ cells would be counted?

Figure 4c and 4d: No error bars are provided, although replicate numbers are given.

Figure 4f and 4h: Given that the PAD1+ cells have a greater cell area and shorter N to k distance, the calculation of the mean values in the whole populations and then use of these to measure significant difference seems inappropriate. The data in Figure 4h should be provided before 4f and used for statistical analysis. The data in 4f should then be separated such that there are six sets of data per graph (WT PAD1-, WT PAD1+ and so on). This would exclude the ability to compare data in 4f with statistics, but would clearly demonstrate the point being made.

Definition of error bars is in the methods rather than in each of the figure legends. Multiple cases of the number of replicates not being included in Figure legends (i.e. all of the figure 2 panels, Figure 4e).

REVIEWER COMMENTS

Reviewer #1 (Remarks to the Author):

Horakova et al present a comprehensive study demonstrating the haemoglobin uptake in African trypanosomes is mediated almost exclusively by HpHbR. Moreover, the authors also showed that this process is required for a successful developmental progression into stumpy forms, indicating that resistance to human serum is involved in a trade-off whereby HpHbR evolved to have pleiotropic effects in the acquisition of heme proteins and aerobic metabolism, as well as differentiation. The paper is written very nicely. It was a pleasure to read. The data was presented in a logical manner. Below are some comments for consideration:

general comments:

One of the central points of this paper is that *T. b. gambiense* have reduced ability to turn stumpy. The authors should include references demonstrating this reduced capacity for cyclical transmission in comparison to other trypanosome species to back up these claims. *T. b. gambiense* strains vary in terms of their ability to differentiate. I suggest the authors include a paragraph on this and refer to Janelle et al (2009).

There is a lot of circumstantial evidence for compromised transmissibility of T. b. gambiense by tsetse flies. T. b. gambiense infection rates in field-collected flies are always very low when looking into the published data, even in gambiense-HAT endemic regions. Moreover, even in optimal experimental conditions, it is challenging to obtain mature infections for T. b. gambiense in the tsetse fly (J.v.d.A; unpublished data). As suggested, we have now added a literature overview on this issue (Janelle et al., 2009; Welburn et al., 2016).

The ELIANE strain also used in this paper has been passaged for many years and has essentially become monomorphic, in a similar fashion to 427 for *T. brucei*. So I was pleased to see that a second strain was used to support their findings. However, I would like to know how many passages the Bosendja field strain had undergone? How close to the field is it?

Specific information on this field strain is as follows: Bosendja - AnTAR 6; ZR/KIN001, isolated from a man, Equateur Province, Democratic Republic of Congo, 1972. It was provided to us by Nick Van Reet (Institute of Tropical Medicine, Antwerp; in acknowledgments), who keeps the strain as a stabilate in rodent blood. The parasites were thawed out and injected directly into the Balb/c mice. Unfortunately, the information about how many passages the parasites went through in the animals is not available. On the other hand, they are difficult to be grown in a culture where they can potentially lose their pleiomorphic capacity. Since they were not produced as such, we can consider them as potentially pleiomorphic.

Both *T. b. gambiense* strains appear to cause very high parasitaemias and the experiments had to be terminated early.

However, most *T. b. gambiense* strains cause extremely low parasitaemias in the mouse model and in humans. I would urge the authors to be cautious about extrapolating their finding to the whole sub-species.

With all due respect, we would rather disagree here since it is only the field strain that causes high parasitemia, but not the laboratory strain, which consistently gives very low parasitemia. Nevertheless, we hope that we are now more cautious about the formulations in the text.

I would also like to raise the point that the authors are only considering T.b. gambiense in an animal model (i.e. in the absence of human serum). All their experiments seem valid if T.b. gambiense only takes up Hb via the HpHbR, which tryptophans must do in animal serum where HpHb is only source. But in human serum you have the TLF1 and TLF2 complexes. While T.b. gambiense can't take up TLF-1 or Hp-Hb via the HpHbR, that still leaves uptake of TLF2 via VSG-IgM. Could the authors speculate on heme uptake via TLF2? Would this provide enough heme to allow hemoprotein activity, stumpyfication etc in humans?

These are elegant thoughts indeed; we much appreciate this comment. Unfortunately, we can only speculate here, using mice as a model organism for apparent reasons. TLF2 may work independently of HpHbR (Raper et al., 1996) while it has Hpr and therefore could contain hemoglobin, which may constitute an entry point of heme when T. b. gambiense grows in humans. This hypothetical TLF2-based heme uptake could help explaining why T. b. gambiense non-human and non-primate reservoirs are limited. Intriguingly, T. b. gambiense may prefer human blood for its TLF2.

If this is true it would be interesting to speculate that T.b. gambiense can infect animals but that this is an infection deadend/ not a reservoir as the lack of heme prevents them being effective in making stumpies to infect tsetse.

It may be the case, but we decided to refrain from this speculation because more data shall be collected to support such a substantial claim.

I do not necessarily agree with the conclusion that Heme uptake could have such key role in controlling chronicity in gambiense HAT. The finding reported here are certainly encouraging and must be considered when understanding dynamics of infection, but I believe more work is required clarifying whether T.b. gambiense does not develop into stumpy forms in vivo. A more thorough measurement of other stumpy markers must be included, either in the form of qRT-PCR, IFAs or more unbiased approaches such as transcriptomics to determine if this is the case.

While we believe that the evidence we have provided in the original submission supports the (partial) loss of the capacity to form stumpies, we agree with this reviewer that more evidence is welcome. She/he suggested qRT-PCR, which we now provide for several marker genes for the stumpy form. The analysis is now part of extended Figure 5 (Suppl. Fig. 5).

I would also like the authors to comment and reference recent studies suggesting long slenders are able to infect tsetse flies and so the differentiation to stumpies is not essential for transmission.

Whether the short stumpy form (BS-ST) is essential for the establishment of the infection in the tsetse midgut remains controversial. Even in the older literature, this was already questioned based on experimental observations (see overview in Janelle et al., 2009). Personal experience of one of us (J. v. d. A.) is that giving the first bloodmeal to tsetse with

long slender forms (BS-SL) can result in tsetse midgut infections, although at a much lower rate than an infective bloodmeal with a majority of BS-ST. This experiment was done with in vivo grown trypanosomes isolated from an infected mouse, not from in vitro culture like in Schuster et al. (A modification to the life cycle of the parasite Trypanosoma brucei. bioRxiv717975; doi: <https://doi.org/10.1101/717975> [2020]), which is highly artificial. When we added reduced L-glutathione (GSH) to bloodmeal, the midgut establishment rate raised to very high levels when infected with BS-SL. So, intrinsically it seems that BS-SL can do a short-cut transformation into the midgut procyclics (PS). However, it appears that BS-ST is pre-adapted to better survival in the hostile midgut environment (withstanding possible oxidative burst) when ingested by the tsetse fly. The addition of GSH in the first infective bloodmeal seems to suppress this hostility for BS-SL, resulting in better initial survival and higher PS midgut establishment.

We already cite the above publication (Schuster et al., bioRxiv 2020) in our study, and we have also added several older publications, which question the necessity of BS-ST for transmissibility of T. b. gambiense.

Specific comments

The introduction is written nicely, but I feel the outcomes can be explained in more details. In particular, in line 48, the authors could mention explicitly which these two processes are (aerobic metabolism and differentiation?).

The part of the abstract was reformulated, and the two processes are now explicitly named as suggested.

Line 61, the authors could include a description of TLF-2.

TLF2 is now described along with appropriate references.

Line 66, the authors could include a reference describing the intravascular haemolysis of haemoglobin.

As suggested, we have now added a relevant publication (Schaer et al., 2013) describing the intravascular haemolysis of haemoglobin.

Line 69, is it known whether TbHRG is developmentally regulated? If so, perhaps this can be mentioned in the text to stress that in the mammalian host, the only possible/known receptor involved in Heme uptake is HpHbR.

Yes, indeed. The TbHrg transcript is developmentally regulated during the life cycle, which is now described in the text.

Line 212, perhaps it would be good to cite a review here to work describing this.

A proper reference now accompanies the relevant sentence.

Line 231, please add references describing this phenomenon.

The following references have been added as suggested: Brun et al., 1981; Czichos et al., 1986; Ziegelbauer et al., 1990.

Results:

Figure 1

- Can you stress that this was an in vitro assay using cultured cells?

Yes, it is an in vitro assay, but trypanosomes were isolated from mice, therefore ex vivo. To make this clear, the text has been extended as follows: "Moreover, ex vivo cells collected on day 4 from infected mice were examined morphologically and the functionality of HpHbR was verified by the uptake of the fluorescently-labeled HpHb complexes (Figs. 3C, D)".

- Are the levels of hCAT comparable between cell lines? Based on D and E, it seems like hCAT is more abundant in *T. b. gambiense*.

Yes, the expression of HpHbR differs to some extent in various cell lines. In any case, the higher expression in *T. b. gambiense* supports the idea that the enzyme is made but remains inactive due to the missing heme cofactor.

Figure 2

- Although the results CYP51 KO cell line is very clear, I think the information regarding the lack of a complete mRNA knockdown using RNAi should be better documented as it might help with the narrative as well as in future studies attempting this approach.

- In line 156, the authors could include a reference to the figure they refer to when mentioning the CYP51 RNAi and KO cells to ensure readability.

While revising the manuscript, we found the data generated on CYP51 RNAi redundant (due to the comparable data on CYP51 KO). Therefore, we decided to remove the CYP51 RNAi data, which helped the clarity and readability of the text.

Looks like figures 2f and 2g would benefit from including graphs similar to those in figure 2d, either as part of the main figure or as a supplementary file. Also, we believe that in addition the IC50 values reported in figure 2d should also include standard deviations and number of replicates, either in the figure itself or in the figure caption.

Figure 2 has been simplified, and the graphs have been modified according to the referee's suggestions. In Fig. 2D (now 2C) standard deviation for IC50 has been added.

Figure 4

- Figure 4b lacks phase images in the left panel

The left panel of Figure 4b has been removed since it is difficult to judge the morphology under such magnification. Moreover, the right panel provides a representative figure.

- Perhaps the authors could elaborate on the differences observed between the in situ AB vs the 18S rRNA AB?

Per request, we have now added the expression for 18S rRNA AB and the in situ AB (Suppl. Fig. 3C). As anticipated, the expression profile shows that 18S rRNA AB has a much higher (50x) expression of HpHbR than in situ AB cells.

- Are the differences observed in figure 4a and c significant? If so, statistical information should be added to figure and/or figure legend.

Statistical information has now been added to Figs. 4A and 4C.

Figure 5

- Can the authors elaborate on the seemingly contradictory results observed in figure 5a vs. figure 4f? There is clear nuclear repositioning, but the cells are not changing in volume, indicative of long slender forms.

Indeed, this is not easy to explain. Our view is that these are not identical to the prototypical stumpies of T. b. brucei. We believe that it is reasonable to assume that these T. b. gambiense forms generated under these experimental conditions have distinct features from stumpies, including the unaltered cell volume.

- Can the authors elaborate on the differences in parasitaemia observed between T. b. gambiense LiTat1.3 vs. T. b. gambiense Bosendja? One could argue that WT Tbg LiTat does indeed undergo stumpy formation, as determined by a reduction in the levels of parasitaemia >4 dpi. Moreover, in both cases, the levels of PAD1+ cells seem consistent, begging the question of whether PAD1 is the best descriptor of stumpy formation in T. b. gambiense?

We are aware of these differences between the two strains. They may be caused by subtle differences in their pathogenicity or because LiTat1.3 was routinely kept in the medium while Bosendja was passaged thru mice. Moreover, we can also not exclude the possibility that the clearance of the bloodstream parasitemia is not always linked to the stumpy formation, and other processes such as myeloid cell-based immunity may play a role.

- Given the impact of such observation, I would recommend that the authors expand on the measurement of further canonical stumpy markers, either via qRT-PCR or IFA.

We agree and we have performed further experimentation and added new data to support our claims regarding the stumpies.

Conclusions:

- More information is required on lack of stumpy formation in T. b. gambiense to sustain the claims of the paper, so I would accept this for publication if more information is included (e.g. qPCRs on candidate stumpy markers).

In response to this request, we have explored other candidate markers for stumpies, and the obtained data are now shown in the new Suppl. Fig. 5.

- How do the authors reconcile their findings with previous reports showing stumpy forms in the skin? Similarly, how do the authors reconcile their findings with the existence of endemic foci where transmission still occur?

These are exciting and relevant questions, for which answers may still not be unequivocal.

Let us first address the question regarding stumpies in the skin. Indeed, the evidence for the presence of T. brucei stumpies in the skin and their transmissibility is convincing (as reviewed in Alfituri et al. 2020). However, these studies examined mainly T. b. brucei subspecies. Capewell et al. in 2016 estimated that stumpy forms represent 20% of the skin-dwelling population. Interestingly, they also evaluated T. b. gambiense PA strain and found out that in some cases, it has even higher skin parasitemia than T.b. brucei control. The histological sections showed stained parasites most likely of BS-LS morphology. A recent study in a gHAT focus in the Republic of Guinea reported the presence of extravascular skin-dwelling T. b. gambiense parasites in HAT-seropositive humans. Still, their morphology was not further examined (Camara et al., 2020). Altogether it seems that T. brucei cells can be sequestered in extravascular tissues irrespective of the bloodstream stage, but there are still many open questions worth exploring.

Regarding the second question. As highlighted recently in a review by Alfituri et al. (2020), the number of humans infected by HAT decreases, and WHO aims to interrupt transmission by 2030. The occurrence of latent human infections (asymptomatic individuals but with possible skin-dwelling parasites) may challenge these goals. Although experimental tsetse fly infection studies with T. b. brucei revealed that skin-residing (stumpy) parasites could infect tsetse flies when ingested by the fly, the jury is still out whether this finding can be extrapolated to T. b. gambiense. It is puzzling where and how the tsetse flies get infected with the HAT-parasite in an endemic region and sustain transmission.

We want to point out that T. b. gambiense is less prone than T. b. brucei to make stumpies under the conditions used. It remains a possibility that T. b. gambiense could form stumpies in other anatomical sites or within alternative species.

Line 103, 449, Were the heme assays performed on cultured T.b. gambiense in the absence of human serum?

The assay was performed on the ex vivo cells isolated from mice.

Reviewer #2 (Remarks to the Author):

This manuscript analyzes the capacity for heme uptake in Trypanosoma brucei and human infective Trypanosoma gambiense and links expression of the HpHbR to this. The authors further demonstrate that the lack of a heme-dependent enzyme renders parasites less sensitive to an inhibitor of that enzyme, which can be used as an assay for heme uptake and functional delivery. Finally, the authors observe that there are changes in the capacity for the development of parasites to the stumpy stage (BS-ST) when HpHbR is deleted, and in T. b. gambiense (which has a naturally mutant HpHbR), and suggest the developmental capacity is restored, although incompletely, with reintroduction of a T. brucei HpHbR.

The heme uptake assays are clear (although this part is already known I think) and the assays of Cyp51 are consistent with the overall thesis of the study- i.e. that HpHbR allows heme uptake and supply to enzymes such as CYP51. However, for this reviewer there are quite a

few experiments required to substantiate the link between heme uptake and the developmental capacity of *T. b. gambiense* and/or heme uptake capacity. The interpretation of the current experiments also needs clarification.

1. The experiments presented in Figure 1 appear clear, although I think the main conclusion is already known.

We dare to disagree. The finding that *T. b. gambiense* contains a negligible amount of heme is novel, and also the monitoring of the activities of heme-carrying proteins. We want to stress that the reported cell-associated heme in wild types (Vanhollebeke et al., Science 2008) could trivially represent endocytic cargo, and Fig. 1 and Fig. 2 are the first direct evidence for heme-based metabolism in the bloodstreams.

2. On Figure 2 the effects on cell growth caused by ketoconazole would be better normalized for each cell line to their own growth in the absence of drug (rather than to the wild type cells in the absence of drug).

All graphs in Fig. 2 have been modified according to reviewer's suggestions.

3. Figure 3a measures luciferase activity, which many not reflect mRNA (or even protein) levels. The result seems OK, but levels of Luc determined by qRT-PCR or western blot might be valuable.

Quantification of Luciferase mRNA has been added as Suppl. Fig. 2B.

4. Figure 4e shows large variation based on the error bars. With mouse infections this is normal, but it is valuable to show the profile of the infection in each mouse to assess if the higher or lower average parasitaemia is a consequence of only one or two infections being atypical.'

In compliance with this request, we have now added the information on individual infections (see Suppl. Fig. 4).

5. In the work presented in figure 4 I consider it important to add back the *T. b. gambiense* HpHbR to the *T. brucei* KO cells. This should not restore developmental capacity, unlike TbHpHbR. Otherwise there is the risk that the developmental phenotype caused by TbHpHbR is simply a consequence of heterologous receptor/protein expression unrelated to receptor function. This is relevant because cells experiencing unphysiological conditions can show some progression to BS-ST characteristics. It would also be interesting to see the phenotypes generated by either TbHpHbR or TgHpHbR expressed from the 18s rRNA locus, since the expression from the in situ locus is weak compared to wild type. If it is as strong as in wild type cells, then much more obvious developmental phenotypes should be seen if TbHpHbR expression is limiting (and little would be seen with Tg HpHbR expression, again confirming the importance of receptor function).

We thank this reviewer for this important control. We decided to perform the technically non-trivial experiments suggested by this reviewer. Hence, we have generated new strains for this aim. The HpHbR KO expressing *T. b. gambiense* HpHbR from the 18S rRNA locus showed a similar expression to its *T. b. brucei* variant (see new Suppl. Fig. 3). This allowed us to compare the two strains in vivo along with their resulting phenotypes. The new add

back cell line showed a lower ability to rescue the HpHbR KO phenotype than the original T. b. brucei variant (new Suppl. Fig. 4). This confirms the importance of a fully functional receptor for stumpy formation.

6. Conversely, in Figure 5 I consider it would be important to 'add back' T. b. gambiense HpHbR expression in WT T. b. gambiense from the 18S rRNA locus to match the expression of the T. brucei add-back in that cell line. If the story is correct, the TgHpHbR should not drive any developmental response/PAD1 expression where the TbHpHbR does. This would confirm the growth effect and possible development seen with the expression of the T. brucei protein is due to its receptor function under conditions of equivalent protein expression.

Following this suggestion, we generated a new cell line of T. b. gambiense overexpressing T. b. gambiense HpHbR and scrutinized it in vivo (see Fig. 5). We found this cell line to give the same parasitemia in the mouse model (shallow parasitic wave peaking at day 4) as the T. b. brucei variant. However, strikingly, there was no increase of the PAD1-positive cells observed for the newly generated strain, as reported for the T. b. brucei variant, further supporting our hypothesis about the ability of the T. b. brucei receptor to enhance the stumpy formation.

7. Following on from point 6, what is the level of the expression of the add back T. brucei HpHbR in the T. b. gambiense cells? The restoration of developmental competence is quite subtle (notwithstanding the growth effect in vivo- the growth in vitro is not presented but would be of interest) : morphologically BS-ST cells are not clearly generated (by cell volume) and PAD1 expression is very limited. If expression of the T brucei protein were efficient I would have expected a better level of development if this receptor is an important contributor to the developmental response. The cell cycle status of cells expressing the TbHpHbR should be presented in addition to their growth also- since development to BS-ST should show an increase in cell cycle arrested forms (1K1N).

We now newly show a relative expression of HpHbR in all addback cell lines generated in the study (see Suppl. Fig. 3C).

8. If heme levels are important in the process of development, the capacity of T brucei and T gambiense parasites to express PAD1 and develop to BS-ST forms could be explored in vitro using heme deficient or haptoglobin depleted medium, or in media with excess heme. These assays would allow quantitative and titratable effects to be observed to link uptake efficacy and development and are tractable for monitoring BS-SL and BS-ST expression in vitro.

These are undoubtedly relevant experiments to address the direct role of heme in cell differentiation which we want to study in more detail in the follow-up study. For now, we softened our wording about this possibility.

Overall, I am convinced by the data on heme uptake and the importance of the HpHbR in this process- this is a valuable finding (although some of it is known). I remain to be convinced that there is a consistent and causal relationship between HpHbR and developmental capacity in T. b. gambiense. As the authors state (line 369-370), different isolates of T. b gambiense show different levels of slender and stumpy formation in different field isolates, although presumably all have HpHbR mutations to permit human infectivity. Hence, evidence that there is a robust relationship between these two observations (HpHbR function and

developmental capacity) is not yet established – the *T. b. gambiense* lines used in their study are of uncertain passage history in mice; other field isolates may behave differently.

We newly added one more *T.b. gambiense* strain (PA, see Suppl. Fig. 5) which indeed behaved differently in terms of stumpy marker genes expression, and we reformulated our text accordingly.

I also found the interpretation in the discussion to be quite muddy. On one hand, evidence is cited that *T. b. gambiense* is transmitted by flies poorly, which would be consistent with a proposed link between HpHbR efficacy and development. On the other hand, an unpublished BioRxv report is cited of equivalent fly infectivity for slender and stumpy forms (albeit using unusually permissive flies) and used to suggest that the developmental capacity is not evolutionary constraint for *T. b. gambiense*. These arguments seem contradictory.

The low ‘intrinsic’ transmissibility of *T. brucei gambiense* for tsetse flies has been reported in the literature. It is based on data from field-collected flies in HAT-endemic regions and experimental infection studies. Notably, reported low transmissibility is mainly due to a low success in developing a procyclic midgut infection into final metacyclic salivary gland infections. Still, tsetse flies can be infected with these parasites, and transmission of gambiense-HAT occurs solely through these flies.

Indeed, Schuster et al. (preprint bioRxiv717975) reported the ability of slender *T. b. brucei* to infect tsetse flies, with comparable infection rates as obtained for the short stumpy forms. The reviewer correctly noted that this experiment seems to be performed in strong permissive flies generating high infection rates for both trypanosome forms, including the long slenders. The ability of *T. b. brucei* long slender bloodstream forms to transform into a procyclic infection in the tsetse midgut has already been reported in the older literature and it was recently confirmed by us (JVDA, personal communication). However, these experiments always revealed a strong and significant difference in the resulting midgut infection rates, with *T. b. brucei* long slender proliferative forms giving rise to only a few midgut infected flies, if any. Specifically, 3% (4/134) versus 25,2% (37/134) midgut infected flies after an infective blood meal with *T. b. brucei* AnTAR1 long slender (morphologically 100% slenders) and short stumpy trypanosomes (> 60% short stumpy morphology), collected from infected mice at a different stage of the parasitemia. These data confirm the intrinsic biological differences between those two bloodstream stages in their ability to survive and/or transform in the tsetse midgut into an established procyclic infection. Interestingly, the very low transmissibility of the long slender forms could be counteracted by adding 10 mM reduced L-glutathione to the infective blood meal (raising the midgut infection rates to > 70% for the long slenders and 86% for the short stumpies). This suggests that the addition of this antioxidant makes the midgut more permissive (less hostile) for the ingested bloodstream forms, especially for the long slenders. The ‘contradictory outcome’ on the low tsetse transmissibility of long slender *T. b. brucei* bloodstream forms reported in the literature and confirmed by our experiments and the data of Schuster et al. could be possibly linked to their different experimental conditions, i.e. tsetse flies with a higher midgut permissivity for bloodstream forms, the use of in vitro cultured /collected long slender & short stumpies, instead of the in vivo grown ones in previous/our experiment, among maybe other experimental differences (e.g. blood source to feed the flies) that could have modulated the trypanosome permissiveness in the fly midgut.

We have tried to incorporate this valuable information into the manuscript (where appropriate) to indicate the existing controversy.

Finally, I assume that the heme dependent response is secondary to density dependent stumpy formation by quorum sensing (the dominant mechanism driving transmissibility)? Since heme availability and uptake would likely not be density dependent (nor limiting in vitro), it seems the developmental effects (if validated) would be a consequence of a heme-dependent processes in the BS-ST pathway downstream of the initial stimulation. This can make sense and would not be surprising given the mitochondrial involvement and metabolic adaptations inherent in BS-ST development. However, without some further experimentation and validation I left uncertain of the importance or mechanism of the developmental aspect of the work, despite interest in the observations reported.

The density-dependent stumpy formation is undoubtedly a well-documented mechanism. We have the data showing in vitro differentiation with cis-aconitate, indicating that heme would act downstream of that process. Moreover, we see the crosstalk between the stumpy formation and heme uptake (via HpHbR), a connection not noticed before.

Minor issues

1. Figure 1d would benefit from molecular weight size markers (the relevant bands are labelled but the size would be good to add).

As requested, the markers have been added.

2. Line 125, referring to figure 2A, says that CYP51 is differentially transcribed in procyclic, slender and stumpy forms. Since most regulation is posttranscriptional, it is more likely to have differential mRNA abundance but not be differentially transcribed. This should be amended.

Indeed, it has now been rephrased as follows: "The genome of T. brucei s. l. encodes a single copy of CYP51, with the highest mRNA abundance in PS compared to BS-SL and BS-ST cells with relatively low levels (Fig. 2A)."

3. On Figure 2f and g the Y axis is broken to stretch out the region between 50-100% and to compress 0-40%. I don't think this is necessary and is potentially misrepresentative of the scale of effects seen.

The graphs in Fig. 2 have been modified to take into account all referee's suggestions.

4. Line 181 refers to the PS to BS differentiation, but the differentiation is actually BS to PS. Please correct.

Has been corrected.

5. Line 240 'bulky HpHbR expression'. I don't think bulky is quite the right word. There are several other places where the phraseology or English usage would benefit from attention.

We replaced the "bulky" for "robust" in the text.

Reviewer #3 (Remarks to the Author):

The human infectivity of Tbg is, unlike Tbr, multifactorial. Decreased functionality and decreased expression of the HpHbR is known to be one of the evolutionary modifications that have allowed Tbg to infect humans, through reduced uptake of TLF1. Here, the authors main claims are: (1) That both Tbb HpHbR KO and Tbg cells lack hemoprotein activity and therefore proliferate in the absence of heme. (2) That HpHbR KO prevents differentiation of the parasites into stumpy forms. (3) That Tbg is poorly competent to differentiate into stumpy forms, due to reduced functionality of HpHbR

This is an interesting and timely study that would certainly contribute to the field and raise questions for further study. There are some revisions that are required in order to support the claims presented and the text needs to be clarified to state only what is experimentally shown.

Main claim 1: That both Tbb HpHbR KO and Tbg cells lack hemoprotein activity and therefore proliferate in the absence of heme.

The Tbb HpHbR KO cell line has previously been generated and characterised for its inability to internalise HpHb or free heme, despite little growth effect in vitro (Vanhollebeke et al., 2008). It was also previously stated in the same work, that: "The steady-state heme content of WT cells isolated from mice was 2.3 ng/mg of protein, whereas heme was undetectable in KO cells" and that "heme appeared to be mostly incorporated into hemoproteins" which would together then strongly support that the Tbb HpHbR KO cell line should indeed have an absence of hemoprotein activity and proliferate in the absence of heme. Here, this previous data is confirmed with further experimental support. The measurement of the intracellular heme content of Tbg is novel and of significant interest, given the previously observed mutations and changes in gene expression of the TbgHpHbR. The authors also assess, for the first time, hemoprotein activity between these cell lines via introduction of a human hemoprotein in Tbb WT, Tbb HpHbR KO and Tbg cells and via endogenous CY51 hemoprotein activity in Tbb WT, Tbb HpHbR KO and Tbr cells.

Heme measurements are conducted and quantified, with measurements for TbbHpHbR KO and Tbg close to the limits of detection and with no significant difference detected between them. That being said, the chromatogram from Fig 1a and quantification from Fig 1b, does imply that Tbg has some, albeit very little, internal heme.

We would question the reviewer's assumption that the heme is internal. It could be sticky heme to the coat, secondary to the isolation process.

The addition of human catalase into Tbb, TbbHpHbR KO and Tbg cells was a good experiment as this allows the cell lines to be compared directly for hemoprotein activity without consideration for other mutations or changes in gene expression that may be distinct between the cell lines. This data demonstrates that the acquisition of heme in Tbg does not support any significant activity of human catalase, although again, the data does imply a very mild activity as the shape of the measurement curve (Fig 1f) does have a small jump at the time of H₂O₂ addition, unlike the TbbHpHbR KO cells.

Given the high level of intracellular heme in the Tbb WT cells and the trace amount in Tbg cells, it could be argued that the amount of heme present in Tbg is enough for some activity of endogenous hemoproteins but is not in excess and therefore unable to fuel the human catalase. The authors themselves describe this as 'potent heme-dependent activity [through overexpression]' (line 319). For this reason, it would be particularly valuable for the authors to assess the endogenous hemoproteins of Tbg, as they have done for Tbb WT and TbbHpHbR KO cells in Figure 2. In fact, it is very unclear as to why the authors have included a ketoconazole sensitivity experiment for Tbr (Figure 2G), when they have not included this for Tbg. This should be added and would considerably strengthen the manuscript. Also, with Tbr not particularly the focus of this work, it is not clear why the experiment in Fig 2G was conducted on Tbr at all, rather than the Tbb WT and Tbb HpHbR KO cell line.

We want to add that the experiment was meant to validate the HpHbR requirement through its cognate ligand, the HpHb complex. Therefore Hp-deficient serum has to be used, either Hp-/- mice serum or Hp-/- human serum on cells with the fully functional receptor. T. b. rhodesiense grows well in human serum (hence this choice), while growing T. b. brucei in vitro in mouse serum has always proven difficult (and obviously, T. b. brucei cannot be grown in human serum due to the lysis by TLF1).

Throughout, the manuscript text could be clarified to ensure that the authors are clear on the conclusion that they are presenting regarding the TbgHpHbR expression / function based on their data and previously published work. Absence is not the same as a little and poorly capable is not incapable. The data in this manuscript and in published literature fully supports a decreased functionality (i.e. reduced binding capacity for HpHb) and decreased expression of the TbgHpHbR (mRNA and protein). It does not support a complete loss of the TbgHpHbR. Perhaps the authors would like to comment or speculate on the observation that there has not been a complete loss either via expression or a true loss-of-function mutation. Would this imply that there is indeed some residual use of the receptor and/or benefit to maintain it?

Thank you for the valuable comment. We tried our best to be more explicit with our wording. We agree that the gene will probably be eventually lost if not used/needed throughout evolution. We have to consider both the possibility that the time frame was too short for its loss until the present or that it was kept for another function.

Main claim 2: That HpHbR KO prevents differentiation of the parasites into stumpy forms.

HpHbR is downregulated upon transition of BS-SL cells to BS-ST cells. The authors investigate if this due to position on the polycistronic unit, which it is not. This section seems to be addressing a rather different hypothesis to the focus of the rest of the manuscript (mechanisms of gene expression during differentiation rather than evolutionary trade-offs of reduction of heme uptake) and could be moved to supplementary.

We agree, and Figs. 3A and B have been moved to the supplementary material (now Suppl. Fig. 2C, D).

The authors also show that the presence of HpHbR in BS-ST forms does not in any way hinder the transition to BS-ST or PS forms. The authors could more clearly indicate what the hypothesis for this experiment was; was it that there may be a selective downregulation of the

HpHbR at this stage as opposed to being subject to the general translational repression observed in BS-ST cells (Brecht and Parsons 1998)? Upon expression of the HpHbR in BS-ST forms, the authors state that the cells "...still differentiate into the BS-ST and subsequent PS and retain the ability to infect tsetse flies" (line 208). This language implies that they anticipated a hinderance.

Indeed, Fig. 3C-G (now Fig. 3A-E) shows a gain of function approach. One could think that the continuous import of heme into cell cycle-arrested stumpy cells could be detrimental. More so, the abrupt reduction of heme influx could have been a mechanism triggering differentiation into stumpy cells. These experiments exclude these hypotheses, which are now better explained in the text.

To the contrary, given that they later state that presence of heme is required for differentiation of the parasites into stumpy, did they consider if the extended expression of HpHbR may actually benefit this transition? As shown in Figure 3d, the HpHbR-3'PAD1 cell line, as an average, peaks a full day earlier than the WT and at a moderately lower density which would be consistent with an earlier transition to BS-STs. Although I can see that the error bars on parasite load would likely overlap, the trend on rising and falling parasitaemia seems clear. Is this a reproducible difference between individual mice and/or different experiments? Was this ever studied with 12 hour time points between day 3-6? If the BS-ST forms or PC forms have a greater reliance on hemoproteins, then perhaps the authors might consider if the extended expression of the HpHbR in this case increases heme availability/hemoprotein activity during this transition.

Thank you for this comment; this is an exciting thought. On the other hand, we don't think that HpHbR-3'PAD1 cells are significantly different from WTs. The same WT cells (90-13) were also used in Fig. 4E where they peak later (day 5-6).

A pleomorphic HpHbR KO and two add-back cell lines (18S and in situ) are produced; no validation of the add-back cell lines is included and the authors should therefore add this to Supp Fig 2 (genetic validation or assessment of expression levels). These cell lines are assessed functionally through the number of Hp+ cells after exposure to labelled Hp. The WT cells have a value of 80% with the add-backs providing 70 or 20%. Could a low number of positive cells in the in situ add-back also represent a non-clonal culture where not all cells are expressing the gene?

The qPCR analysis of HpHbR expression for all variants of the add back cell lines has been performed and included (Suppl. Fig. 3C, D). Although we cannot exclude the non-uniformity of the population, all the cell lines were prepared consistently by limiting dilution.

At this stage, the authors have sought to "...test whether heme and/or HpHbR play any role in [BS-ST formation]" (line 217). The data collectively presented in Figure 4 convincingly demonstrates that the TbHpHbR KO cell line does not form BS-STs and that this is due to the absence of the HpHbR. But this does not differentiate between heme and/or HpHbR and it is not clear from the authors discussion what their conclusion for the molecular basis for this is. They state "The fact that in the absence of HpHbR, the key BS-SL to BS-ST transition is disrupted, prompts us to suggest that heme uptake may be an additional player in this life cycle progression" (line 340), and that "...HpHbR facilitates the developmental progression

by inducing PAD-1 expression" (line 44). Both of these statements are implicating the HpHbR itself in developmental regulation and/or signalling. However this is distinct from a more likely hypothesis that BS-ST forms simply have at least one required hemoprotein and therefore unlike BS-SL forms, BS-ST forms do require heme. It would be beneficial to this work if the authors did experimentally address this distinction, by expressing the TbHrg in the TbHpHbR KO or assessing hemoprotein activity in BS-ST, for example. This also has implications for claim 3.

Those are interesting and stimulating proposals, but we believe they are out of the scope of this study. We know that TbHrg is indeed upregulated in BS-ST compared to BS-LS cells from the qPCR analysis we performed here (see Suppl. Fig. 5). It is well documented for stumpies that they have more branched mitochondrion with respiratory complexes present, demanding heme (Capewell et al., 2013). So, yes, we agree, stumpies probably need heme, but HpHbR does not provide it anymore.

Main claim 3: That Tbg is poorly competent to differentiate into stumpy forms, due to reduced functionality of HpHbR.

The finding that Tbg produces little/no BS-S Relative *HpHbR* mRNA level and of interest to the field. However, while the authors have clearly uncovered some interesting biology here, the data presented does not satisfactorily support the general claim that Tbg is poorly competent to produce BS-ST. It is very possible that, in evolving the adjust to the modifications of the reduced functionality of the TbGpHbR, that Tbg BS-ST are not identical to Tbb BS-ST (perhaps reducing reliance on hemoproteins). This would be very interesting and authors have clearly considered this possibility as they state "WT Tbg has a poor capability to generate typical BS-ST" (line 343). The data presented should be published and is of interest, but I do not think it can be summarised as demonstrating that "Tbg is poorly competent to differentiate into stumpy forms".

Figure 5a-d clearly demonstrates that very few PAD1+ cells/ morphologically BS-ST cells are identified in the Tbg WT and this is moderately recovered with add-back of the TbHpHbR. However, the growth curves from the mouse experiment indicates that the differences in dynamic here may be more complex. The authors have shown that an absence of HpHbR for Tbb results in an outgrowth of the parasites which would ultimately overwhelm the host (Figure 4e). This uncontrolled growth is typical of a 'monomorphic' cell line which does not produce BS-ST. This is not what is observed with the Tbg WT, where the parasite numbers do fall on day 5. Indeed, it is known than Tbg causes a chronic infection in humans and can result in asymptomatic infections, whereas an absence of BS-ST forms in Tbb (as per Fig 4e) is shown to cause uncontrolled growth. This again supports a hypothesis that there is a distinct situation in Tbg, but not simply through a lack of BS-ST forms.

The addition of the TbHpHbR to Tbg very much suppressed parasitaemia which, based on cell number alone, might indicate a dramatic shift towards differentiation to BS-ST, but this is not reflected in the PAD1+ assay, where only around 30% of the cells are positive. A straightforward additional experiment of interest here would be to assess the parasites on day 4 for their cell cycle status to determine if they are or are not undergoing a cell cycle arrest. Further, a consequence of loss of BS-ST formation for the TbbHpHbR KO was an additional reduced capacity to generate PS, which was not assessed for Tbg. Ideally it should be, since the authors do discuss a reduced capacity to infect flies in their discussion (Line 364) and it

would support their hypothesis if they saw increased fly infectivity upon TbHpHbR overexpression in Tbg.

We newly generated the cell line overexpressing TgHpHbR in Litat 1.3 strain, and in vivo the parasitemia was also suppressed similarly to TbHpHbR (see new Fig. 5A), but independently of stumpy forms which were not detected (Fig. 5C). To address T.b. gambiense differentiation in vitro is not that straightforward. Litat 1.3 is the only T.b. gambiense strain that can grow in the culture. Still, after so many years in the laboratory conditions, it is essentially a monomorphic strain that lost the capacity to differentiate into PS.

The data presented in Figure 5e-h regarding the Tbg Bosendja Field strain was generated to address if this strain "... sustains the ability to produce waves of parasitemia". This is not addressed as the experiment is terminated on day 4. It would have hugely benefited from being continued to day 5. The parasitaemia in the mice could have dropped as for the Tbg WT (Fig 5a) or continued to rise as for the Tbb HpHbR KO (Fig 4e) and so no conclusions can be drawn either way.

We tried to extend the experiment up to day 5 for Bosendja strain, but unfortunately, the mice did not survive, most likely due to elevated parasitemia.

My final concern with the statement that Tbg is poorly competent to generate BS-ST is that "... publications describe a high variability in the proportion of the BS-SL to BS-ST cells in different field strains of Tbg" (line 369) and we know that different field isolates have different levels of HpHbR expression as well (Kieft et al., 2010). The data presented should be considered more broadly within the context of the variability of Tbg isolates, which may actually support the authors hypothesis, if the isolates with higher levels of reported BS-ST forms had correlatively higher levels of HpHbR expression.

We are newly showing the transcription profiles of genes upregulated or downregulated in the BS-ST forms in different T.b. gambiense strains (see Suppl. Fig. 5). We must admit that the picture is rather complex since even cells from the same group (T.b. gambiense Group 1) express various levels of the marker genes. We are therefore more cautious about the extrapolation of our finding to the whole T.b. gambiense group. We also checked the level of HpHbR expression, and it is not in line with the published data (Kieft et al., 2010). Unlike in their study, here we report higher HpHbR transcripts in T.b. gambiense cells than in T.b. brucei controls. On the other hand, we have to consider that the ex vivo cells may be isolated from the host at different time points, which may cause a significant difference in HpHbR expression. Moreover, since BS-ST cells do not produce HpHbR anymore, the hypothesis may be formulated the other way around, meaning more HpHbR less BS-ST and vice versa.

Text

There are a few statements that are either incorrect or written in such a way as the scientific interpretation is incorrect or ambiguous. These should be modified to ensure accurate reflection of the literature or point being made.

Line 36: "... the physiological function of the receptor remains to be elucidated". Do the authors mean the function is yet to be elucidated in Tbg specifically? Because the function of

the HpHbR as a receptor for HpHb in Tb is already known (Vanhollebeke et al., 2008; Stødkilde et al., 2014).

The fact that HpHbR binds HpHb was known. Why and when in the life cycle this is important to the parasites is novel, though. Moreover, what was not studied in detail is where the heme is needed as a cofactor in the cell. Therefore, we hope we addressed the physiological function by scrutinizing the genuine hemoproteins such as CYP51 in bloodstream cells lacking the HpHbR.

Line 44: "... HpHbR facilitates the developmental progression by inducing PAD-1 expression..." The authors do not show that HpHbR induces PAD1 expression.

***We agree. The sentence has now been corrected as follows:
"We further show that HpHbR facilitates the developmental progression to cell cycle-arrested stumpy forms in T. b. brucei."***

Line 63: "...HpHbR, the only invariant cell surface receptor known to date in kinetoplastid parasites". T. brucei Factor H receptor. (Macleod et al., 2020).

Thank you very much for this addition, as we overlooked this recent publication at the time of the writing. We now refer to it and reformulate the sentence as follows: "Hemoglobin receptor (HpHbR), one of the few cell surface receptors known to date in kinetoplastid parasites (Vanhollebeke, B. et al., 2008; Stødkilde et al. 2014, Macleod et al., 2020)."

Lines 79-82: "Retaining the HpHbR expression contribute to trypanosomes fitness in their animal reservoir hosts, providing positive selection pressures for the conservation of this receptor ". Are the authors referring to Tbg or non-Tbg Tb species? It is not clear if the point being made is that Tbg have retained the HpHbR and this contributes to fitness in animals or that there is advantage to non-gambienese Tb, which Tbg no longer have.

Here we meant fitness of T. b. brucei specifically. Therefore, we have now rephrased the text to make it more clear as follows: "Retaining the HpHbR expression contributes to T. b. brucei fitness in their animal reservoir hosts, providing positive selection pressures for the conservation of this receptor (6, 14)."

Line 82-85: Reference(s) required.

The reference has been added (Welburn et al., 2016).

Line 174-176: Reference or names of 'other genes' and/or data required. By 'involved' do the authors mean differentially regulated (of which there are very many genes) or do they mean playing an active role in the transition?

As suggested here, this data (former Fig. 3A-C) has been moved to the supplementary section (now Suppl. Fig. 2B-D). We simplified this part in the text accordingly, and we are now mentioning specifically only the HpHbR gene.

Line 238-240 and lines 262-264: Can the authors clarify the justification here. What do they mean by bulky? Expression levels have not been shown in this manuscript. Further, the data in Figure 4a indicates that the 18S add back is more similar to the WT and the authors proceed to use the 18S locus for the Tbg add back.

We are not using "bulky expression" in the text anymore, but "robust expression" to describe the expression from the 18S rRNA locus. Additionally, we newly show the level of expression for all the add backs used in this study and its comparison to the respective wild types (see Suppl. Fig. 3C, D). In T. b. gambiense add back we took advantage of the cell line T. b. gambiense +b1 (newly named T. b. gambiense + b^{18S}), which some of us previously generated (Uzureau et al., 2013).

Line 260: The title of this section and Figure 5 are "Restoration of stumpy formation in Tbg". This is not ideal, as it indicates that the authors have already shown Tbg do not form BS-ST. "Overexpression of TbbHpHbR in Tbg facilitates increased BS-ST formation" or similar, would be more suitable.

Thank you for the helpful comment. The section is now named as follows: "Overexpression of T. b. brucei HpHbR in T. b. gambiense increase stumpy formation".

Line 267: "... on day 6..." should be on day 5.

Has been corrected as suggested.

Line 271: "did not detect any PAD1-expressing cells". Data in 5c does have some indication of a very small number of PAD1+ cells.

The text has been reformulated as follows: "In the WT T. b. gambiense, we detect only a negligible amount of PAD1-expressing cells."

Lines 323-329: The authors discuss the anaemia of animal trypanosomiasis and then comment on the lack of such observations in human Tbg studies. Are the authors implying a cause-and-effect relationship and in which case, which direction? Do Tbg have less use for a HpHbR because there is less free HpHb, which is not necessarily logical because reduced concentration of ligand in blood could select for greater binding capacity in order to compete with host macrophages.

No, we do not imply a direct cause-and-effect relationship between the anemia and T.b. gambiense HpHbR. We assign the lack of anemia in T.b. gambiense to the nature of the disease with a chronic course of infection.

Lines 330-336: Please clarify this hypothesis, particularly line 334-336.

We agree that the statement following the references to what is known is rather vague. It has been reformulated as follows: “ Thus, we propose that the excess of free hemoglobin released by trypanosome infection possibly modulates heme uptake and, subsequently, parasitic waves.”

Line 357-359: "... the lower pathogenicity of the HpHbR KO T. b. brucei for both the Hp-carrying and Hp-lacking mice (Vanhollebeke et al., 2008)". This is not what the data in this manuscript supports, which demonstrates increased pathogenicity after day 6 (Fig4e).

We have to stress that the older publication (Vanhollebeke et al., 2008) used monomorphic HpHbRKO strain in contrast to this study where we newly generated the HpHbR KO in the pleomorphic strain. Both studies refer to the earlier parasitemia (day 4 and 5 p.i.) in mice where the HpHbR KO is attenuated compared to WT strain. We attribute the later (day 6 p.i.) increased pathogenicity to the lack of stumpy forms and not to the fitness of the cells.

Lines 367-369: "To the best of our knowledge, the reporting of putative BS-ST cells in T. b. gambiense was based solely on morphological criteria and may have resulted in a misassignment to a different T. brucei sub-species that is more prone to BS-ST transition" Are the authors stating they believe that publications 50, 51, 52 and 53 have all incorrectly utilised Tbb instead of Tbg in their studies?

No, we certainly do not claim that those publications incorrectly utilized the T. brucei subspecies; we raise the possibility that this might have happened in some cases, especially in the older publications. However, to prevent any misjudgment from our side, we removed the statement about the misassignment to different T. brucei subspecies.

Figures

Figure 2b: mRNA levels of CYP51 knock down should be shown in supplementary data to allow the reader to assess extent of knock-down.

BF CYP51 RNAi has been removed; only the CYP51 KO is now shown to streamline the manuscript.

Figure 2d: I question the normalisation of the CYP51 RNAi +dox to the growth of the CYP51 RNAi –dox and the CYP51 KO to the WT in this experiment because these cell lines already have a difference in growth rate and so at 0uM ketoconazole the growth over 48 hours would be reduced. This makes interpretation of the data somewhat problematic. The authors could replot the data with each cell normalised to its own growth rate to determine if this improves clarity. Particularly, it should then be clearer to observe the presence vs absence of the biphasic curve. This is simply a suggestion to try and improve visualisation of this data. Without being able to see what that would look like, I do appreciate that upon doing that it may not improve the figure. In either case, the area of the figure that is of most interest is the range of 1-10uM (given this is the range then used in Figure 2f and 2g, as well as the area required on the WT curves to observe the biphasic nature of the curve).

This is not particularly clear in the figure and this should be modified where possible to improve this (data point size reduced, possibly open boxes for the CYP51 KD / KO).

Fig. 2D (now 2C) has been modified with normalized growth for WT and CYP51 KO.

Figure 3d: A supplementary file showing individual mouse infection profiles would be beneficial here.

As suggested, individual mouse infections are now shown as Suppl. Fig. 4.

Figure 4c: The Y-axis is labelled procyclics. Is this accurate in that only 'procyclic' cells would be counted?

We are pretty confident that only the procyclic stage was counted based on morphology. Nevertheless, the cells were not labeled with procyclin marker in this particular experiment. Therefore, we decided to relabel the Y-axis as Cells.

Figure 4c and 4d: No error bars are provided, although replicate numbers are given.

The error bars have now been provided.

Figure 4f and 4h: Given that the PAD1+ cells have a greater cell area and shorter N to k distance, the calculation of the mean values in the whole populations and then use of these to measure significant difference seems inappropriate. The data in Figure 4h should be provided before 4f and used for statistical analysis. The data in 4f should then be separated such that there are six sets of data per graph (WT PAD1-, WT PAD1+ and so on). This would exclude the ability to compare data in 4f with statistics, but would clearly demonstrate the point being made.

We partially modified this figure by reshuffling the individual section as suggested. Nevertheless, we believe that splitting the PAD1 positive and PAD1 negative cells would be somewhat confusing.

Definition of error bars is in the methods rather than in each of the figure legends. Multiple cases of the number of replicates not being included in Figure legends (i.e. all of the figure 2 panels, Figure 4e).

The number of replicates and statistics are now given in each Figure legend.

References:

Alfituri, O. A., Quintana, J. F., MacLeod, A., Garside, P., Benson, R. A., Brewer, J. M., Mabbott, N. A., Morrison, L. J., & Capewell, P. To the Skin and Beyond: The Immune Response to African Trypanosomes as They Enter and Exit the Vertebrate Host. Front Immunol. 11, 1250 (2020).

Camara, M., Soumah, A. M., Ilboudo, H., Travaillé, C., Clucas, C., Cooper, A., Kuispond Swar, N. R., Camara, O., Sadissou, I., Calvo Alvarez, E., Crouzols, A., Bart, J. M., Jamonneau, V., Camara, M., MacLeod, A., Bucheton, B., & Rotureau, B. Extravascular Dermal Trypanosomes in Suspected and Confirmed Cases of gambiense Human African Trypanosomiasis. *Clin Infect Dis.* 73(1), 12–20 (2021).

Capewell et al. Regulation of *Trypanosoma brucei* total and polysomal mRNA during development within its mammalian host. *PLoS One* 8:e67069 (2013).

Capewell, P. et al. The skin is a significant but overlooked anatomical reservoir for vector-borne African trypanosomes. *eLife* 5, e17716 (2016).

Janelle, J. et al. Monitoring the pleomorphism of *Trypanosoma brucei* gambiense isolates in mouse: impact on its transmissibility to *Glossina palpalis gambiensis*. *Infect. Genet. Evol.* 9, 1260-1264 (2009).

Macleod, O., Bart, J. M., MacGregor, P., Peacock, L., Savill, N. J., Hester, S., Ravel, S., Sunter, J. D., Trevor, C., Rust, S., Vaughan, T. J., Minter, R., Mohammed, S., Gibson, W., Taylor, M. C., Higgins, M. K., & Carrington, M. A receptor for the complement regulator factor H increases transmission of trypanosomes to tsetse flies. *Nat Commun.* 11(1), 1326 (2020).

Raper, J., Nussenzweig, V., & Tomlinson, S. (1996). The main lytic factor of *Trypanosoma brucei brucei* in normal human serum is not high density lipoprotein. *J Exp Med.* 183(3), 1023–1029.

Uzureau, P. et al. Mechanism of *Trypanosoma brucei* gambiense resistance to human serum. *Nature* 501, 430-434 (2013).

Vanhollebeke, B. et al. A haptoglobin-hemoglobin receptor conveys innate immunity to *Trypanosoma brucei* in humans. *Science* 320, 677-681 (2008).

Welburn, S. C., Molyneux, D. H., & Maudlin, I. Beyond Tsetse--Implications for Research and Control of Human African Trypanosomiasis Epidemics. *Trends Parasitol.* 32(3), 230–241 (2016).

REVIEWER COMMENTS

Reviewer #2 (Remarks to the Author):

I am impressed by the lengths that the authors have gone to to address the comments from all the referees. In particular, I thank the authors for putting in the important effort to generate Tbg add backs at different loci to reassure on the consequences of deletion of the receptor and its relative function in the two trypanosome species. These experiments will have been a significant effort but add substantially to the value of the paper. I think there are questions still to be addressed concerning the contribution of HpHbR expression and the developmental capacity of *T. b. gambiense* versus *T. b. brucei* but the data in this paper has sufficient interest to satisfy this reviewer at present, but with expectation that further work will dissect the observations more fully in future.

Suggested changes:

Experiments

I suggest to repeat Figure 4 C- the message seems clear but the error bars are very large.

Text

I would suggest some modifications in the text discussion of the data where I feel there remains some lack of clarity, risking misinterpretation.

The first relates to the 'controversy' concerning the ability of slender or stumpy forms to progress the life cycle in tsetse (line 393-397). As highlighted in the rebuttal, some old data and a recent paper (Schuster et al., eLife 2021) has reported that slender cells can establish in the fly experimentally, whereas the authors' report their own unpublished fly experiments demonstrating that stumpy cells establish preferentially (supporting the traditional view). In fact, these observations are compatible. Simply, in unusually permissive flies (or with glutathione supplement in feeds) slender cells survive sufficiently to transition to cells with stumpy molecular characteristics (not morphology), particularly PAD1 expression, allowing their onward development in the fly (as shown by Schuster et al, 2021). The same is the case for *T. congolense*, where no morphological transition occurs, but the cells exhibit functional quorum sensing and an intact density dependent signalling pathway (i.e., functionally but not morphological stumpy; Silvester et al, Nature Microbiology, 2017). This generates a quite consistent

view I believe- stumpy cells preferentially infect flies over slender cells but slender cells can do so if they survive long enough to express the stumpy molecules required for differentiation (PAD1 and likely other identified signalling components). So, for the current paper there seems little necessity to invoke a controversy - T. b. gambiense are less able to be transmitted in the field (as opposed to in the lab) if they produce fewer stumpy cells as a consequence of expression of their mutant HpHb receptor.

The second relates, perhaps from my own misunderstanding of the proposed model. How do the authors reconcile that different T b gambiense isolates generate different levels of stumpy cells despite a shared defective HpHg receptor? I see discussion of this in the response to referee 3 but I find the related text in the manuscript (line 398-406) quite difficult to follow. Please could the authors explain their thinking more clearly?

Reviewer #3 (Remarks to the Author):

1) With the exception of figure numbers, the authors have insufficiently signposted in the response to reviewers (i.e. with paragraph or line numbers).

For example, where the authors state that they have been "...more cautious about the formulations in the text" in response to a query from Reviewer 1, the reviewer is left doing all the work to find those sections.

A second example is that Reviewer 1 asks about stumpy forms in the skin. The authors write three paragraphs in response to this reviewer, but make no reference to whether they made any modifications to the manuscript based on this comment or not. If the authors did not then they should state that they haven't and why (i.e. too speculative) or signpost us to where a new comment is made on this subject in the manuscript.

This lack of signposting repeats throughout the response to reviewers and makes for considerable extra workload for the reviewers.

2) All three reviewers question gave similar independent feedback on the conclusions regarding the general ability of Tbg to differentiate into stumpy forms and the direct role of heme uptake in controlling chronicity.

The authors provide strong and informative experiments around this topic (enough to publish and of value to the field) and they have also added qRT-PCR data on stumpy markers, beyond the original manuscript.

However the generality of this has still not been shown and indeed the data on the newly included PA strain demonstrates the diversity involved. The authors acknowledge the complexity of this and have made some suitable adjustments in the text. However, given that this is part of the manuscript title and a key output in the abstract, I still think that the text needs further reworded to clarify between the outcomes as have been shown from the experimental data presented (and its potential significance) and what is still not known.

This is also true for the role of the HpHbR in differentiation in general i.e. there is a difference between it being (i) a fundamental part of the differentiation process involved in the signalling pathway or control of chronicity and (ii) a lack of heme preventing ST formation due to a essentially a nutrient deficiency. The language used (particularly in summarising the work) still doesn't satisfactorily make this clear enough.

REVIEWER COMMENTS

Reviewer #2 (Remarks to the Author):

I am impressed by the lengths that the authors have gone to to address the comments from all the referees. In particular, I thank the authors for putting in the important effort to generate Tbg add backs at different loci to reassure on the consequences of deletion of the receptor and its relative function in the two trypanosome species. These experiments will have been a significant effort but add substantially to the value of the paper. I think there are questions still to be addressed concerning the contribution of HpHbR expression and the developmental capacity of *T. b. gambiense* versus *T. b. brucei* but the data in this paper has sufficient interest to satisfy this reviewer at present, but with expectation that further work will dissect the observations more fully in future.

Suggested changes:

Experiments

I suggest to repeat Figure 4 C- the message seems clear but the error bars are very large.

*As requested, we have repeated this measurement several times and now provide new Figure 4c with all the data combined. The difference between WT and KO strain is more significant now (****). The addback cell lines $ABb^{in situ}(**)$ and $ABb^{18S}(ns)$ rescued the phenotype differently, and the latter strain does not statistically differ from the WT strain.*

Text

I would suggest some modifications in the text discussion of the data where I feel there remains some lack of clarity, risking misinterpretation.

The first relates to the 'controversy' concerning the ability of slender or stumpy forms to progress the life cycle in tsetse (line 393-397). As highlighted in the rebuttal, some old data and a recent paper (Schuster et al., eLife 2021) has reported that slender cells can establish in the fly experimentally, whereas the authors' report their own unpublished fly experiments demonstrating that stumpy cells establish preferentially (supporting the traditional view). In fact, these observations are compatible. Simply, in unusually permissive flies (or with glutathione supplement in feeds) slender cells survive sufficiently to transition to cells with stumpy molecular characteristics (not morphology), particularly PAD1 expression, allowing their onward development in the fly (as shown by Schuster et

al, 2021). The same is the case for *T. congolense*, where no morphological transition occurs, but the cells exhibit functional quorum sensing and an intact density dependent signalling pathway (i.e., functionally but not morphological stumpy; Silvester et al, Nature Microbiology, 2017). This generates a quite consistent view I believe- stumpy cells preferentially infect flies over slender cells but slender cells can do so if they survive long enough to express the stumpy molecules required for differentiation (PAD1 and likely other identified signalling components). So, for the current paper there seems little necessity to invoke a controversy - *T. b. gambiense* are less able to be transmitted in the field (as opposed to in the lab) if they produce fewer stumpy cells as a consequence of expression of their mutant HpHb receptor.

We fully agree with this reviewer's explanation regarding the transmission to tsetse via slender and stumpy trypanosomes. We have altered the text to reflect this and refer to the critical Schuster et al. (2021) paper that was not available at the time of our submission.

Lines: 391-420.

The second relates, perhaps from my own misunderstanding of the proposed model. How do the authors reconcile that different *T. b. gambiense* isolates generate different levels of stumpy cells despite a shared defective HpHg receptor? I see discussion of this in the response to referee 3 but I find the related text in the manuscript (line 398-406) quite difficult to follow. Please could the authors explain their thinking more clearly?

We admit that reconciling our data on the stumpy forms in *T. b. gambiense* obtained in the frame of this work with the patchy and sometimes controversial data available about them in the literature was not straightforward. We hope that we have expressed our view in a way that this reviewer will accept and understand well in the present version. We feel that the strongest argument supporting the importance of HpHbR in the life cycle differentiation is the upregulation of fully functional receptor (TbHpHbR) in *T. b. gambiense* and the subsequent upregulation of the PAD1-positive cells. Although the studied strains of *T. b. gambiense* share the same mutated HpHbR, it is not being expressed to the same extent, which may explain the different expressions of the stumpy marker genes.

Lines: 311-319; 403-411.

Reviewer #3 (Remarks to the Author):

1) With the exception of figure numbers, the authors have insufficiently signposted in the response to reviewers (i.e. with paragraph or line numbers).

For example, where the authors state that they have been "...more cautious about the formulations in the text" in response to a query from Reviewer 1, the reviewer is left doing all the work to find those sections.

We apologize for this oversight and ensure that this did not happen in the second resubmission.

A second example is that Reviewer 1 asks about stumpy forms in the skin. The authors write three paragraphs in response to this reviewer, but make no reference to whether they made any modifications to the manuscript based on this comment or not. If the authors did not then they should state that they haven't and why (i.e. too speculative) or signpost us to where a new comment is made on this subject in the manuscript.

As anticipated, we considered this issue important and devoted part of the discussion to it. The manuscript refers to the studies mentioned in the original rebuttal. Therefore, the possible presence of stumpy forms in the skin is now addressed in lines 412-421.

This lack of signposting repeats throughout the response to reviewers and makes for considerable extra workload for the reviewers.

We are genuinely sorry that we insufficiently signposted our changes in the text. See below the original version of the response to reviewers with line numbers to make any changes as evident and easy-to-track as possible. At the same time, please find the resubmitted version (see file Horakova-MS-first-resubmission_tracked) and this version (see file Horakova-HpHbR-MS-second-resubmission_tracked) with tracked changes for your better orientation.

2) All three reviewers question gave similar independent feedback on the conclusions regarding the general ability of Tbg to differentiate into stumpy forms and the direct role of heme uptake in controlling chronicity.

The authors provide strong and informative experiments around this topic (enough to publish and of value to the field) and they have also added qRT-PCR data on stumpy markers, beyond the original manuscript.

However the generality of this has still not been shown and indeed the data on the newly included PA strain demonstrates the diversity involved. The authors acknowledge the complexity of this and have made some suitable adjustments in the text. However, given

that this is part of the manuscript title and a key output in the abstract, I still think that the text needs further reworded to clarify between the outcomes as have been shown from the experimental data presented (and its potential significance) and what is still not known.

This is also true for the role of the HpHbR in differentiation in general i.e. there is a difference between it being (i) a fundamental part of the differentiation process involved in the signalling pathway or control of chronicity and (ii) a lack of heme preventing ST formation due to a essentially a nutrient deficiency. The language used (particularly in summarising the work) still doesn't satisfactorily make this clear enough.

We agree that these points are very valid and remain controversial. Hence, we did our best to clarify our view on them in the text. We mainly modified the discussion to articulate better our main points and novelties (lines 393-421). We emphasized the main finding of this work that the altered heme uptake (via HpHbR KO) leads to significant changes in the differentiation from the SL to ST bloodstream stage (lines 380-381; 425-426).

We also simplified the summary (lines: 422-427), which is now as follows:

*“In summary, we provided the first direct evidence for heme-based metabolism in bloodstream trypanosomes and their independence on this cofactor. We newly documented the unique depletion of heme in *T. b. gambiense* cells due to the attenuated haptoglobin-hemoglobin receptor. The loss of functional heme receptor in some trypanosomes results in the reduced occurrence of transmission-competent BS-ST cells from their life cycle and further affects the resulting parasitemia in the vertebrate host.”*

*The statement “However, this disadvantage may be compensated by lower pathogenicity and prolonged chronic disease typical for *T. b. gambiense*-caused human African sleeping sickness.” falls into the hypothesis, which was not experimentally addressed in the manuscript. Therefore we removed this sentence from the summary.*

We very much hope that the rewording will now meet the requirements of this reviewer. If not, we will be happy to communicate directly with this reviewer (in whatever way the editorial office would consider appropriate) to optimize the critical sentences.

REVIEWER COMMENTS (1. st resubmission)

Reviewer #1 (Remarks to the Author):

Horakova et al present a comprehensive study demonstrating the haemoglobin uptake in African trypanosomes is mediated almost exclusively by HpHbR. Moreover, the authors also showed that this process is required for a successful developmental progression into stumpy forms, indicating that resistance to human serum is involved in a trade-off whereby HpHbR evolved to have pleiotropic effects in the acquisition of heme proteins and aerobic metabolism, as well as differentiation. The paper is written very nicely. It was a pleasure to read. The data was presented in a logical manner. Below are some comments for consideration:

general comments:

One of the central points of this paper is that *T. b. gambiense* have reduced ability to turn stumpy. The authors should include references demonstrating this reduced capacity for cyclical transmission in comparison to other trypanosome species to back up these claims. *T.b. gambiense* strains vary in terms of their ability to differentiate. I suggest the authors include a paragraph on this and refer to Janelle et al (2009).

There is a lot of circumstantial evidence for compromised transmissibility of T. b. gambiense by tsetse flies. T. b. gambiense infection rates in field-collected flies are always very low when looking into the published data, even in gambiense-HAT endemic regions. Moreover, even in optimal experimental conditions, it is challenging to obtain mature infections for T. b. gambiense in the tsetse fly (J.v.d.A; unpublished data). As suggested, we have now added a literature overview on this issue (Janelle et al., 2009; Welburn et al., 2016). Line 398-406.

The ELIANE strain also used in this paper has been passaged for many years and has essentially become monomorphic, in a similar fashion to 427 for *T. brucei*. So I was pleased to see that a second strain was used to support their findings. However, I would like to know how many passages the Bosendja field strain had undergone? How close to the field is it?

Specific information on this field strain is as follows: Bosendja - AnTAR 6; ZR/KIN001, isolated from a man, Equateur Province, Democratic Republic of Congo, 1972. It was provided to us by Nick Van Reet (Institute of Tropical Medicine, Antwerp; in acknowledgments), who keeps the strain as a stablate in rodent blood. The parasites were thawed out and injected directly into the Balb/c mice. Unfortunately, the information about how many passages the parasites went through in the animals is not available. On the other hand, they are difficult to be grown in a culture where they can potentially lose their pleiomorphic capacity. Since they were not produced as such, we can consider them as potentially pleiomorphic.

Both T.b. gambiense strains appear to cause very high parasitaemias and the experiments had to be terminated early.

However, most T. b. gambiense strains cause extremely low parasiteamias in the mouse model and in humans. I would urge the authors to be cautious about extrapolating there finding to the whole sub-species.

*With all due respect, we would rather disagree here since it is only the field strain that causes high parasitemia, but not the laboratory strain, which consistently gives very low parasitemia. Nevertheless, we hope that we are now more cautious about the formulations in the text. **Line 310-318.***

I would also like to raise the point that the authors are only considering T.b. gambiense in an animal model (i.e. in the absence of human serum). All their experiments seem valid if T.b. gambiense only takes up Hb via the HpHbR, which tryps must do in animal serum where Hp-Hb is only source. But in human serum you have the TLF1 and TLF2 complexes. While T.b. gambiense can't take up TLF-1 or Hp-Hb via the HpHbR, that still leaves uptake of TLF2 via VSG-IgM. Could the authors speculate on heme uptake via TLF2? Would this provide enough heme to allow hemoprotein activity, stumpification etc in humans?

These are elegant thoughts indeed; we much appreciate this comment. Unfortunately, we can only speculate here, using mice as a model organism for apparent reasons. TLF2 may work independently of HpHbR (Raper et al., 1996) while it has Hpr and therefore could contain hemoglobin, which may constitute an entry point of heme when T. b. gambiense grows in humans. This hypothetical TLF2-based heme uptake could help explaining why T. b. gambiense non-human and non-primate reservoirs are limited. Intriguingly, T. b. gambiense may prefer human blood for its TLF2.

If this is true it would be interesting to speculate that T.b. gambiense can infect animals but that this is an infection deadend/ not a reservoir as the lack of heme prevents them being effective in making stumpies to infect tsetse.

It may be the case, but we decided to refrain from this speculation because more data shall be collected to support such a substantial claim.

I do not necessarily agree with the conclusion that Heme uptake could have such key role in controlling chronicity in gambiense HAT. The finding reported here are certainly encouraging and must be considered when understanding dynamics of infection, but I believe more work is require clarifying whether T.b. gambiense does not develop into stumpy forms in vivo. A more thorough measurement of other stumpy markers must be included, either in the form of qRT-PCR, IFAs or more unbiased approaches such as transcriptomics to determine if this is the case.

*While we believe that the evidence we have provided in the original submission supports the (partial) loss of the capacity to form stumpies, we agree with this reviewer that more evidence is welcome. She/he suggested qRT-PCR, which we now provide for several marker genes for the stumpy form. The analysis is now part of extended Figure 5 (Suppl. Fig. 5). **Line 310-318.***

I would also like the authors to comment and reference recent studies suggesting long

slenders are able to infect tsetse flies and so the differentiation to stumpies is not essential for transmission.

Whether the short stumpy form (BS-ST) is essential for the establishment of the infection in the tsetse midgut remains controversial. Even in the older literature, this was already questioned based on experimental observations (see overview in Janelle et al., 2009). Personal experience of one of us (J. v. d. A.) is that giving the first bloodmeal to tsetse with long slender forms (BS-SL) can result in tsetse midgut infections, although at a much lower rate than an infective bloodmeal with a majority of BS-ST. This experiment was done with in vivo grown trypanosomes isolated from an infected mouse, not from in vitro culture like in Schuster et al. (A modification to the life cycle of the parasite Trypanosoma brucei. bioRxiv717975; doi: <https://doi.org/10.1101/717975> [2020]), which is highly artificial. When we added reduced L-glutathione (GSH) to bloodmeal, the midgut establishment rate raised to very high levels when infected with BS-SL. So, intrinsically it seems that BS-SL can do a short-cut transformation into the midgut procyclics (PS). However, it appears that BS-ST is pre-adapted to better survival in the hostile midgut environment (withstanding possible oxidative burst) when ingested by the tsetse fly. The addition of GSH in the first infective bloodmeal seems to suppress this hostility for BS-SL, resulting in better initial survival and higher PS midgut establishment.

*We already cite the above publication (Schuster et al., bioRxiv 2020) in our study, and we have also added several older publications, which question the necessity of BS-ST for transmissibility of T. b. gambiense. **Line 391-406.***

Specific comments

The introduction is written nicely, but I feel the outcomes can be explained in more details. In particular, in line 48, the authors could mention explicitly which these two processes are (aerobic metabolism and differentiation?).

The part of the abstract was reformulated, and the two processes are now explicitly named as suggested.

Altogether, we identify heme-deficient metabolism and disrupted cellular differentiation as two distinct HpHbR-dependent evolutionary trade-offs for T. b. gambiense human infectivity.

Line 61, the authors could include a description of TLF-2.

TLF2 is now described along with appropriate references.

When injected by the blood feeding insect vector (tsetse fly; Glossina spp.) into the human tissue, animal-infecting T. brucei brucei is rapidly killed by a potent arm of the innate immune system, represented by trypanosome lytic factors (TLF) 1 and 2 (2, 3). Both factors are high-density lipoprotein complexes containing haptoglobin-related protein (4) and apolipoprotein L-1 (5). In addition, TLF2 contains IgM molecules through which its uptake is mediated (6, 7).

Line 66, the authors could include a reference describing the intravascular haemolysis of haemoglobin.

As suggested, we have now added a relevant publication (Schaer et al., 2013) describing the intravascular haemolysis of haemoglobin. Line 68.

*Schaer, D. J., Buehler, P. W., Alayash, A. I., Belcher, J. D., & Vercellotti, G. M. Hemolysis and free hemoglobin revisited: exploring hemoglobin and heme scavengers as a novel class of therapeutic proteins. *Blood* 121(8), 1276–1284 (2013).*

Line 69, is it known whether TbHRG is developmentally regulated? If so, perhaps this can be mentioned in the text to stress that in the mammalian host, the only possible/known receptor involved in Heme uptake is HpHbR.

Yes, indeed. The TbHrg transcript is developmentally regulated during the life cycle, which is now described in the text. Line 72-74.

The TbHrg transcript is developmentally regulated with the highest expression in the PS cells residing in the tsetse posterior midgut, gradually decreasing in the subsequent life cycle stages (13).

Line 212, perhaps it would be good to cite a review here to work describing this.

A proper reference now accompanies the relevant sentence.

The BS trypanosomes undergo extensive cellular differentiation in preparation for an abrupt transmission from the mammalian blood into tsetse fly (37).

*Szöör, B., Silvester, E., & Matthews, K. R. A Leap Into the Unknown - Early Events in African Trypanosome Transmission. *Trends Parasitol.* 36(3), 266–278 (2020).*

Line 231, please add references describing this phenomenon.

“Next, the capacity to undergo differentiation in vitro was analyzed by exposing the individual cell lines to cis-aconitate and a temperature decrease to 27 °C, which is known to trigger the BS to PS transformation.”

The following references have been added as suggested: Brun et al., 1981; Czichos et al., 1986; Ziegelbauer et al., 1990.

*Brun, R., & Schönenberger, M. Stimulating effect of citrate and cis-Aconitate on the transformation of *Trypanosoma brucei* bloodstream forms to procyclic forms in vitro. *Z Parasitenkd.* 66(1), 17–24 (1981).*

*Czichos, J., Nonnengaesser, C., & Overath, P. *Trypanosoma brucei*: cis-aconitate and temperature reduction as triggers of synchronous transformation of bloodstream to procyclic trypomastigotes in vitro. *Exp parasitol.* 62(2), 283–291 (1986).*

*Ziegelbauer, K., Quinten, M., Schwarz, H., Pearson, T. W., & Overath, P. Synchronous differentiation of *Trypanosoma brucei* from bloodstream to procyclic forms in vitro. *Eur J Biochem.* 192(2), 373–378 (1990).*

Results:

Figure 1

- Can you stress that this was an in vitro assay using cultured cells?

Yes, it is an in vitro assay, but trypanosomes were isolated from mice, therefore ex vivo. To make this clear, the text has been extended as follows: "Moreover, ex vivo cells collected on day 4 from infected mice were examined morphologically and the functionality of HpHbR was verified by the uptake of the fluorescently-labeled HpHb complexes (Figs. 3C, D)".

Line 196

- Are the levels of hCAT comparable between cell lines? Based on D and E, it seems like hCAT is more abundant in *T. b. gambiense*.

*Yes, the expression of HpHbR differs to some extent in various cell lines. In any case, the higher expression in *T. b. gambiense* supports the idea that the enzyme is made but remains inactive due to the missing heme cofactor.*

Figure 2

- Although the results CYP51 KO cell line is very clear, I think the information regarding the lack of a complete mRNA knockdown using RNAi should be better documented as it might help with the narrative as well as in future studies attempting this approach.

- In line 156, the authors could include a reference to the figure they refer to when mentioning the CYP51 RNA and KO cells to ensure readability.

While revising the manuscript, we found the data generated on CYP51 RNAi redundant (due to the comparable data on CYP51 KO). Therefore, we decided to remove the CYP51 RNAi data, which helped the clarity and readability of the text.

Looks like figures 2f and 2g would benefit from including graphs similar to those in figure 2d, either as part of the main figure or as a supplementary file. Also, we believe that in addition the IC50 values reported in figure 2d should also include standard deviations and number of replicates, either in the figure itself or in the figure caption.

Figure 2 has been simplified, and the graphs have been modified according to the referee's suggestions. In Fig. 2D (now 2C) standard deviation for IC50 has been added.

Figure 4

- Figure 4b lacks phase images in the left panel

The left panel of Figure 4b has been removed since it is difficult to judge the morphology under such magnification. Moreover, the right panel provides a representative figure.

- Perhaps the authors could elaborate on the differences observed between the in situ AB vs the 18S rRNA AB?

Per request, we have now added the expression for 18S rRNA AB and the in situ AB (Suppl. Fig. 3C). As anticipated, the expression profile shows that 18S rRNA AB has a much higher (50x) expression of HpHbR than in situ AB cells.

- Are the differences observed in figure 4a and c significant? If so, statistical information should be added to figure and/or figure legend.

Statistical information has now been added to Figs. 4A and 4C.

Figure 5

- Can the authors elaborate on the seemingly contradictory results observed in figure 5a vs. figure 4f? There is clear nuclear repositioning, but the cells are not changing in volume, indicative of long slender forms.

Indeed, this is not easy to explain. Our view is that these are not identical to the prototypical stumpies of T. b. brucei. We believe that it is reasonable to assume that these T. b. gambiense forms generated under these experimental conditions have distinct features from stumpies, including the unaltered cell volume.

- Can the authors elaborate on the differences in parasitaemia observed between T. b. gambiense LiTat1.3 vs. T. b. gambiense Bosendja? One could argue that WT Tbg LiTat does indeed undergo stumpy formation, as determined by a reduction in the levels of parasitaemia >4 dpi. Moreover, in both cases, the levels of PAD1+ cells seem consistent, begging the question of whether PAD1 is the best descriptor of stumpy formation in T. b. gambiense?

We are aware of these differences between the two strains. They may be caused by subtle differences in their pathogenicity or because LiTat1.3 was routinely kept in the medium while Bosendja was passaged thru mice. Moreover, we can also not exclude the possibility that the clearance of the bloodstream parasitemia is not always linked to the stumpy formation, and other processes such as myeloid cell-based immunity may play a role.

- Given the impact of such observation, I would recommend that the authors expand on the measurement of further canonical stumpy markers, either via qRT-PCR or IFA.

We agree and we have performed further experimentation and added new data to support our claims regarding the stumpies.

Conclusions:

- More information is required on lack of stumpy formation in T. b. gambiense to sustain the claims of the paper, so I would accept this for publication if more information is included (e.g. qPCRs on candidate stumpy markers).

In response to this request, we have explored other candidate markers for stumpies, and the obtained data are now shown in the new Suppl. Fig. 5.

- How do the authors reconcile their findings with previous reports showing stumpy forms in the skin? Similarly, how do the authors reconcile their findings with the existence of endemic foci where transmission still occur?

These are exciting and relevant questions, for which answers may still not be unequivocal.

Let us first address the question regarding stumpies in the skin. Indeed, the evidence for the presence of T. brucei stumpies in the skin and their transmissibility is convincing (as reviewed in Alfituri et al. 2020). However, these studies examined mainly T. b. brucei subspecies. Capewell et al. in 2016 estimated that stumpy forms represent 20% of the skin-dwelling population. Interestingly, they also evaluated T. b. gambiense PA strain and found out that in some cases, it has even higher skin parasitemia than T.b. brucei control. The histological sections showed stained parasites most likely of BS-LS morphology. A recent study in a gHAT focus in the Republic of Guinea reported the presence of extravascular skin-dwelling T. b. gambiense parasites in HAT-seropositive humans. Still, their morphology was not further examined (Camara et al., 2020). Altogether it seems that T. brucei cells can be sequestered in extravascular tissues irrespective of the bloodstream stage, but there are still many open questions worth exploring.

Line 360-362

The reasons behind T. b. gambiense disappearance from the bloodstream is not entirely clear, but they were reported to invade other tissues even more efficiently than T. b. brucei (55, 56).

Büscher, P. et al. Do cryptic reservoirs threaten gambiense-sleeping sickness elimination? Trends Parasitol. 34, 197-207 (2018).

Alfituri, O. A., Quintana, J. F., MacLeod, A., Garside, P., Benson, R. A., Brewer, J. M., Mabbott, N. A., Morrison, L. J., & Capewell, P. To the Skin and Beyond: The Immune Response to African Trypanosomes as They Enter and Exit the Vertebrate Host. Front Immunol. 11, 1250 (2020).

Regarding the second question. As highlighted recently in a review by Alfituri et al. (2020), the number of humans infected by HAT decreases, and WHO aims to interrupt transmission by 2030. The occurrence of latent human infections (asymptomatic individuals but with possible skin-dwelling parasites) may challenge these goals. Although experimental tsetse fly infection studies with T. b. brucei revealed that skin-residing (stumpy) parasites could infect tsetse flies when ingested by the fly, the jury is still out whether this finding can be extrapolated to T. b. gambiense. It is puzzling where and how the tsetse flies get infected with the HAT-parasite in an endemic region and sustain transmission.

We want to point out that T. b. gambiense is less prone than T. b. brucei to make stumpies under the conditions used. It remains possible that T. b. gambiense could form stumpies in other anatomical sites or within alternative species.

Line 103, 449, Were the heme assays performed on cultured T.b. gambiense in the absence of human serum?

The assay was performed on the ex vivo cells isolated from mice.

Reviewer #2 (Remarks to the Author):

This manuscript analyzes the capacity for heme uptake in *Trypanosoma brucei* and human infective *Trypanosoma gambiense* and links expression of the HpHbR to this. The authors further demonstrate that the lack of a heme-dependent enzyme renders parasites less sensitive to an inhibitor of that enzyme, which can be used as an assay for heme uptake and functional delivery. Finally, the authors observe that there are changes in the capacity for the development of parasites to the stumpy stage (BS-ST) when HpHbR is deleted, and in *T. b. gambiense* (which has a naturally mutant HpHbR), and suggest the developmental capacity is restored, although incompletely, with reintroduction of a *T. brucei* HpHbR.

The heme uptake assays are clear (although this part is already known I think) and the assays of Cyp51 are consistent with the overall thesis of the study- i.e. that HpHbR allows heme uptake and supply to enzymes such as CYP51. However, for this reviewer there are quite a few experiments required to substantiate the link between heme uptake and the developmental capacity of *T. b. gambiense* and/or heme uptake capacity. The interpretation of the current experiments also needs clarification.

1. The experiments presented in Figure 1 appear clear, although I think the main conclusion is already known.

We dare to disagree. The finding that *T. b. gambiense* contains a negligible amount of heme is novel, and also the monitoring of the activities of heme-carrying proteins. We want to stress that the reported cell-associated heme in wild types (Vanhollebeke et al., Science 2008) could trivially represent endocytic cargo, and Fig. 1 and Fig. 2 are the first direct evidence for heme-based metabolism in the bloodstreams.

2. On Figure 2 the effects on cell growth caused by ketoconazole would be better normalized for each cell line to their own growth in the absence of drug (rather than to the wild type cells in the absence of drug).

All graphs in Fig. 2 have been modified according to reviewer's suggestions.

3. Figure 3a measures luciferase activity, which many not reflect mRNA (or even protein) levels. The result seems OK, but levels of Luc determined by qRT-PCR or western blot might be valuable.

Quantification of Luciferase mRNA has been added as Suppl. Fig. 2B.

4. Figure 4e shows large variation based on the error bars. With mouse infections this is normal, but it is valuable to show the profile of the infection in each mouse to assess if the higher or lower average parasitaemia is a consequence of only one or two infections being atypical.'

In compliance with this request, we have now added the information on individual infections (see Suppl. Fig. 4).

5. In the work presented in figure 4 I consider it important to add back the T.b . gambiense HpHbR to the T. brucei KO cells. This should not restore developmental capacity, unlike TbHpHbR. Otherwise there is the risk that the developmental phenotype caused by TbHpHbR is simply a consequence of heterologous receptor/protein expression unrelated to receptor function. This is relevant because cells experiencing unphysiological conditions can show some progression to BS-ST characteristics. It would also be interesting to see the phenotypes generated by either TbHpHbR or TgHpHbR expressed from the 18s rRNA locus, since the expression from the in situ locus is weak compared to wild type. If it is as strong as in wild type cells, then much more obvious developmental phenotypes should be seen if TbHpHbR expression is limiting (and little would be seen with Tg HpHbR expression, again confirming the importance of receptor function).

We thank this reviewer for this important control. We decided to perform the technically non-trivial experiments suggested by this reviewer. Hence, we have generated new strains for this aim. The HpHbR KO expressing T. b. gambiense HpHbR from the 18S rRNA locus showed a similar expression to its T. b. brucei variant (see new Suppl. Fig. 3). This allowed us to compare the two strains in vivo along with their resulting phenotypes. The new add back cell line showed a lower ability to rescue the HpHbR KO phenotype than the original T. b. brucei variant (new Suppl. Fig. 4). This confirms the importance of a fully functional receptor for stumpy formation.

6. Conversely, in Figure 5 I consider it would be important to 'add back' T. b. gambiense HpHbR expression in WT T. b. gambiense from the 18S rRNA locus to match the expression of the T. brucei add-back in that cell line. If the story is correct, the TgHpHbR should not drive any developmental response/PAD1 expression where the TbHpHbR does. This would confirm the growth effect and possible development seen with the expression of the T. brucei protein is due to its receptor function under conditions of equivalent protein expression.

Following this suggestion, we generated a new cell line of T. b. gambiense overexpressing T. b. gambiense HpHbR and scrutinized it in vivo (see Fig. 5). We found this cell line to give the same parasitemia in the mouse model (shallow parasitic wave peaking at day 4) as the T. b. brucei variant. However, strikingly, there was no increase of the PAD1-positive cells observed for the newly generated strain, as reported for the T. b. brucei variant, further supporting our hypothesis about the ability of the T. b. brucei receptor to enhance the stumpy formation.

7. Following on from point 6, what is the level of the expression of the add back T. brucei HpHbR in the T. b. gambiense cells? The restoration of developmental competence is quite subtle (notwithstanding the growth effect in vivo- the growth in vitro is not presented but would be of interest) : morphologically BS-ST cells are not clearly generated (by cell volume) and PAD1 expression is very limited. If expression of the T brucei protein were efficient I would have expected a better level of development if this receptor is an important contributor to the developmental response. The cell cycle status of cells expressing the TbHpHbR should be presented in addition to their growth also- since development to BS-ST should show an increase in cell cycle arrested forms (1K1N).

We now newly show a relative expression of HpHbR in all addback cell lines generated in the study (see Suppl. Fig. 3C).

8. If heme levels are important in the process of development, the capacity of *T. brucei* and *T. gambiense* parasites to express PAD1 and develop to BS-ST forms could be explored in vitro using heme deficient or haptoglobin depleted medium, or in media with excess heme. These assays would allow quantitative and titratable effects to be observed to link uptake efficacy and development and are tractable for monitoring BS-SL and BS-ST expression in vitro.

These are undoubtedly relevant experiments to address the direct role of heme in cell differentiation which we want to study in more detail in the follow-up study. For now, we softened our wording about this possibility.

Line 367-369

Thus, we propose that the excess of free hemoglobin released by trypanosome infection possibly modulates heme uptake and, subsequently, parasitic waves.

Line 372-374

In the absence of HpHbR, the key BS-SL to BS-ST transition is disrupted, prompting us to suggest that heme uptake may be an additional player in this life cycle progression.

Overall, I am convinced by the data on heme uptake and the importance of the HpHbR in this process- this is a valuable finding (although some of it is known). I remain to be convinced that there is a consistent and causal relationship between HpHbR and developmental capacity in *T. b. gambiense*. As the authors state (line 369-370), different isolates of *T. b. gambiense* show different levels of slender and stumpy formation in different field isolates, although presumably all have HpHbR mutations to permit human infectivity. Hence, evidence that there is a robust relationship between these two observations (HpHbR function and developmental capacity) is not yet established – the *T. b. gambiense* lines used in their study are of uncertain passage history in mice; other field isolates may behave differently.

*We newly added one more *T.b. gambiense* strain (PA, see Suppl. Fig. 5) which indeed behaved differently in terms of stumpy marker genes expression, and we reformulated our text accordingly.*

Line 310-318.

I also found the interpretation in the discussion to be quite muddy. On one hand, evidence is cited that *T. b. gambiense* is transmitted by flies poorly, which would be consistent with a proposed link between HpHbR efficacy and development. On the other hand, an unpublished BioRxiv report is cited of equivalent fly infectivity for slender and stumpy forms (albeit using unusually permissive flies) and used to suggest that the developmental capacity is not evolutionary constraint for *T. b. gambiense*. These arguments seem contradictory.

*The low ‘intrinsic’ transmissibility of *T. brucei gambiense* for tsetse flies has been reported in the literature. It is based on data from field-collected flies in HAT-endemic regions and experimental infection studies. Notably, reported low transmissibility is mainly due to a low success in developing a procyclic midgut infection into final metacyclic salivary gland infections. Still, tsetse flies can be infected with these parasites, and transmission of *gambiense*-HAT occurs solely through these flies.*

*Indeed, Schuster et al. (preprint bioRxiv717975) reported the ability of slender *T. b. brucei* to infect tsetse flies, with comparable infection rates as obtained for the short stumpy forms. The reviewer correctly noted that this experiment seems to be performed in strong*

permissive flies generating high infection rates for both trypanosome forms, including the long slenders. The ability of T. b. brucei long slender bloodstream forms to transform into a procyclic infection in the tsetse midgut has already been reported in the older literature and it was recently confirmed by us (JVDA, personal communication). However, these experiments always revealed a strong and significant difference in the resulting midgut infection rates, with T. b. brucei long slender proliferative forms giving rise to only a few midgut infected flies, if any. Specifically, 3% (4/134) versus 25,2% (37/134) midgut infected flies after an infective blood meal with T. b. brucei AnTARI long slender (morphologically 100% slenders) and short stumpy trypanosomes (> 60% short stumpy morphology), collected from infected mice at a different stage of the parasitemia. These data confirm the intrinsic biological differences between those two bloodstream stages in their ability to survive and/or transform in the tsetse midgut into an established procyclic infection. Interestingly, the very low transmissibility of the long slender forms could be counteracted by adding 10 mM reduced L-glutathione to the infective blood meal (raising the midgut infection rates to > 70% for the long slenders and 86% for the short stumpies). This suggests that the addition of this antioxidant makes the midgut more permissive (less hostile) for the ingested bloodstream forms, especially for the long slenders. The 'contradictory outcome' on the low tsetse transmissibility of long slender T. b. brucei bloodstream forms reported in the literature and confirmed by our experiments and the data of Schuster et al. could be possibly linked to their different experimental conditions, i.e. tsetse flies with a higher midgut permissivity for bloodstream forms, the use of in vitro cultured /collected long slender & short stumpies, instead of the in vivo grown ones in previous/our experiment, among maybe other experimental differences (e.g. blood source to feed the flies) that could have modulated the trypanosome permissiveness in the fly midgut.

We have tried to incorporate this valuable information into the manuscript (where appropriate) to indicate the existing controversy.

Line 391-406.

Finally, I assume that the heme dependent response is secondary to density dependent stumpy formation by quorum sensing (the dominant mechanism driving transmissibility)? Since heme availability and uptake would likely not be density dependent (nor limiting in vitro), it seems the developmental effects (if validated) would be a consequence of a heme-dependent processes in the BS-ST pathway downstream of the initial stimulation. This can make sense and would not be surprising given the mitochondrial involvement and metabolic adaptations inherent in BS-ST development. However, without some further experimentation and validation I left uncertain of the importance or mechanism of the developmental aspect of the work, despite interest in the observations reported.

The density-dependent stumpy formation is undoubtedly a well-documented mechanism. We have the data showing in vitro differentiation with cis-aconitate, indicating that heme would act downstream of that process. Moreover, we see the crosstalk between the stumpy formation and heme uptake (via HpHbR), a connection not noticed before.

Minor issues

1. Figure 1d would benefit from molecular weight size markers (the relevant bands are labelled but the size would be good to add).

As requested, the markers have been added.

2. Line 125, referring to figure 2A, says that CYP51 is differentially transcribed in procyclic, slender and stumpy forms. Since most regulation is posttranscriptional, it is more likely to have differential mRNA abundance but not be differentially transcribed. This should be amended.

*Indeed, it has now been rephrased as follows: "The genome of *T. brucei* s. l. encodes a single copy of CYP51, with the highest mRNA abundance in PS compared to BS-SL and BS-ST cells with relatively low levels (Fig. 2A)."*

Line 128-130.

3. On Figure 2f and g the Y axis is broken to stretch out the region between 50-100% and to compress 0-40%. I don't think this is necessary and is potentially misrepresentative of the scale of effects seen.

The graphs in Fig. 2 have been modified to take into account all referee's suggestions.

4. Line 181 refers to the PS to BS differentiation, but the differentiation is actually BS to PS. Please correct.

Has been corrected.

5. Line 240 'bulky HpHbR expression'. I don't think bulky is quite the right word. There are several other places where the phraseology or English usage would benefit from attention.

We replaced the "bulky" for "robust" in the text.

Reviewer #3 (Remarks to the Author):

The human infectivity of Tbg is, unlike Tbr, multifactorial. Decreased functionality and decreased expression of the HpHbR is known to be one of the evolutionary modifications that have allowed Tbg to infect humans, through reduced uptake of TLF1. Here, the authors main claims are: (1) That both Tbb HpHbR KO and Tbg cells lack hemoprotein activity and therefore proliferate in the absence of heme. (2) That HpHbR KO prevents differentiation of the parasites into stumpy forms. (3) That Tbg is poorly competent to differentiate into stumpy forms, due to reduced functionality of HpHbR

This is an interesting and timely study that would certainly contribute to the field and raise questions for further study. There are some revisions that are required in order to support the claims presented and the text needs to be clarified to state only what is experimentally shown.

Main claim 1: That both Tbb HpHbR KO and Tbg cells lack hemoprotein activity and therefore proliferate in the absence of heme.

The Tbb HpHbR KO cell line has previously been generated and characterised for its inability to internalise HpHb or free heme, despite little growth effect in vitro (Vanhollebeke et al.,

2008). It was also previously stated in the same work, that: "The steady-state heme content of WT cells isolated from mice was 2.3 ng/mg of protein, whereas heme was undetectable in KO cells" and that "heme appeared to be mostly incorporated into hemoproteins" which would together then strongly support that the Tbb HpHbR KO cell line should indeed have an absence of hemoprotein activity and proliferate in the absence of heme. Here, this previous data is confirmed with further experimental support. The measurement of the intracellular heme content of Tbg is novel and of significant interest, given the previously observed mutations and changes in gene expression of the TbgHpHbR. The authors also assess, for the first time, hemoprotein activity between these cell lines via introduction of a human hemoprotein in Tbb WT, Tbb HpHbR KO and Tbg cells and via endogenous CY51 hemoprotein activity in Tbb WT, Tbb HpHbR KO and Tbr cells.

Heme measurements are conducted and quantified, with measurements for TbbHpHbR KO and Tbg close to the limits of detection and with no significant difference detected between them. That being said, the chromatogram from Fig 1a and quantification from Fig 1b, does imply that Tbg has some, albeit very little, internal heme.

We would question the reviewer's assumption that the heme is internal. It could be sticky heme to the coat, secondary to the isolation process.

The addition of human catalase into Tbb, TbbHpHbR KO and Tbg cells was a good experiment as this allows the cell lines to be compared directly for hemoprotein activity without consideration for other mutations or changes in gene expression that may be distinct between the cell lines. This data demonstrates that the acquisition of heme in Tbg does not support any significant activity of human catalase, although again, the data does imply a very mild activity as the shape of the measurement curve (Fig 1f) does have a small jump at the time of H₂O₂ addition, unlike the TbbHpHbR KO cells.

Given the high level of intracellular heme in the Tbb WT cells and the trace amount in Tbg cells, it could be argued that the amount of heme present in Tbg is enough for some activity of endogenous hemoproteins but is not in excess and therefore unable to fuel the human catalase. The authors themselves describe this as 'potent heme-dependent activity [through overexpression]' (line 319). For this reason, it would be particularly valuable for the authors to assess the endogenous hemoproteins of Tbg, as they have done for Tbb WT and TbbHpHbR KO cells in Figure 2. In fact, it is very unclear as to why the authors have included a ketoconazole sensitivity experiment for Tbr (Figure 2G), when they have not included this for Tbg. This should be added and would considerably strengthen the manuscript.

Also, with Tbr not particularly the focus of this work, it is not clear why the experiment in Fig 2G was conducted on Tbr at all, rather than the Tbb WT and Tbb HpHbR KO cell line.

We want to add that the experiment was meant to validate the HpHbR requirement through its cognate ligand, the HpHb complex. Therefore Hp-deficient serum has to be used, either Hp^{-/-} mice serum or Hp^{-/-} human serum on cells with the fully functional receptor. T. b. rhodesiense grows well in human serum (hence this choice), while growing T. b. brucei in vitro in mouse serum has always proven difficult (and obviously, T. b. brucei cannot be grown in human serum due to the lysis by TLF1).

Throughout, the manuscript text could be clarified to ensure that the authors are clear on the conclusion that they are presenting regarding the TbgHpHbR expression / function based on their data and previously published work. Absence is not the same as a little and poorly

capable is not incapable. The data in this manuscript and in published literature fully supports a decreased functionality (i.e. reduced binding capacity for HpHb) and decreased expression of the TbgHpHbR (mRNA and protein). It does not support a complete loss of the TbgHpHbR. Perhaps the authors would like to comment or speculate on the observation that there has not been a complete loss either via expression or a true loss-of-function mutation. Would this imply that there is indeed some residual use of the receptor and/or benefit to maintain it?

Thank you for the valuable comment. We tried our best to be more explicit with our wording. We agree that the gene will probably be eventually lost if not used/needed throughout evolution. We have to consider both the possibility that the time frame was too short for its loss until the present or that it was kept for another function.

Main claim 2: That HpHbR KO prevents differentiation of the parasites into stumpy forms.

HpHbR is downregulated upon transition of BS-SL cells to BS-ST cells. The authors investigate if this due to position on the polycistronic unit, which it is not. This section seems to be addressing a rather different hypothesis to the focus of the rest of the manuscript (mechanisms of gene expression during differentiation rather than evolutionary trade-offs of reduction of heme uptake) and could be moved to supplementary.

We agree, and Figs. 3A and B have been moved to the supplementary material (now Suppl. Fig. 2C, D).

The authors also show that the presence of HpHbR in BS-ST forms does not in any way hinder the transition to BS-ST or PS forms. The authors could more clearly indicate what the hypothesis for this experiment was; was it that there may be a selective downregulation of the HpHbR at this stage as opposed to being subject to the general translational repression observed in BS-ST cells (Brecht and Parsons 1998)? Upon expression of the HpHbR in BS-ST forms, the authors state that the cells "...still differentiate into the BS-ST and subsequent PS and retain the ability to infect tsetse flies" (line 208). This language implies that they anticipated a hinderance.

Indeed, Fig. 3C-G (now Fig. 3A-E) shows a gain of function approach. One could think that the continuous import of heme into cell cycle-arrested stumpies could be detrimental. More so, the abrupt reduction of heme influx could have been a mechanism triggering differentiation into stumpies. These experiments exclude these hypotheses, which are now better explained in the text.

Line189-191.

To the contrary, given that they later state that presence of heme is required for differentiation of the parasites into stumpy, did they consider if the extended expression of HpHbR may actually benefit this transition? As shown in Figure 3d, the HpHbR-3'PAD1 cell line, as an average, peaks a full day earlier than the WT and at a moderately lower density which would be consistent with an earlier transition to BS-STs. Although I can see that the error bars on parasite load would likely overlap, the trend on rising and falling parasitaemia seems clear. Is this a reproducible difference between individual mice and/or different experiments? Was this ever studied with 12 hour time points between day 3-6? If the BS-ST forms or PC forms have a greater reliance on hemoproteins, then perhaps the authors might consider if the extended

expression of the HpHbR in this case increases heme availability/hemoprotein activity during this transition.

Thank you for this comment; this is an exciting thought. On the other hand, we don't think that HpHbR-3'PAD1 cells are significantly different from WTs. The same WT cells (90-13) were also used in Fig. 4E where they peak later (day 5-6).

A pleomorphic HpHbR KO and two add-back cell lines (18S and in situ) are produced; no validation of the add-back cell lines is included and the authors should therefore add this to Supp Fig 2 (genetic validation or assessment of expression levels). These cell lines are assessed functionally through the number of Hp+ cells after exposure to labelled Hp. The WT cells have a value of 80% with the add-backs providing 70 or 20%. Could a low number of positive cells in the in situ add-back also represent a non-clonal culture where not all cells are expressing the gene?

The qPCR analysis of HpHbR expression for all variants of the add back cell lines has been performed and included (Suppl. Fig. 3C, D). Although we cannot exclude the non-uniformity of the population, all the cell lines were prepared consistently by limiting dilution.

At this stage, the authors have sought to "...test whether heme and/or HpHbR play any role in [BS-ST formation]" (line 217). The data collectively presented in Figure 4 convincingly demonstrates that the TbHpHbR KO cell line does not form BS-STs and that this is due to the absence of the HpHbR. But this does not differentiate between heme and/or HpHbR and it is not clear from the authors discussion what their conclusion for the molecular basis for this is. They state "The fact that in the absence of HpHbR, the key BS-SL to BS-ST transition is disrupted, prompts us to suggest that heme uptake may be an additional player in this life cycle progression" (line 340), and that "...HpHbR facilitates the developmental progression by inducing PAD-1 expression" (line 44). Both of these statements are implicating the HpHbR itself in developmental regulation and/or signalling. However this is distinct from a more likely hypothesis that BS-ST forms simply have at least one required hemoprotein and therefore unlike BS-SL forms, BS-ST forms do require heme. It would be beneficial to this work if the authors did experimentally address this distinction, by expressing the TbHrg in the TbHpHbR KO or assessing hemoprotein activity in BS-ST, for example. This also has implications for claim 3.

Those are interesting and stimulating proposals, but we believe they are out of the scope of this study. We know that TbHrg is indeed upregulated in BS-ST compared to BS-LS cells from the qPCR analysis we performed here (see Suppl. Fig. 5). It is well documented for stumpies that they have more branched mitochondrion with respiratory complexes present, demanding heme (Capewell et al., 2013). So, yes, we agree, stumpies probably need heme, but HpHbR does not provide it anymore.

Main claim 3: That Tbg is poorly competent to differentiate into stumpy forms, due to reduced functionality of HpHbR.

The finding that Tbg produces little/no BS-S Relative *HpHbR* mRNA vel and of interest to the field. However, while the authors have clearly uncovered some interesting biology here, the data presented does not satisfactorily support the general claim that Tbg is poorly

competent to produce BS-ST. It is very possible that, in evolving the adjust to the modifications of the reduced functionality of the TbHpHbR, that Tbg BS-ST are not identical to Tbb BS-ST (perhaps reducing reliance on hemoproteins). This would be very interesting and authors have clearly considered this possibility as they state "WT Tbg has a poor capability to generate typical BS-ST" (line 343). The data presented should be published and is of interest, but I do not think it can be summarised as demonstrating that "Tbg is poorly competent to differentiate into stumpy forms".

Figure 5a-d clearly demonstrates that very few PAD1+ cells/ morphologically BS-ST cells are identified in the Tbg WT and this is moderately recovered with add-back of the TbHpHbR. However, the growth curves from the mouse experiment indicates that the differences in dynamic here may be more complex. The authors have shown that an absence of HpHbR for Tbb results in an outgrowth of the parasites which would ultimately overwhelm the host (Figure 4e). This uncontrolled growth is typical of a 'monomorphic' cell line which does not produce BS-ST. This is not what is observed with the Tbg WT, where the parasite numbers do fall on day 5. Indeed, it is known that Tbg causes a chronic infection in humans and can result in asymptomatic infections, whereas an absence of BS-ST forms in Tbb (as per Fig 4e) is shown to cause uncontrolled growth. This again supports a hypothesis that there is a distinct situation in Tbg, but not simply through a lack of BS-ST forms.

The addition of the TbHpHbR to Tbg very much suppressed parasitaemia which, based on cell number alone, might indicate a dramatic shift towards differentiation to BS-ST, but this is not reflected in the PAD1+ assay, where only around 30% of the cells are positive. A straightforward additional experiment of interest here would be to assess the parasites on day 4 for their cell cycle status to determine if they are or are not undergoing a cell cycle arrest. Further, a consequence of loss of BS-ST formation for the TbbHpHbR KO was an additional reduced capacity to generate PS, which was not assessed for Tbg. Ideally it should be, since the authors do discuss a reduced capacity to infect flies in their discussion (Line 364) and it would support their hypothesis if they saw increased fly infectivity upon TbHpHbR overexpression in Tbg.

We newly generated the cell line overexpressing TgHpHbR in Litat 1.3 strain, and in vivo the parasitemia was also suppressed similarly to TbHpHbR (see new Fig. 5A), but independently of stumpy forms which were not detected (Fig. 5C). To address T.b. gambiense differentiation in vitro is not that straightforward. Litat 1.3 is the only T.b. gambiense strain that can grow in the culture. Still, after so many years in the laboratory conditions, it is essentially a monomorphic strain that lost the capacity to differentiate into PS.

The data presented in Figure 5e-h regarding the Tbg Bosendja Field strain was generated to address if this strain "... sustains the ability to produce waves of parasitemia". This is not addressed as the experiment is terminated on day 4. It would have hugely benefited from being continued to day 5. The parasitaemia in the mice could have dropped as for the Tbg WT (Fig 5a) or continued to rise as for the Tbb HpHbR KO (Fig 4e) and so no conclusions can be drawn either way.

We tried to extend the experiment up to day 5 for Bosendja strain, but unfortunately, the mice did not survive, most likely due to elevated parasitemia.

My final concern with the statement that Tbg is poorly competent to generate BS-ST is that

"... publications describe a high variability in the proportion of the BS-SL to BS-ST cells in different field strains of Tbg" (line 369) and we know that different field isolates have different levels of HpHbR expression as well (Kieft et al., 2010). The data presented should be considered more broadly within the context of the variability of Tbg isolates, which may actually support the authors hypothesis, if the isolates with higher levels of reported BS-ST forms had correlatively higher levels of HpHbR expression.

We are newly showing the transcription profiles of genes upregulated or downregulated in the BS-ST forms in different T.b. gambiense strains (see Suppl. Fig. 5). We must admit that the picture is rather complex since even cells from the same group (T.b. gambiense Group 1) express various levels of the marker genes. We are therefore more cautious about the extrapolation of our finding to the whole T.b. gambiense group. We also checked the level of HpHbR expression, and it is not in line with the published data (Kieft et al., 2010). Unlike in their study, here we report higher HpHbR transcripts in T.b. gambiense cells than in T.b. brucei controls. On the other hand, we have to consider that the ex vivo cells may be isolated from the host at different time points, which may cause a significant difference in HpHbR expression. Moreover, since BS-ST cells do not produce HpHbR anymore, the hypothesis may be formulated the other way around, meaning more HpHbR less BS-ST and vice versa.

Text

There are a few statements that are either incorrect or written in such a way as the scientific interpretation is incorrect or ambiguous. These should be modified to ensure accurate reflection of the literature or point being made.

Line 36: "... the physiological function of the receptor remains to be elucidated". Do the authors mean the function is yet to be elucidated in Tbg specifically? Because the function of the HpHbR as a receptor for HpHb in Tb is already known (Vanhollebeke et al., 2008; Stødkilde et al., 2014).

The fact that HpHbR binds HpHb was known. Why and when in the life cycle this is important to the parasites is novel, though. Moreover, what was not studied in detail is where the heme is needed as a cofactor in the cell. Therefore, we hope we addressed the physiological function by scrutinizing the genuine hemoproteins such as CYP51 in bloodstream cells lacking the HpHbR.

Line 44: "... HpHbR facilitates the developmental progression by inducing PAD-1 expression..." The authors do not show that HpHbR induces PAD1 expression.

We agree. The sentence has now been corrected as follows:

"We further show that HpHbR facilitates the developmental progression to cell cycle-arrested stumpy forms in T. b. brucei."

Line 63: "...HpHbR, the only invariant cell surface receptor known to date in kinetoplastid parasites". T. brucei Factor H receptor. (Macleod et al., 2020).

Thank you very much for this addition, as we overlooked this recent publication at the time of the writing. We now refer to it and reformulate the sentence as follows: "Hemoglobin

receptor (HpHbR), one of the few cell surface receptors known to date in kinetoplastid parasites (Vanhollebeke, B. et al., 2008; Stødkilde et al. 2014, Macleod et al., 2020)."

Lines 79-82: "Retaining the HpHbR expression contribute to trypanosomes fitness in their animal reservoir hosts, providing positive selection pressures for the conservation of this receptor ". Are the authors referring to Tbg or non-Tbg Tb species? It is not clear if the point being made is that Tbg have retained the HpHbR and this contributes to fitness in animals or that there is advantage to non-gambienese Tb, which Tbg no longer have.

Here we meant fitness of T. b. brucei specifically. Therefore, we have now rephrased the text to make it more clear as follows: "Retaining the HpHbR expression contributes to T. b. brucei fitness in their animal reservoir hosts, providing positive selection pressures for the conservation of this receptor (6, 14)."

Line 82-85: Reference(s) required.

The reference has been added (Welburn et al., 2016).

Line 174-176: Reference or names of 'other genes' and/or data required. By 'involved' do the authors mean differentially regulated (of which there are very many genes) or do they mean playing an active role in the transition?

As suggested here, this data (former Fig. 3A-C) has been moved to the supplementary section (now Suppl. Fig. 2B-D). We simplified this part in the text accordingly, and we are now mentioning specifically only the HpHbR gene.

Line 238-240 and lines 262-264: Can the authors clarify the justification here. What do they mean by bulky? Expression levels have not been shown in this manuscript. Further, the data in Figure 4a indicates that the 18S add back is more similar to the WT and the authors proceed to use the 18S locus for the Tbg add back.

We are not using "bulky expression" in the text anymore, but "robust expression" to describe the expression from the 18S rRNA locus. Additionally, we newly show the level of expression for all the add backs used in this study and its comparison to the respective wild types (see Suppl. Fig. 3C, D). In T. b. gambiense add back we took advantage of the cell line T. b. gambiense +b1 (newly named T. b. gambiense + b^{18S}), which some of us previously generated (Uzureau et al., 2013).

Line 260: The title of this section and Figure 5 are "Restoration of stumpy formation in Tbg". This is not ideal, as it indicates that the authors have already shown Tbg do not form BS-ST. "Overexpression of TbbHpHbR in Tbg facilitates increased BS-ST formation" or similar, would be more suitable.

Thank you for the helpful comment. The section is now named as follows: "Overexpression of T. b. brucei HpHbR in T. b. gambiense increase stumpy formation".

Line 267: "... on day 6..." should be on day 5.

Has been corrected as suggested.

Line 271: "did not detect any PAD1-expressing cells". Data in 5c does have some indication of a very small number of PAD1+ cells.

The text has been reformulated as follows: "In the WT T. b. gambiense, we detect only a negligible amount of PAD1-expressing cells."

Lines 323-329: The authors discuss the anaemia of animal trypanosomiasis and then comment on the lack of such observations in human Tbg studies. Are the authors implying a cause-and-effect relationship and in which case, which direction? Do Tbg have less use for a HpHbR because there is less free HpHb, which is not necessarily logical because reduced concentration of ligand in blood could select for greater binding capacity in order to compete with host macrophages.

No, we do not imply a direct cause-and-effect relationship between the anemia and T.b. gambiense HpHbR. We assign the lack of anemia in T.b. gambiense to the nature of the disease with a chronic course of infection.

Lines 330-336: Please clarify this hypothesis, particularly line 334-336.

We agree that the statement following the references to what is known is rather vague. It has been reformulated as follows: "Thus, we propose that the excess of free hemoglobin released by trypanosome infection possibly modulates heme uptake and, subsequently, parasitic waves."

Line 357-359: "... the lower pathogenicity of the HpHbR KO T. b. brucei for both the Hp-carrying and Hp-lacking mice (Vanhollebeke et al., 2008)". This is not what the data in this manuscript supports, which demonstrates increased pathogenicity after day 6 (Fig4e).

We have to stress that the older publication (Vahollebeke et al., 2008) used monomorphic HpHbRKO strain in contrast to this study where we newly generated the HpHbR KO in the pleomorphic strain. Both studies refer to the earlier parasitemia (day 4 and 5 p.i.) in mice where the HpHbR KO is attenuated compared to WT strain. We attribute the later (day 6 p.i.) increased pathogenicity to the lack of stumpy forms and not to the fitness of the cells.

Lines 367-369: "To the best of our knowledge, the reporting of putative BS-ST cells in T. b. gambiense was based solely on morphological criteria and may have resulted in a misassignment to a different T. brucei sub-species that is more prone to BS-ST transition" Are the authors stating they believe that publications 50, 51, 52 and 53 have all incorrectly utilised Tbb instead of Tbg in their studies?

No, we certainly do not claim that those publications incorrectly utilized the T. brucei subspecies; we raise the possibility that this might have happened in some cases, especially

in the older publications. However, to prevent any misjudgment from our side, we removed the statement about the misassignment to different T. brucei subspecies.

Figures

Figure 2b: mRNA levels of CYP51 knock down should be shown in supplementary data to allow the reader to assess extent of knock-down.

BF CYP51 RNAi has been removed; only the CYP51 KO is now shown to streamline the manuscript.

Figure 2d: I question the normalisation of the CYP51 RNAi +dox to the growth of the CYP51 RNAi –dox and the CYP51 KO to the WT in this experiment because these cell lines already have a difference in growth rate and so at 0uM ketoconazole the growth over 48 hours would be reduced. This makes interpretation of the data somewhat problematic. The authors could replot the data with each cell normalised to its own growth rate to determine if this improves clarity. Particularly, it should then be clearer to observe the presence vs absence of the biphasic curve. This is simply a suggestion to try and improve visualisation of this data. Without being able to see what that would look like, I do appreciate that upon doing that it may not improve the figure. In either case, the area of the figure that is of most interest is the range of 1-10uM (given this is the range then used in Figure 2f and 2g, as well as the area required on the WT curves to observe the biphasic nature of the curve). This is not particularly clear in the figure and this should be modified where possible to improve this (data point size reduced, possibly open boxes for the CYP51 KD / KO).

Fig. 2D (now 2C) has been modified with normalized growth for WT and CYP51 KO.

Figure 3d: A supplementary file showing individual mouse infection profiles would be beneficial here.

As suggested, individual mouse infections are now shown as Suppl. Fig. 4.

Figure 4c: The Y-axis is labelled procyclics. Is this accurate in that only 'procyclic' cells would be counted?

We are pretty confident that only the procyclic stage was counted based on morphology. Nevertheless, the cells were not labeled with procyclin marker in this particular experiment. Therefore, we decided to relabel the Y-axis as Cells.

Figure 4c and 4d: No error bars are provided, although replicate numbers are given.

The error bars have now been provided.

Figure 4f and 4h: Given that the PAD1+ cells have a greater cell area and shorter N to k distance, the calculation of the mean values in the whole populations and then use of these to measure significant difference seems inappropriate. The data in Figure 4h should be provided before 4f and used for statistical analysis. The data in 4f should then be separated such that

there are six sets of data per graph (WT PAD1-, WT PAD1+ and so on). This would exclude the ability to compare data in 4f with statistics, but would clearly demonstrate the point being made.

We partially modified this figure by reshuffling the individual section as suggested. Nevertheless, we believe that splitting the PAD1 positive and PAD1 negative cells would be somewhat confusing.

Definition of error bars is in the methods rather than in each of the figure legends. Multiple cases of the number of replicates not being included in Figure legends (i.e. all of the figure 2 panels, Figure 4e).

The number of replicates and statistics are now given in each Figure legend.

References:

*Alfituri, O. A., Quintana, J. F., MacLeod, A., Garside, P., Benson, R. A., Brewer, J. M., Mabbott, N. A., Morrison, L. J., & Capewell, P. To the Skin and Beyond: The Immune Response to African Trypanosomes as They Enter and Exit the Vertebrate Host. *Front Immunol.* 11, 1250 (2020).*

*Camara, M., Soumah, A. M., Ilboudo, H., Travaillé, C., Clucas, C., Cooper, A., Kuispond Swar, N. R., Camara, O., Sadissou, I., Calvo Alvarez, E., Crouzols, A., Bart, J. M., Jamonneau, V., Camara, M., MacLeod, A., Bucheton, B., & Rotureau, B. Extravascular Dermal Trypanosomes in Suspected and Confirmed Cases of gambiense Human African Trypanosomiasis. *Clin Infect Dis.* 73(1), 12–20 (2021).*

*Capewell et al. Regulation of Trypanosoma brucei total and polysomal mRNA during development within its mammalian host. *PLoS One* 8:e67069 (2013).*

*Capewell, P. et al. The skin is a significant but overlooked anatomical reservoir for vector-borne African trypanosomes. *eLife* 5, e17716 (2016).*

*Janelle, J. et al. Monitoring the pleomorphism of Trypanosoma brucei gambiense isolates in mouse: impact on its transmissibility to Glossina palpalis gambiensis. *Infect. Genet. Evol.* 9, 1260-1264 (2009).*

*Macleod, O., Bart, J. M., MacGregor, P., Peacock, L., Savill, N. J., Hester, S., Ravel, S., Sunter, J. D., Trevor, C., Rust, S., Vaughan, T. J., Minter, R., Mohammed, S., Gibson, W., Taylor, M. C., Higgins, M. K., & Carrington, M. A receptor for the complement regulator factor H increases transmission of trypanosomes to tsetse flies. *Nat Commun.* 11(1), 1326 (2020).*

*Raper, J., Nussenzweig, V., & Tomlinson, S. (1996). The main lytic factor of Trypanosoma brucei brucei in normal human serum is not high density lipoprotein. *J Exp Med.* 183(3), 1023–1029.*

Uzureau, P. et al. Mechanism of Trypanosoma brucei gambiense resistance to human serum. Nature 501, 430-434 (2013).

Vanhollebeke, B. et al. A haptoglobin-hemoglobin receptor conveys innate immunity to Trypanosoma brucei in humans. Science 320, 677-681 (2008).

Welburn, S. C., Molyneux, D. H., & Maudlin, I. Beyond Tsetse--Implications for Research and Control of Human African Trypanosomiasis Epidemics. Trends Parasitol. 32(3), 230–241 (2016).

REVIEWERS' COMMENTS

Reviewer #2 (Remarks to the Author):

I am satisfied with the response of the authors to my previous review (reviewer 2). Thank you also for extending the analysis in Figure 4 C, that supports the original conclusions.

I think one part of the Discussion is still a little clumsy in its writing and I propose an edit if the authors find it helpful (and does not change their meaning, which is not my intention).

Lines 400-402 "Moreover, the closely....without significant morphological transformation (65)." could be reworded for clarity to "Notably, the closely related zoonotic *T. congolense* infects tsetse flies without producing BS-ST (64), although these parasites generate cell-cycle arrested forms akin to stumpy forms of *T. brucei*, albeit without significant morphological transformation 965)."

REVIEWERS' COMMENTS

Reviewer #2 (Remarks to the Author):

I am satisfied with the response of the authors to my previous review (reviewer 2). Thank you also for extending the analysis in Figure 4 C, that supports the original conclusions.

I think one part of the Discussion is still a little clumsy in its writing and I propose an edit if the authors find it helpful (and does not change their meaning, which is not my intention).

Lines 400-402 "Moreover, the closely....without significant morphological transformation (65)." could be reworded for clarity to "Notably, the closely related zoonotic *T. congolense* infects tsetse flies without producing BS-ST (64), although these parasites generate cell-cycle arrested forms akin to stumpy forms of *T. brucei*, albeit without significant morphological transformation 965)."

We modified the text in the Discussion as suggested by the reviewer.